Manuscript prepared for J. Name
with version 2015/09/17 7.94 Copernicus papers of the LATEX class copernicus.cls.
Date: 14 June 2017

# Open-source sea ice drift algorithm for Sentinel-1 SAR imagery using a combination of feature-tracking and pattern-matching

Stefan Muckenhuber and Stein Sandven

Nansen Environmental and Remote Sensing Center (NERSC), Thormøhlensgate 47,
5006 Bergen, Norway

*Correspondence to:* S. Muckenhuber (stefan.muckenhuber@nersc.no)

**Abstract.** An open-source sea ice drift algorithm for Sentinel-1 SAR imagery is introduced based on the combination of feature-tracking and pattern-matching. Feature-tracking produces an initial drift estimate and limits the search area for the consecutive pattern-matching, that provides small to medium scale drift adjustments and normalised cross-correlation values. The algorithm is designed to combine the two approaches in order to benefit from the respective advantages. The considered feature-tracking method allows a computationally efficient computation of the drift field and the resulting vectors show a high degree of independence in terms of position, lengths, direction and rotation. The considered pattern-matching method on the other side allows better control over vector positioning and resolution. The pre-processing of the Sentinel-1 data has been adjusted to retrieve a feature distribution that depends less on SAR backscatter peak values. Applying the algorithm with the recommended parameter setting, sea ice drift retrieval with a vector spacing of $4\,\mathrm{km}$ on Sentinel-1 images covering $400\,\mathrm{km}$ x $400\,\mathrm{km}$, takes about 4 minutes on a standard $2.7\,\mathrm{GHz}$ processor with $8\,\mathrm{GB}$ memory. The corresponding recommended patch size for the pattern-matching step, that defines the final resolution of each drift vector is $34 \times 34$ pixels ($2.7 \times 2.7\,\mathrm{km}$). To assess the potential performance after finding suitable search restrictions, calculated drift results from 246 Sentinel-1 image pairs have been compared to buoy GPS data, collected in 2015 between $15^{th}$ January and $22^{nd}$ April and covering an area from $80.5°$ N to $83.5°$ N and $12°$ E to $27°$ E. We found a logarithmic normal distribution of the displacement difference with a median at $352.9\,\mathrm{m}$ using HV polarisation and $535.7\,\mathrm{m}$ using HH polarisation. All software requirements necessary for applying the presented sea ice drift algorithm are open-source to ensure free implementation and easy distribution.

## 1 Introduction

Sea ice drift has a strong impact on sea ice distribution on different temporal and spatial scales. Motion of sea ice due to wind and ocean currents causes convergence and divergence zones, resulting in formation of ridges and opening/closing of leads. On large scales, ice export from the Arctic and

Antarctic into lower latitudes, where the ice eventually melts away, contributes to a strong season-
ality of total sea ice coverage (IPCC, 2013). Due to a lack of ground stations in sea ice covered
areas, satellite remote sensing represents the most important tool for observing sea ice conditions
on medium to large scales. Despite the strong impact of sea ice drift and the opportunities given by
latest satellite remote sensing techniques, there is a lack of extensive ice drift data sets providing
sufficient resolution for estimating sea ice deformation on a spatial scaling of less than 5 km.

    Our main regions of interest are the ice covered seas around Svalbard and East of Greenland.
Characteristic for this area are a large variation of different ice types (Marginal Ice Zone, First Year
Ice, Multi Year Ice etc.), a strong seasonality of ice cover and a wide range of drift velocities. Focus
was put on the winter/spring period, since the area of interest experiences the highest ice cover during
this time of the year.

    Early work from Nansen (1902) established the rule-of-thumb that sea ice velocity resembles 2 %
of the surface wind speed with a drift direction of about $45°$ to the right (Northern Hemisphere) of
the wind. This wind driven explanation can give a rough estimate for instantaneous ice velocities.
However, the respective influence of wind and ocean current strongly depends on the temporal and
spatial scale. Only about 50 % of the long-term (several months) averaged ice drift in the Arctic can
be explained by geostrophic winds, whereas the rest is related to mean ocean circulation. This pro-
portion increases to more than 70 % explained by wind, when considering shorter time scales (days
to weeks). The wind fails to explain large-scale ice divergence patterns and its influence decreases
towards the coast (Thorndike and Colony, 1982).

Using GPS drift data from the International Arctic Buoy Program (IABP), Rampal et al. (2009)
analysed the general circulation of the Arctic sea ice velocity field and found that the fluctuations
follow the same diffusive regime as turbulent flows in other geophysical fluids. The monthly mean
drift using 12 h displacements was found to be in the order of 0.05 to 0.1 m/s and showed a strong
seasonal cycle with minimum in April and maximum in October. The IABP dataset also revealed a
positive trend in the mean Arctic sea ice speed of +17 % per decade for winter and +8.5 % for summer
considering the time period 1979–2007. This is unlikely to be the consequence of increased external
forcing. Instead, the thinning of the ice cover is suggested to decrease the mechanical strength which
eventually causes higher speed given a constant external forcing (Rampal et al. , 2009b).

    Fram Strait represents the main gate for Arctic ice export and high drift velocities are generally
found in this area with direction southward. Based on moored Doppler Current Meters mounted
near $79°$ N $5°$ W, Widell et al. (2003) found an average southward velocity of 0.16 m/s for the period
1996–2000. Daily averaged values were usually in the range 0–0.5 m/s with very few occasions
above 0.5 m/s.

    GPS buoys and Current Meters are important tools to measure ice drift at specific locations. How-
ever, to monitor sea ice drift on medium to large scales, satellite remote sensing represents the most
important data source today. The polar night and a high probability for cloud cover over sea ice limit

the capability of optical sensors for reliable year-round sea ice monitoring. Unlike optical sensors, Space-borne Synthetic Aperture Radar (SAR) are active sensors, operate in the microwave spectrum and can produce high resolution images regardless of solar illumination and cloud cover. Since the early 1990's SAR sensors are delivering systematic acquisitions of sea ice covered oceans and Kwok et al. (1990) showed that sea ice displacement can be calculated from consecutive SAR scenes.

The geophysical processor system from Kwok et al. (1990) has been used to calculate sea ice drift fields in particular over the Western Arctic (depending on SAR coverage) once per week with a spatial resolution of 10-25 km for the time period 1996–2012. This extensive dataset makes use of SAR data from RADARSAT-1 operated by the Canadian Space Agency, and from ENVISAT (Environmental Satellite) ASAR (Advanced Synthetic Aperture Radar) operated by ESA (European Space Agency).

To resolve drift details on a finer scale, a high-resolution sea ice drift algorithm for SAR images from ERS-1 (European Remote-sensing Satellite from ESA) based on pattern-matching was introduced by Thomas et al. (2008), that allowed drift calculation with up to 400 m resolution. Hollands and Dierking (2011) implemented their own modified version of this algorithm to derive sea ice drift from ENVISAT ASAR data.

To provide drift estimates also in areas where areal matching procedures (like cross and phase correlation) fail, Berg and Eriksson (2014) introduced a hybrid algorithm for sea ice drift retrieval from ENVISAT ASAR data using phase correlation and a feature based matching procedure that is activated if the phase correlation value is below a certain threshold.

The current generation of SAR satellites including RADARSAT-2 and Sentinel-1 are able to provide images with more than one polarisation. Komarov and Barber (2014) and Muckenhuber et al. (2016) have evaluated the sea ice drift retrieval performance with respect to the polarisation using a combination of phase/cross-correlation and feature-tracking based on corner detection respectively. Muckenhuber et al. (2016) has shown that feature-tracking provides on average around four times as many vectors using HV polarisation compared to HH polarisation.

After the successful start of the Sentinel-1 mission in early 2014, high-resolution SAR images are delivered for the first time in history within a few hours after acquisition as open-source data to all users. This introduced a new era in SAR Earth observation with great benefits for both scientists and other stack holders. Easy, free and fast access to satellite imagery facilitate the possibility to provide products on an operational basis. The Danish Technical University (Pedersen et al. (2015), http://www.seaice.dk/) provides an operational sea ice drift product based on Sentinel-1 data with 10 km resolution as part of the Copernicus Marine Environment Monitoring Service (CMEMS, http://marine.copernicus.eu).

The sea ice covered oceans in the European Arctic Sector represent an important area of interest for the Sentinel-1 mission and due to the short revisit time in the Arctic, our area of interest is monitored by Sentinel-1 on a daily basis (ESA, 2012).

This paper follows up the work from Muckenhuber et al. (2016), who published an open-source feature-tracking algorithm to derive computationally efficient sea ice drift from Sentinel-1 data based on the open-source ORB algorithm from Rublee et al. (2011), that is included in the OpenCV Python package. We aim to improve the feature-tracking approach by combining it with pattern-matching. Unlike Berg and Eriksson (2014), the feature-tracking step is performed initially and serves as a first guess to limit the search area of the pattern-matching step.

From a methodological point of view, algorithms for deriving displacement vectors between two consecutive SAR images are based either on feature-tracking or pattern-matching.

Feature-tracking detects distinct patterns (features) in both images and tries to connect similar features in a second step without the need for knowing the locations. This can be done computationally efficient and the resulting vectors are often independent of their neighbours in terms of position, lengths, direction and rotation, which can potentially be an important advantage for resolving shear zones, rotation and divergence/convergence zones. The considered feature-tracking approach identifies features without taking the position of other features into account and matches features from one image to the other without taking the drift and rotation information from surrounding vectors into account (Muckenhuber et al., 2016). However, due to the independent positioning of the features, very close features may share some pixels and since all vectors from the resolution pyramid are combined, the feature size varies among the matches, which implies a varying resolution. In addition, the resulting vector field is not evenly distributed in space and large gaps may occur between densely covered areas, which can eventually lead to missing a shear or divergence/convergence zone.

Pattern-matching, on the other hand, takes a small template from the first image at the starting location of the vector and tries to find a match on a larger template from the second image. Simple pattern-matching methods based on normalised cross-correlation often demand a considerable computational effort. Nevertheless, this approach is widely used, since it allows to define the vector positions. For practical reasons, a pyramid approach is generally used to derive high-resolution ice drift. This speeds up the processing, but potentially limits the independence of neighbouring vectors, since they depend on a lower resolution estimate (Thomas et al., 2008).

The objective of this paper is to combine the two approaches in order to benefit from the respective advantages. The main advantages of the considered feature-tracking approach are the computational efficiency and the independence of the vectors in terms of position, lengths, direction and rotation. The considered pattern-matching method on the other side allows better control over vector positioning and resolution, which is a necessity for computing divergence, shear and total deformation.

The presented algorithm, all necessary software requirements (python incl. Nansat, openCV and SciPy) and the satellite data from Sentinel-1 are open-source. A free and user friendly implementation shall support an easy distribution of the algorithm among scientists and other stakeholders.

The paper is organised as follows: The used satellite products and buoy data are introduced in Section 2. The algorithm description including data pre-processing is given in Section 3, together

with tuning and performance assessment methods. Section 4 presents the pre-processing, parameter tuning and performance assessment results and provides a recommended parameter setting for the area and time period of interest. The discussion including outlook can be found in Section 5.

## 2  Data

The Sentinel-1 mission is a joint initiative of the European Commission and the European Space Agency (ESA) and represents the Radar Observatory for the Copernicus Programme, a European system for monitoring the Earth with respect to environmental and security issues. The mission includes two identical satellites, Sentinel-1A (launched in April 2014) and Sentinel-1B (launched in April 2016), each carrying a single C-band SAR with a centre frequency of 5.405 GHz and

dual-polarisation support (HH+HV, VV+VH) also for wide swath mode. Both satellites fly in the same near-polar, sun-synchronous orbit and the revisit time is less than 1 day in the Arctic (ESA, 2012). The main acquisition mode of Sentinel-1 over sea ice covered areas is Extra Wide Swath Mode Ground Range Detected with Medium Resolution (EW GRDM) and the presented algorithm is built for processing this data type. The covered area per image is 400 km $\times$ 400 km and the data

are provided with a pixel spacing of 40 m $\times$ 40 m in both HV and HH polarisation. The introduced algorithm can utilise both HV and HH channel. However, the focus of this paper is put on using HV polarisation (mainly acquired over the European Arctic and the Baltic sea), since this channel provides in our area of interest on average four times more feature tracking vectors than HH (Muckenhuber et al., 2016), representing a better initial drift estimate for the combined algorithm.


To illustrate the algorithm performance and explain the individual steps, we use an image pair acquired over Fram Strait. The acquisition times of the two consecutive images are 2015-03-28 07:44:33 (UTC) and 2015-03-29 16:34:52 (UTC), and the covered area is shown in Figure 1. This image pair covers a wide range of different ice conditions (multiyear ice, first-year ice, marginal ice

zone etc.) and the ice situation is representative for our area and time period of interest.

To evaluate suitable search limitations and assess the potential algorithm performance, we use GPS data from drift buoys that have been set out in the ice covered waters north of Svalbard as part of the Norwegian Young Sea Ice Cruise (N-ICE2015) project of the Norwegian Polar Institute

(Spreen and Itkin, 2015). The ice conditions during the N-ICE2015 expedition are describe on the project website (http://www.npolar.no/en/projects/n-ice2015.html) as challenging. The observed ice pack, mainly consisting of 1.3-1.5 m thick multiyear and first-year ice, drifted faster than expected and was very dynamic. Closer to the ice edge, break up of ice floes has been observed due to rapid ice drift and the research camp had to be evacuated and re-established four times. This represents

a good study field, since these challenging conditions are expected in our area and time period of

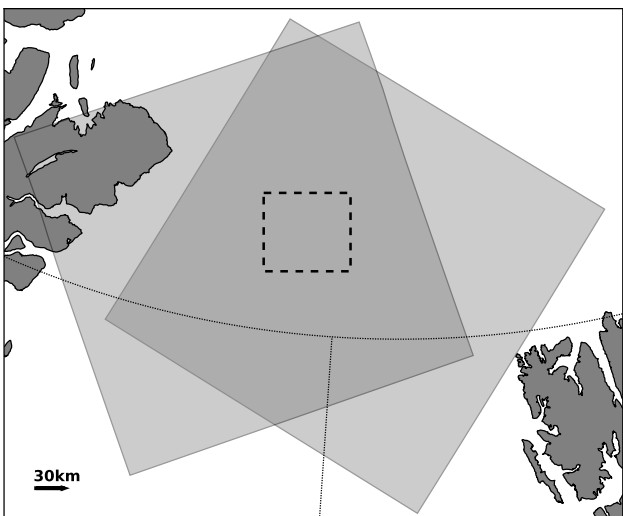

**Figure 1.** Coverage of image pair Fram Strait that is used as representative image pair to explain the algorithm approach. The dashed rectangle depicts the area shown in Figure 4 to illustrate the vector distribution of the algorithm steps.

interest. The considered GPS data have been collected in 2015 between $15^{th}$ January and $22^{nd}$ April, and cover an area ranging from $80.5°$ N to $83.5°$ N and $12°$ E to $27°$ E. The buoys recorded their positions either hourly or every three hours. In the later case, the positions have been interpolated for each hour.

## 3 Method

### 3.1 Data pre-processing

To process Sentinel-1 images within Python (extraction of backscatter values and corresponding geolocations, reprojection, resolution reduction etc.), we use the Python toolbox Nansat (Korosov et al., 2016), that builds on the Geospatial Data Abstraction Library (http://www.gdal.org). As done in Muckenhuber et al. (2016), we change the projection of the the provided ground control points (latitude/longitude values given for certain pixel/line coordinates) to stereographic and use spline interpolation to calculate geographic coordinates. This provides a good geolocation accuracy also at high latitudes. The pixel spacing of the image is changed by averaging from 40 m to 80 m, which is closer to the sensor resolution of 93 m range $\times$ 87 m azimuth, and decreases the computational effort.

For each pixel $p$, the Sentinel-1 data file provides a digital number $DN_p$ and a normalisation coefficient $A_p$, from which the normalised radar cross section $\sigma^0_{\text{raw}}$ is derived by the following equation:

$$\sigma_{\mathrm{raw}}^0 = DN_p^2/A_p^2 \tag{1}$$

The normalised radar cross section $\sigma_{\mathrm{raw}}^0$ reveals a logarithmic distribution and the structures in the sea ice are mainly represented in the low and medium backscatter values rather than in the highlights. Therefore, we change the linear scaling of the raw backscatter values $\sigma_{\mathrm{raw}}^0$ to a logarithmic scaling and get the backscatter values $\sigma^0 = 10 * \lg(\sigma_{\mathrm{raw}}^0)$ [dB]. A representative backscatter distribution over sea ice is shown in Figure 2. Using a logarithmic scaling provides a keypoint distribution for the

feature tracking algorithm that depends less on high peak values, while the total number of vectors increases.

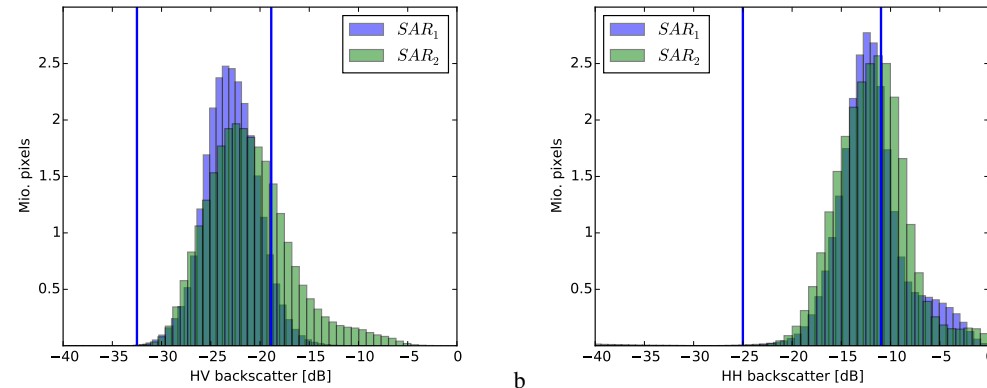

a                                                                                          b

**Figure 2.** Histogram of (a) HV and (b) HH backscatter values $\sigma^0$ from image pair Fram Strait. The lower and upper brightness boundaries for HV ($\sigma_{\mathrm{min}}^0 = -32.5\,\mathrm{dB}$, $\sigma_{\mathrm{max}}^0 = -18.86\,\mathrm{dB}$) and HH ($\sigma_{\mathrm{min}}^0 = -25.0\,\mathrm{dB}$, $\sigma_{\mathrm{max}}^0 = -10.97\,\mathrm{dB}$) are shown with blue lines and illustrate the domain for the intensity values $i$.

    To apply the feature-tracking algorithm from Muckenhuber et al. (2016), the SAR backscatter values $\sigma^0$ have to be converted into intensity values $i$ with $0 \leq i \leq 255$ for $i \in \mathbb{R}$. This conversion is done by using Eq. (2) and setting all values outside the domain to 0 and 255.

$$i = 255 \cdot \frac{\sigma^0 - \sigma_{\mathrm{min}}^0}{\sigma_{\mathrm{max}}^0 - \sigma_{\mathrm{min}}^0} \tag{2}$$

    The upper brightness boundary $\sigma_{\mathrm{max}}^0$ is set according to the recommended value from Muckenhuber et al. (2016), i.e. -18.86 dB and -10.97 dB for HV and HH respectively. The lower boundary $\sigma_{\mathrm{min}}^0$ was chosen to be -32.5 dB (HV) and -25.0 dB (HH), since this was found to be a reasonable range of expected backscatter values. Figure 3 shows the image pair Fram Strait after the conversion

into intensity values. For the sake of computational efficiency, the same intensity value scaling is used for the pattern-matching step.

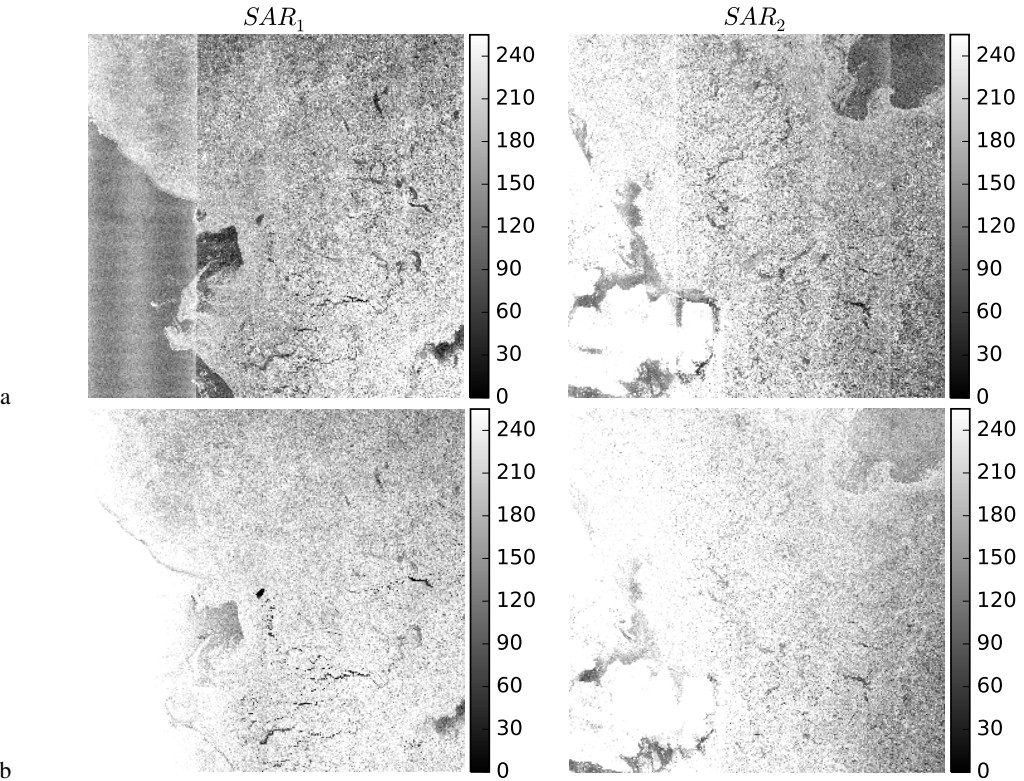

**Figure 3.** Image pair Fram Strait in (a) HV and (b) HH polarisation after conversion (Equation 2) from backscatter values $\sigma^0$ into intensity values with range $0 \leq i \leq 255$ using lower and upper brightness boundaries for HV: $\sigma^0_{\min} = -32.5\,\mathrm{dB}$ and $\sigma^0_{\max} = -18.86\,\mathrm{dB}$ and HH: $\sigma^0_{\min} = -25.0\,\mathrm{dB}$, $\sigma^0_{\max} = -10.97\,\mathrm{dB}$.

### 3.2 Sea ice drift algorithm

The presented sea ice drift algorithm is based on a combination of feature-tracking and pattern-matching, and is designed to utilise the respective advantages of the two considered approaches.
Computationally efficient feature-tracking is used to derive a first estimate of the drift field. The provided vectors serve as initial search position for pattern-matching, that provides accurate drift vectors at each given location including rotation estimate and maximum cross-correlation value. As illustrated in the flowchart in Figure 4, the algorithm consists of five main steps: I Feature tracking, II Filter, III First guess, IV Pattern matching and V Final drift product.


**I Feature-tracking**

The feature-tracking algorithm used in this work is an adjusted version from Muckenhuber et al. (2016), who introduced a computationally efficient sea ice drift algorithm for Sentinel-1 based on
the ORB (Oriented FAST and Rotated BRIEF) algorithm from Rublee et al. (2011). ORB uses the

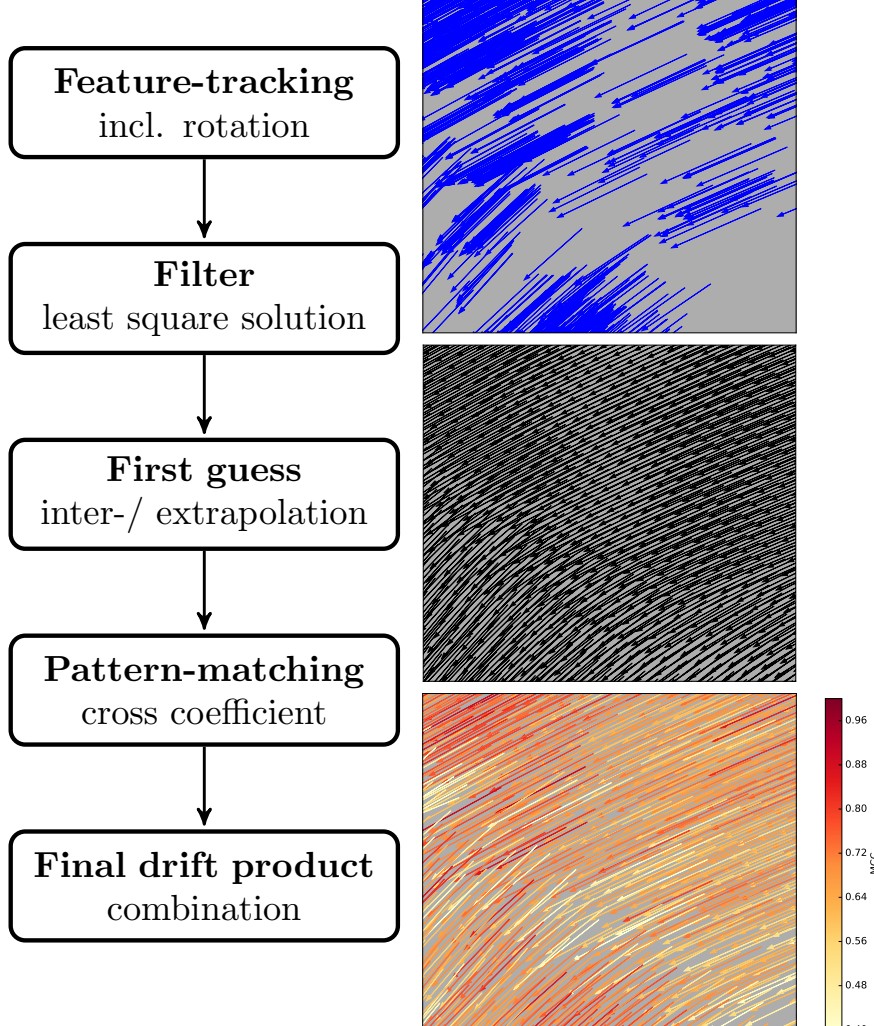

**Figure 4.** The flowchart on the left depicts the five main steps of the algorithm. The right column illustrates the evolution of the drift results using image pair Fram Strait in HV polarisation and a grid with 4 km spacing. (NB: the part of the image pair that is depicted here is marked with a dashed rectangle in Figure 1.) Blue vectors are derived applying an adjusted version of the feature tracking algorithm from Muckenhuber et al. (2016). Black vectors indicate the initial drift estimate (first guess) based on filtered feature-tracking vectors. The final drift product (yellow to red vectors) are derived from combining the first guess with pattern-matching adjustment and applying a minimum cross-correlation value. A total of 4725 vectors have been found on image pair Fram Strait with a $MCC$ value above 0.4 in 4 min.

concept of the FAST keypoint detector (Rosten and Drummond, 2006) to find corners on several resolution levels. The patch around each corner is then described using an modified version of the binary BRIEF descriptor from Calonder et al. (2010). To ensure rotation invariance, the orientation of the patch is calculated using the intensity-weighted centroid. Muckenhuber et al. (2016) applies
a Brute Force matcher that compares each feature from the first image to all features in the second

image. The comparison of two features is done using the Hamming distance, that represents the number of positions in which the two compared binary feature vectors differ from each other. The best match is accepted if the ratio of the shortest and second shortest Hamming distances is below a certain threshold. Given a suitable threshold (and unique features), the ratio test will discard a high number of false matches, while eliminating only a few correct matches.

Muckenhuber et al. (2016) found a suitable parameter setting for our area and time period of interest, including a Hamming distance threshold of 0.75, a maximum drift filter of $0.5\,\mathrm{m/s}$, a patch size of $34 \times 34$ pixels and a resolution pyramid with 7 steps combined with a scaling factor of 1.2. Due to the resolution pyramid, the considered feature area varies from $2.7 \times 2.7\,\mathrm{km}$ to $9.8 \times 9.8\,\mathrm{km}$ and the resulting drift field represents a resolution mixture between these boundaries.

We adjust the algorithm from Muckenhuber et al. (2016) by applying a logarithmic scaling for the SAR backscatter values $\sigma_0$ instead of the previous used linear scaling (Section 3.1). In addition, we extract for each vector the rotation information $\alpha$, i.e. how much the feature rotates from the first to the second image.

Applying the adjusted feature-tracking algorithm provides a number of un-evenly distributed vectors (e.g. blue vectors in Figure 4) with start positions $x_{1f}$, $y_{1f}$ on the first image ($SAR_1$), end positions $x_{2f}$, $y_{2f}$ on the subsequent image ($SAR_2$) and corresponding rotation values $\alpha_{raw\,f}$. The index $f$ represents a feature-tracking vector and ranges from 1 to $F$, with $F$ being the total number of derived feature-tracking vectors. For the sake of computational efficiency, the vectors from all resolution pyramid levels are treated equally.

To avoid zero-crossing issues during the following filter and inter-/extrapolation process (in case the image rotation $\delta$ between $SAR_1$ and $SAR_2$ is close to $0°$), a factor $|180 - \delta|$ is added to the raw rotation values $\alpha_{raw\,f}$ using the following Equation:

$$\alpha_f = \begin{cases} \alpha_{raw\,f} + |180 - \delta| & \text{if } \alpha_{raw\,f} + |180 - \delta| < 360 \\ \alpha_{raw\,f} + |180 - \delta| - 360 & \text{if } \alpha_{raw\,f} + |180 - \delta| > 360 \end{cases} \tag{3}$$

This centres the reasonable rotation values in the proximity of $180°$. After applying the filter and inter-/extrapolation process, the estimated rotation $\alpha$ is corrected by subtracting $|180 - \delta|$.

**II Filter**

To reduce the impact of potentially erroneous feature-tracking vectors on the following steps, outliers are filtered according to drift and rotation estimates derived from least squares solutions

using a third degree polynomial function. Considering a matrix $\mathbf{A}$, that contains all end positions $x_{2f}$, $y_{2f}$ in the following form

$$\mathbf{A} = \begin{pmatrix} 1 & x_{21} & y_{21} & x_{21}^2 & y_{21}^2 & x_{21}*y_{21} & x_{21}^3 & y_{21}^3 \\ 1 & x_{22} & y_{22} & x_{22}^2 & y_{22}^2 & x_{22}*y_{22} & x_{22}^3 & y_{22}^3 \\ \vdots & \vdots & \vdots & \vdots & \vdots & \vdots & \vdots & \vdots \\ 1 & x_{2F} & y_{2F} & x_{2F}^2 & y_{2F}^2 & x_{2F}*y_{2F} & x_{2F}^3 & y_{2F}^3 \end{pmatrix} \qquad (4)$$

, we derive three vectors $\boldsymbol{b}_{x_1}$, $\boldsymbol{b}_{y_1}$ and $\boldsymbol{b}_\alpha$, that represent the least squares solutions for $\mathbf{A}$ and $\boldsymbol{x}_1 = (x_{11}, ..., x_{1F})$, $\boldsymbol{y}_1 = (y_{11}, ..., y_{1F})$ and $\alpha = (\alpha_1, ..., \alpha_F)$ respectively. The starting position $x_{1f}$, $y_{1f}$ and the rotation $\alpha_f$ of each vector can then be simulated using a third degree polynomial function $f(x_{2f}, y_{2f}, \boldsymbol{b})$ depending on the end position $x_{2f}$, $y_{2f}$ and the corresponding least squares solution
$\boldsymbol{b} = (b_0, b_1, b_2, b_3, b_4, b_5, b_6, b_7)$.

$$f(x_{2f}, y_{2f}, \boldsymbol{b}) = b_0 + b_1 x_{2f} + b_2 y_{2f} + b_3 x_{2f}^2 + b_4 y_{2f}^2 + b_5 x_{2f} y_{2f} + b_6 x_{2f}^3 + b_7 y_{2f}^3 \qquad (5)$$

If the simulated start position, derived from $f(x_{2f}, y_{2f}, \boldsymbol{b})$, deviates from the feature-tracking start position $x_{1f}$, $y_{1f}$ by more than 100 pixels, the vector is deleted. The same accounts for rotation outliers. If the simulated rotation deviates from the feature-tracking rotation $\alpha_f$ by more than $60°$,
the vector is deleted. We found a third degree polynomial function to be a good compromise between allowing for small to medium scale displacement and rotation discontinuities, and excluding very unlikely vectors, that eventually would disturb the following steps. The parameters for the filter process, i.e. 100 pixels (displacement) and $60°$ (rotation), have been chosen according to visual interpretation using several representative image pairs. Figure 5 illustrates the filter process by
depicting the results from image pair Fram Strait.

### III First guess

The remaining feature-tracking vectors are used to estimate the drift incl. rotation on the entire
first image, i.e. estimated $x_2$, $y_2$ and $\alpha$ values are provided for each pixel on SAR$_1$ (Figure 6). The quality of this 'first guess', however depends on the density of the feature-tracking vector field and the local ice conditions.

Between the feature-tracking vectors, estimated values are constructed by triangulating the input data and performing linear barycentric interpolation on each triangle. That means, the estimated val-
ues represent the weighted mean of the three neighbouring feature-tracking values. The interpolated value $v_p$ at any pixel $p$ inside the triangle is given by Equation 6, where $v_1$, $v_2$, $v_3$ represent the feature-tracking values at the corners of the triangle and $A_1$, $A_2$, $A_3$ are the areas of the triangle

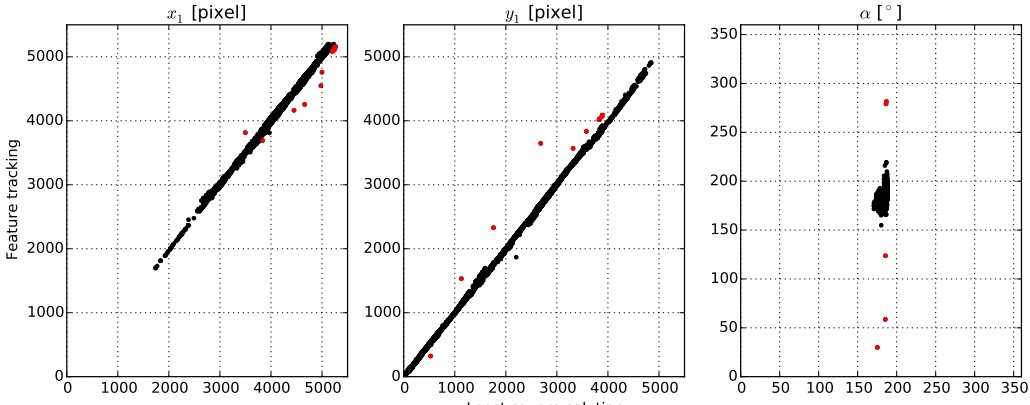

**Figure 5.** Filter process applied on image pair Fram Strait in HV polarisation. The x-axis represent the simulated start position and rotation, derived from $f(x_{2f}, y_{2f}, \boldsymbol{b})$ and the y-axis represent the feature-tracking start position $x_{1f}$, $y_{1f}$ and rotation $\alpha_f$. NB: the image rotation is $\delta = 129.08°$, which means the rotation was adjusted by $50.92°$ (Equation 3). Red points were identified as outliers and deleted.

constructed by $p$ and the two opposite corners, e.g. $A_1$ is the area between $p$, and the corners with value $v_2$ and $v_3$.

$$v_p = \frac{A_1 v_1 + A_2 v_2 + A_3 v_3}{A_1 + A_2 + A_3} \qquad (6)$$

To provide a first guess for the surrounding area, values are estimated based on the least squares solutions using a linear combination of $x_1$ and $y_1$. Considering a matrix $\mathbf{C}$, that contains all start positions $x_{1f}$, $y_{1f}$ in the following form

$$\mathbf{C} = \begin{pmatrix} 1 & x_{11} & y_{11} \\ 1 & x_{12} & y_{12} \\ \vdots & \vdots & \\ 1 & x_{1F} & y_{1F} \end{pmatrix} \qquad (7)$$

, we derive three vectors $\boldsymbol{d}_{x_2}$, $\boldsymbol{d}_{y_2}$ and $\boldsymbol{d}_{\alpha}$, that represent the least squares solutions for $\mathbf{C}$ and $\boldsymbol{x}_2 = (x_{21}, ..., x_{2F})$, $\boldsymbol{y}_2 = (y_{21}, ..., y_{2F})$ and $\alpha = (\alpha_1, ..., \alpha_F)$ respectively. The estimated end position $x_2$, $y_2$ and rotation $\alpha$ at any location can then be simulated using the linear function $f(x_1, y_1, \boldsymbol{c})$ depending on the start position $x_1$, $y_1$ and the corresponding least squares solution $\boldsymbol{d} = (d_0, d_1, d_2)$.

$$f(x_1, y_1, \boldsymbol{d}) = d_0 + d_1 x_1 + d_2 y_1 \qquad (8)$$

As mentioned above, the rotation estimates $\alpha$ are now corrected for the adjustment applied in Equation 3, by subtracting $|180 - \delta|$.

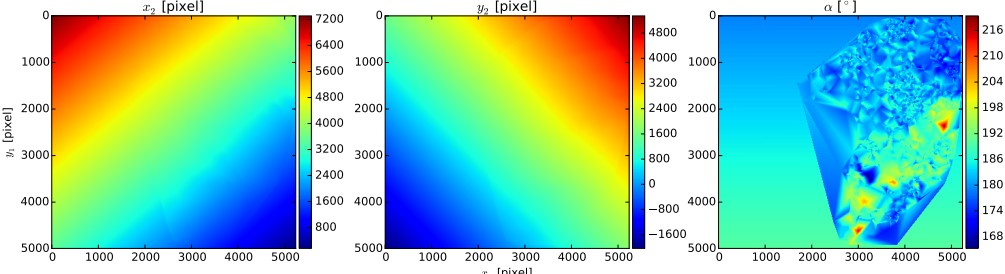

**Figure 6.** Example of estimated drift and rotation (first guess) based on filtered feature-tracking vectors using image pair Fram Strait in HV polarisation. The three panels show the components $x_2$, $y_2$ of the estimated end positions and the estimated rotation $\alpha$ for each pixel on on the coordinate system $x_1$, $y_1$ of the first image (SAR$_1$).

An example for the resulting first guess, i.e. estimated values for $x_2$, $y_2$ and $\alpha$ on SAR$_1$, is shown in Figure 6 (this figure illustrates the matrices that the algorithm considers as first guess) and corresponding vectors are shown in black in Figure 4. Note that rotation $\alpha$ has already been corrected by subtracting $|180-\delta|$. It includes now both the relative image rotation $\delta$ from $SAR_1$ to $SAR_2$ and the actual rotation of the feature itself. The introduced algorithm provides also the image rotation $\delta$ by projecting the left corners of $SAR_2$ onto $SAR_1$ and calculating the angle between the left edges of $SAR_1$ and $SAR_2$. The actual rotation of the features can easily be obtained by subtracting $\delta$ from $\alpha$.

## IV Pattern-matching

The estimated drift field derived from feature-tracking provides values for $x_2$, $y_2$ and $\alpha$ at any location on $SAR_1$. The representativeness of this estimate however, depends on the distance $d$ to the closest feature-tracking vector. Therefore, small to medium scale adjustments of the estimates are necessary, depending on the distance $d$ (NB: the representativeness also depends on the variability of the surrounding vectors, but for the sake of computational efficiency, we only consider the distance $d$ as representativeness measure). We apply pattern-matching at chosen points of interest to adjust the drift and rotation estimate at these specific locations.

The used pattern-matching approach is based on the maximisation of the normalised cross-correlation coefficient. Considering a small template $t_1$ around the point of interest from $SAR_1$ with size $t_{1s} \times t_{1s}$ and a larger template $t_2$ around the location $x_2$, $y_2$ (defined by the corresponding first guess) from $SAR_2$ with size $t_{2s} \times t_{2s}$, the normalised cross-correlation matrix **NCC** is defined as (Hollands , 2012):

$$\mathbf{NCC}(x,y) = \frac{\sum_{x',y'}(t_1'(x',y')t_2'(x+x',y+y'))}{\sqrt{\sum_{x',y'}t_1'(x',y')^2 \sum_{x',y'}t_2'(x+x',y+y'))^2}} \tag{9}$$

$$t_1'(x',y') = t_1(x',y') - \frac{1}{t_{1s}^2}\sum_{x'',y''} t_1(x'',y'') \tag{10}$$

$$t_2'(x+x',y+y') = t_2(x+x',y+y') - \frac{1}{t_{1s}^2}\sum_{x'',y''} t_2(x+x'',y+y'') \tag{11}$$

with $t_1(x',y')$ and $t_2(x',y')$ representing the value of $t_1$ and $t_2$ at location $x',y'$. The summations are done over the size of the smaller template, i.e. $x'$, $y'$, $x''$ and $y''$ go from 1 to $t_{1s}$. Template $t_1$ is
moved with step size 1 pixel over template $t_2$ both in horizontal ($x$) and vertical ($y$) direction and the cross-correlation values for each step are stored in the matrix **NCC** with size $(1 + t_{s2} - t_{s1}) \times (1 + t_{s2} - t_{s1})$. The highest value in the matrix **NCC**, i.e. the the maximum normalised cross-correlation value $MCC$, represents the location of the best match and the corresponding location adjustment is given by $dx$ and $dy$.

$$\left(\frac{1 + t_{s2} - t_{s1}}{2} + dx, \frac{1 + t_{s2} - t_{s1}}{2} + dy\right) = \underset{x,y}{argmax}(\mathbf{NCC}(x,y)) \tag{12}$$

To restrict the search area $t_{2s}$ to a circle, we set all values of **NCC** that are further than $t_{2s}/2$ away from the centre position to zero. This limits the distance from the first guess to a constant value, rather than to an arbitrary value depending on the looking angle of the satellite. c To account for rotation adjustment, the matrix **NCC** is calculated several times: template $t_1$ is rotated around the
initially estimated rotation $\alpha$ from $\alpha - \beta$ to $\alpha + \beta$ with step size $\Delta\beta$. The angle $\beta$ is the maximum additional rotation and represents therefore the rotation restriction. The **NCC** matrix with the highest cross-correlation value $MCC$ is returned.

To illustrate the pattern-matching process, an example, taken from image pair Fram Strait, is shown in Figure 7.


The described process demands the specification of four parameters: $t_{1s}$, $t_{2s}$, $\beta$ and $\Delta\beta$.

The size of the small template $t_{1s} \times t_{1s}$ defines the considered area that is tracked from one image to the next and hence, affects the resolution of the resulting drift product. Sea ice drift might be different on different resolution scales. This is particularly an issue in the case of rotation. The
feature-tracking vectors provide the first guess and this vector field should represent the same drift resolution as considered by the pattern-matching step. In order to be consistent with the resolution of the feature-tracking step and achieve our goal of a sea ice drift product with a spatial scaling of less than 5 km, we use the size of the feature-tracking patch of the pyramid level with the highest resolution to define the size of $t_1$. That means, we use $t_{s1} = 34$ pixels (2.7 km).
The size of the larger template $t_{2s} \times t_{2s}$ restricts the search area on $SAR_2$, i.e. how much the first guess can be adjusted geographically, and the angle $\beta$ restricts the rotation adjustment of the

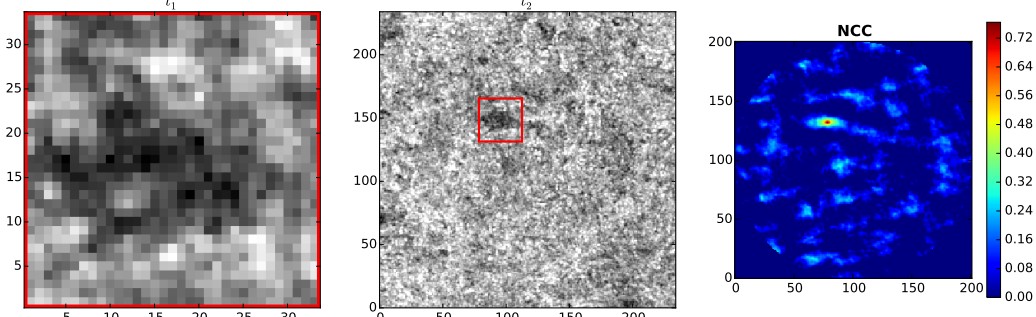

**Figure 7.** Pattern-matching using initial drift estimate from feature-tracking: The small template $t_1$ (left) around the point of interest on SAR$_1$ is rotated from $\alpha - \beta$ to $\alpha + \beta$ and matched with the large template $t_2$ (middle) from SAR$_2$, that has its centre at the estimated end position $x_2, y_2$. The right contour plot shows the normalised cross-correlation matrix **NCC** of the rotation $\beta^*$ that provided the highest maximum cross-correlation coefficient $MCC$. The estimated end position $x_2, y_2$ of this example has to be adjusted by $dx = -21$ pixels, $dy = 32$ pixels to fit with the location of $MCC = 0.71$. Rotation adjustment $\beta^*$ was found got be $3°$. NB: $X$ and $Y$-axis represent pixel coordinates.

first guess $\alpha$. The three parameter $t_{2s}$, $\beta$ and $\Delta\beta$ have a strong influence on the computational efficiency of the drift algorithm. Meaning that an increase of $t_{2s}$, $\beta$ and a decrease of $\Delta\beta$ increase the computational effort of the pattern-matching step. Based on visual interpretation of several representative image pairs, we found $\Delta\beta = 3°$ to be a good compromise between matching performance and computational efficiency.

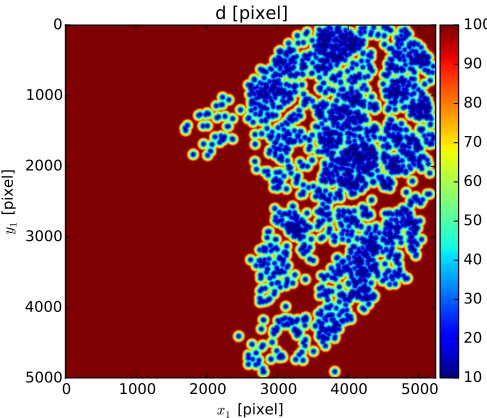

**Figure 8.** Example to illustrate the distribution of distance $d$ to the closest feature-tracking vector using image pair Fram Strait in HV polarisation. Values outside the range $d_{min} \leq d \leq d_{max}$ are set to $d_{min} = 10$ and $d_{max} = 100$. The points with value $d_{min}$ represent the start positions $x_{1f}, y_{1f}$ of the feature-tracking vectors on the coordinate system $x_1, y_1$ of $SAR_1$. The figure depicts the matrix that the algorithm considers for the distribution of $d$.

Since the representativeness of the first guess decreases with distance $d$ to the closest feature-tracking vector (an example to illustrate the distribution of $d$ is shown Figure 8), the search restrictions $t_{2s}$ and $\beta$ should increase with $d$. Based on the performed search restriction evaluation (Section 4), we found the following functions to represent useful restrictions for our area and time period of interest.

$$t_{2s}(d) = t_{1s} + 2d \qquad d_{min} \leq d \leq d_{max} \qquad d \in \mathbb{N} \tag{13}$$

$$\beta(d) = \begin{cases} 9 & \text{if } d < d_{max} \\ 12 & \text{if } d \geq d_{max} \end{cases} \tag{14}$$

The values for $d_{min}$, $d_{max}$, $\beta$ and $\Delta\beta$ can easily be varied in the algorithm to adjust for e.g. different areas, drift conditions or a different compromise between matching performance and computational efficiency.

**V Final drift product**

In the last step, the small to medium scale displacement adjustments from pattern-matching are added to the estimated first guess derived from feature-tracking. Using buoy comparison, we found that the probability for large displacement errors decreases with increasing $MCC$ value (Section 4). Therefore, vectors that have a $MCC$ value below the threshold $MCC_{min}$ are removed. We found $MCC_{min} = 0.4$ to be a good filter value, but this value can easily be adjusted in the algorithm depending on the sought compromise between amount of vectors and error probability. The algorithm returns the final drift vectors in longitude, latitude, the corresponding first guess rotation $\alpha$ and the rotation adjustment $\beta$ in degrees and the maximum cross-correlation value $MCC$. An example for the final product is depicted with yellow to red coloured vectors in Figure 4. The colour scale refers to the $MCC$ value, indicating the probability for an erroneous vector.

**3.3 Comparison with buoy data**

Sentinel-1 image pairs have been selected automatically according to position and timing of the GPS buoy data from the N-ICE2015 expedition. Each pair yielded more than 300 drift vectors applying the feature-tracking algorithm from Section 3.2 and had a time difference between the two acquisitions of less than three days. Drift vectors have been calculated with the presented algorithm starting at the buoy GPS position with the least time difference to the acquisition of the first satellite image. The distance $D$ between the calculated end position on the second image and the buoy GPS posi-

tion with the least time difference to the second satellite acquisition has been calculated using the following equation:

$$D = \sqrt{(u-U)^2 + (v-V)^2} \qquad (15)$$

395    where $u$ and $v$ represent eastward and northward drift components of the displacement vector derived by the algorithm, and $U$ and $V$ the corresponding drift components of the buoy.

## 4   Results

### 4.1   Search restrictions evaluation

To find suitable values for restricting the size of the search window $t_{2s}$ and the rotation range de-
400   fined by $\beta$, we calculated drift vectors, that can be compared to the considered GPS buoy dataset, using restrictions that are computationally more demanding than we anticipate for the recommended setting, i.e. $t_{2s} = 434$ pixels and $\beta = 18°$. These values corresponds to a possible pattern-matching adjustment of up to 200 pixels (16 km) and $18°$ in any direction independent of the distance $d$ to the closest feature-tracking vector.

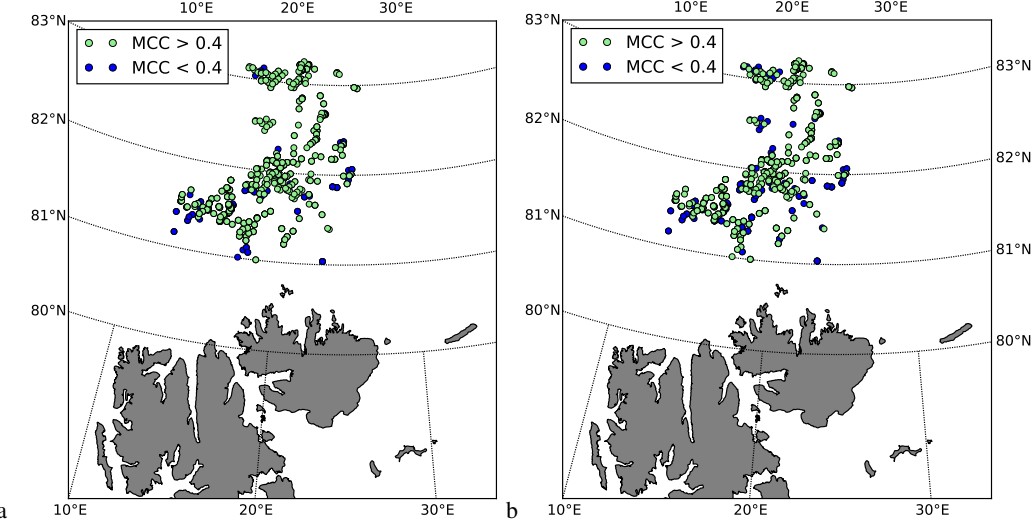

**Figure 9.** Considered buoy locations from the N-ICE2015 expedition that were used for comparison with algorithm results. Green and blue colour indicates start locations (on $SAR_1$) to which the algorithm provided vectors with a $MCC$ value above and below 0.4 using (a) HV and (b) HH polarisation.

405    Based on an automatic search, we found 244 matching Sentinel-1 image pairs (consisting of 111 images), that allowed for comparison with 711 buoy vectors (buoy locations are shown in Figure 9). The distance $D$ (Equation 15) between the buoy location at the time of the second image $SAR_2$

and the corresponding algorithm result, represents the error estimate for one vector pair. To identify algorithm results that are more likely erroneous, vector pairs with a value $D$ above 1000 m are marked with red dots in Figure 10 and Figure 11. Vector pairs with $D < 1000$ m are plotted with black dots.

Figure 10 and Figure 11 show the resulting pattern-matching adjustment of location $(dx, dy)$ and rotation $(d\beta)$ using the computationally demanding restrictions. The values are plotted against distance $d$ to the next feature tracking vector in order to identify the dependence of the parameters on $d$. The blue lines in Figure 10 and Figure 11 indicate the recommended restrictions. This represents a compromise between computational efficiency and allowing the algorithm to adjust the first guess as much as needed for our time period and area of interest. The corresponding functions for $t_{2s}(d)$ and $\beta(d)$ are given in Equation 13 and Equation 14 and the recommended boundary values for distance $d$ are $d_{min} = 10$ and $d_{max} = 100$.

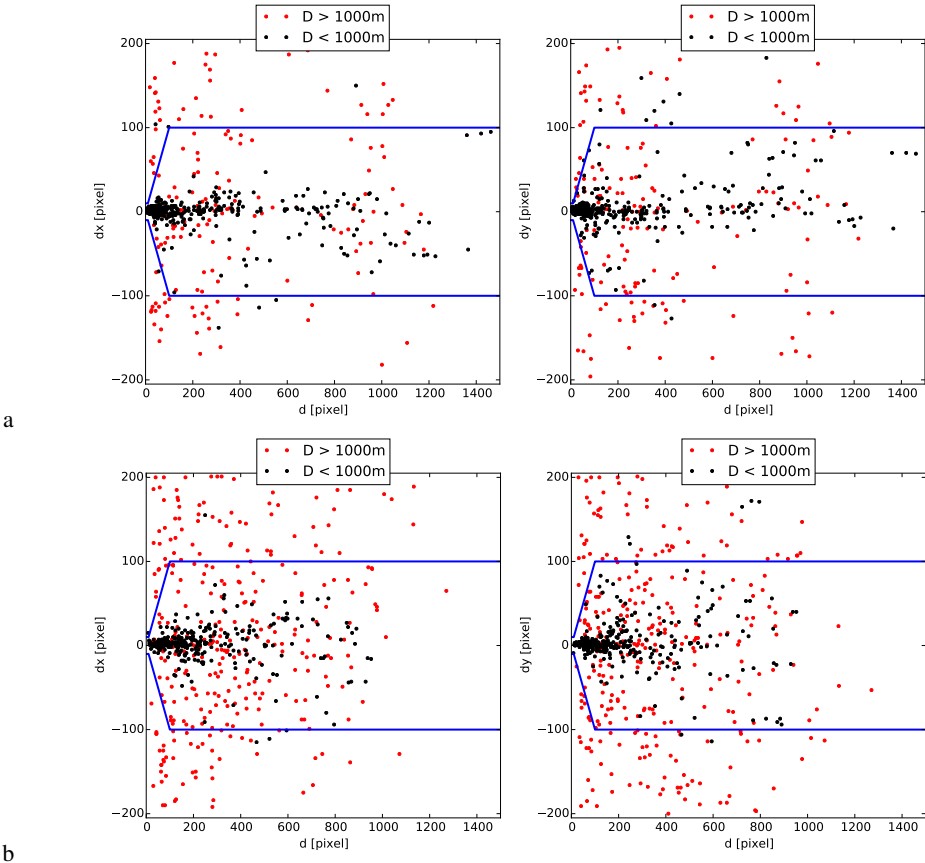

**Figure 10.** Pattern-matching location adjustment $dx$ and $dy$ in $x$ and $y$ direction versus distance $d$ to closest feature tracking vector using (a) HV and (b) HH polarisation. $D$ represents the difference between buoy GPS position and algorithm result. The blue lines indicate the recommended setting for $t_{2s}$ (Equation 13) with $d_{min} = 10$ and $d_{max} = 100$.

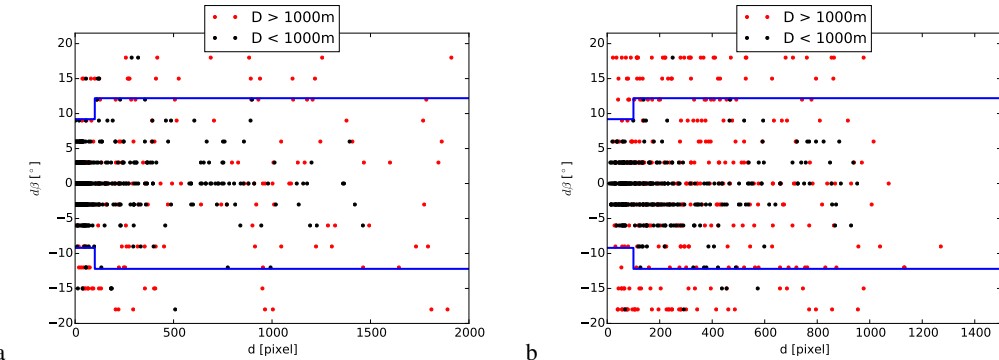

a   b

**Figure 11.** Pattern-matching rotation adjustment $d\beta$ versus distance $d$ to closest feature tracking vector using (a) HV and (b) HH polarisation. $D$ represents the difference between buoy GPS position and algorithm result. The blue lines indicate the recommended setting for $\beta$ (Equation 14) with $d_{min} = 10$ and $d_{max} = 100$.

## 4.2 Performance assessment

Using the recommended search restrictions from above, the algorithm has been compared to the N-ICE2015 GPS buoy data set (Figure 9) to assess the potential performance after finding suitable search restrictions for the area and time period of interest. The automatic search provided 246 image pairs (consisting of 111 images) and 746 vectors for comparison for the considered time period (15$^{th}$ January to 22$^{nd}$ April) and area (80.5° N to 83.5° N and 12° E to 27° E). NB: this is a higher number of vectors than found for the evaluation of the search restrictions, since the used search windows $t_2$ are smaller and vectors closer to the SAR edge may be included.

The results of the conducted performance assessment are shown in Figure 12. We found that the probability for a large $D$ value (representative for the error) decreases with increasing maximum cross-correlation value $MCC$. Therefore we suggest to exclude matches with a $MCC$ value below a certain threshold $MCC_{min}$. This option is embedded into the algorithm, but can easily be adjusted or turned off by setting $MCC_{min} = 0$. Based on the findings shown in Figure 12, we recommend a cross-correlation coefficient threshold $MCC_{min} = 0.4$ for our time period and area of interest. Using the suggested threshold reduces the number of vector pairs from 746 to 588 for the HV channel and to 478 for the HH channel.

The conducted performance assessment also reveals a logarithmic normal distribution of the distance $D$ (Equation 15) that can be expressed by the following probability density function (solid red line in Figure 12):

$$lnN(D; \mu, \sigma) = \frac{1}{\sigma D \sqrt{2\pi}} e^{-\frac{(\ln D - \mu)^2}{2\sigma^2}} \tag{16}$$

with $\mu$ and $\sigma$ being the mean and standard deviation of the variable's natural logarithm. We found the mean and variance of the distribution $lnN$ to be $\mu = 5.866$ and $\sigma^2 = 1.602$ for HV polarisation

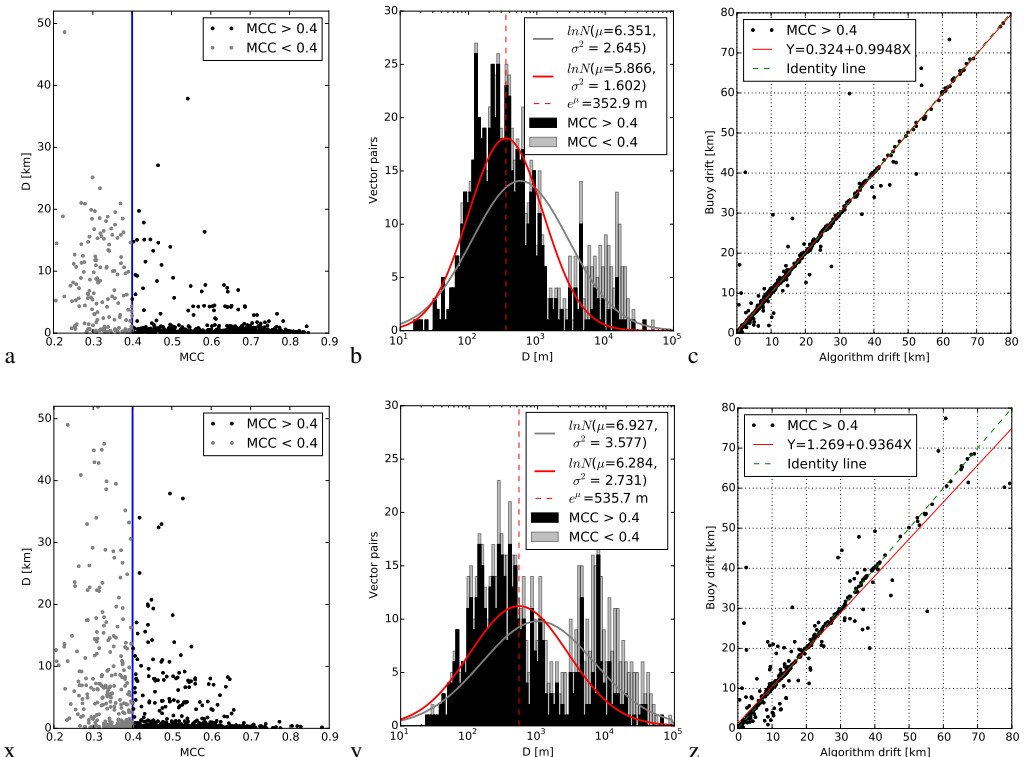

**Figure 12.** Calculated ice drift using recommended search restrictions compared to buoy GPS data using (a,b,c) HV and (x,y,z) HH polarisation. Light grey represents vectors with maximum cross-correlation values $MCC < 0.4$ and results after using the suggested threshold $MCC_{min} = 0.4$ are shown in black. (a,x) $MCC$ values against distance $D$ (Equation 15) between algorithm and buoy end position. The blue line indicates the recommended setting for $MCC_{min} = 0.4$. (b,y) Logarithmic histogram of distance $D$ with 100 bins between 10 m and $10^5$ m including two logarithmic normal distributions that were fitted to all results (grey) and to the filtered results with $MCC > 0.4$ (solid red line). (c,z) Comparison of drift distance derived from algorithm against buoy displacement for the filtered results with $MCC > 0.4$.

and $\mu = 6.284$ and $\sigma^2 = 2.731$ for HH polarisation (solid red lines in Figure 12). The medians of the logarithmic normal distribution are $e^\mu = 352.9\,m$ for HV polarisation and $e^\mu = 535.7\,m$ for HH polarisation (dashed red lines in Figure 12).

### 4.3 Recommended parameter setting

Based on the restriction evaluation, our experience with the algorithm behaviour, and considering a good compromise between computational efficiency and high quality of the resulting vector field, we recommend the parameter setting shown in Table 1 for our area and time period of interest. The corresponding recommended values for $t_{2s}(d)$ and $\beta(d)$ are given in Equation 13 and Equation 14.

**Table 1.** Recommended parameter setting for sea ice drift retrieval from Sentinel-1 using the presented algorithm.

| Parameter | Meaning | Recommended setting |
|---|---|---|
| $[\sigma^0_{\min}, \sigma^0_{\max}]$ (HH) | Brightness boundaries for HH channel | [-25 dB, -10.97 dB] |
| $[\sigma^0_{\min}, \sigma^0_{\max}]$ (HV) | Brightness boundaries for HV channel | [-32.5 dB, -18.86 dB] |
| $t_{1s}$ | Size of template $t_1$ | 34 pixels (2.7 km) |
| $[d_{min}, d_{max}]$ | Boundaries for distance $d$ | [10 pixels, 100 pixels] |
| $MCC_{min}$ | Threshold for cross-correlation | 0.4 |
| $\Delta\beta$ | Rotation angle increment | $3°$ |

## 4.4 Computational efficiency

The processing time depends on the parameter setting and the chosen vector distribution. Using the recommended parameter setting from Table 1, allows high-resolution sea ice drift retrieval from a Sentinel-1 image pair within a few minutes. Figure 4 depicts calculated ice drift vectors for the image pair Fram Strait on a grid with 4 km (50 pixels) spacing. The corresponding processing times are shown in Table 2. The calculations have been done using a MacBook Pro from early 2013 with a 2.7 GHz Intel Core i7 processor and 8 GB 1600 MHz DDR3 memory. The total processing time for 4725 vectors with a normalised cross-correlation value above 0.4, is about 4 minutes. This can be considered a representative value for an image pair with large overlap, good coverage with feature-tracking vectors and 4 km grid spacing.

The initial process in Table 2 'Create Nansat objects from Sentinel-1 image pair and read matrixes' takes the same amount of computational effort for all image pairs consisting of Sentinel-1 images with 400x400 km coverage.

The process 'I Feature-tracking' depends on the setting of the feature-tracking algorithm and varies strongly with the chosen number of features. Using the recommended setting from Muckenhuber et al. (2016), that includes the number of features to be 100000, the presented computational effort can be considered representative for all image pairs, independent of chosen points of interest and overlap of the SAR scenes.

The last process 'II Pattern-matching and III Combination' however, depends on the considered image pair and the chosen drift resolution. The computational effort is proportional to the number of chosen points of interest. Given an evenly distributed grid of points of interest, the computational effort increases with overlapping area of the SAR scenes, since pattern-matching adjustments are only calculated in the overlapping area. The effort potentially decreases with a higher number of well distributed feature-tracking vectors, since the size of the search windows $t_2$ (and slightly the range of the angle $\beta$) increases with distance $d$ to the closest feature-tracking vector.

**Table 2.** Processing time for sea ice drift retrieval from image pair Fram Strait on a grid with 4 km (50 pixels) spacing using HV polarisation (Figure 4). Representative for an image with large overlap and good coverage with feature-tracking vectors.

| Process | Time [s] |
|---|---|
| Create Nansat objects from Sentinel-1 image pair and read matrixes | 70 |
| I Feature-tracking | 66 |
| II Pattern-matching and III Combination | 107 |
| $\sum$      Sea ice drift retrieval | 243 |

## 5 Discussion and outlook

To estimate the potential performance of the introduced algorithm for given image pairs, given ice conditions, given region and given time, we compared drift results from 246 Sentinel-1 image pairs with corresponding GPS positions from the N-ICE2015 buoy data set. We found a logarithmic error distribution with a median at 352.9 m for HV and 535.7 m for HH (Figure 12). The derived error values represent a combination of the following error sources:

- Timing: Buoy GPS data were collected every 1-3 hours and the timing does not necessarily match with the satellite acquisition time.

- Resolution: The algorithm returns the drift of a pattern (recommended size = 34 pixels, see Table 1), whereas the buoy measures the drift at a single location.

- Conditions: The ice conditions around the buoy is not known well enough to exclude the possibility that the buoy is floating in a lead. In this case, the buoy trajectory could represent a drift along the lead rather then the drift of the surrounding sea ice.

- actual error of the algorithm.

A main advantage of the combined algorithm compared to simple feature-tracking, is the user defined positioning of the drift vectors. The current algorithm setup allows the user to choose whether the drift vectors should be positioned at certain points of interest or on a regular grid with adjustable spacing. Constricting the pattern-matching process to the area of interest minimises the computational effort according to the individual needs.

The recommended parameters shown in Table 1 are not meant as a fixed setting, but should rather give a suggestion and guideline to estimate the expected results and the corresponding computational effort. The parameters can easily be varied in the algorithm setup and should be chosen according to

availability of computational power, needed resolution, area of interest and expected ice conditions (e.g. strong rotation).

The presented combination of feature-tracking and pattern-matching can be applied to any other application that aims to derive displacement vectors computationally efficient from two consecutive images. The only restriction is that images need to depict edges, that can be recognised as keypoints for the feature-tracking algorithm, and the conversion into intensity values $i$ (Equation 2) needs to be adjusted according to the image type.

The remote sensing group at NERSC is currently developing a new pre-processing step to remove thermal noise on HV images over ocean and sea ice. First tests have shown a significant improvement of the sea ice drift results using this pre-processing step before applying the presented algorithm. This is ongoing work and will be included into a future version of the algorithm.

The European Space Agency is also in the process of improving their thermal noise removal for Sentinel-1 imagery. Noise removal in range direction is driven by a function that takes measured noise power into account. Until now, noise measurements are done at the start of each data acquisition, i.e. every 10-20 minutes, and a linear interpolation is performed to provide noise values every 3 seconds. The distribution of noise measurements showed a bimodal shape and it was recently discovered that lower values are related to noise over ocean while higher values are related to noise over land. This means, that Sentinel-1 is able to sense the difference of the earth surface brightness temperature similar to a passive radiometer. When the data acquisition includes a transition from ocean to land or vice versa, the linear interpolation fails to track the noise variation. The successors of Sentinel-1A/B are planned to include more frequent noise measurements. Until then, ESA wants to use the 8-10 echoes after the burst that are recorded while the transmitted pulse is still travelling and the instrument is measuring the noise. This will provide noise measurements every 0.9 seconds and allows to track the noise variations in more detail. In addition, ESA is planning to introduce a change in the data format during 2017 that shall remove the noise shaping in azimuth. These efforts are expected to improve the performance of the presented algorithm significantly (Personal Communication with Nuno Miranda, January 2017).

Having a computationally efficient algorithm with adjustable vector positioning allows not only to provide near-real time operational drift data, but also the investigation of sea ice drift over large areas and long time periods. Our next step is to embed the algorithm into a super-computing facility to further test the performance in different regions, time periods and ice conditions and evaluate and combine the results of different polarisation modes. The goal is to deliver large ice drift datasets and open-source operational sea ice drift products with a spatial resolution of less than 5 km.

This work is linked to the question how to combine the different timings of the individual image pairs in a most useful way. Having more frequent satellite acquisitions, as we get with the Sentinel-1 satellite constellation, enables to derive displacements for shorter time gaps and the calculated vectors will reveal more details e.g. rotational motion due to tides. As part of a scientific cruise with

KV-Svalbard in July 2016, we deployed three GPS trackers on loose ice floes and pack-ice in Fram

Strait. The trackers send their position every 5-30 min to deliver drift information with high temporal resolution. This efforts shall help to gain a better understanding of short-term drift variability and by comparison with calculated sea ice drift, we will investigate how displacement vectors from subsequent satellite images relate to sea ice displacements with higher temporal resolution.

The focus of this paper in terms of polarisation was put on the HV channel, since this polarisation

provides on average four times more feature-tracking vectors (using our feature-tracking approach) than HH and therefore delivers a finer initial drift for the first guess. We found our area of interest well covered with HV images, but other areas in the Arctic and Antarctic are currently only monitored in HH polarisation. Considering the four representative feature-tracking image pairs from Muckenhuber et al. (2016), the relatively best HH polarisation performance (i.e. most vectors from

HH, while at the same time fewest vectors from HV) was provided by the image pair that had the least time difference, i.e. 8 h, compared to 31 h, 33 h and 48 h. Therefore, we assume that the HV polarisation provides more corner features that are better preserved over time. And more consistent features could potentially also favour the performance of the pattern-matching step, but this is only an assumption and has not been tested yet. Another argument is that the presented feature-tracking

approach identifies and matches corners, which represent linear features. The linear features on HH images are more sensitive to changes in incidence angle, orbit and ice conditions than the linear features on HV images. This could explain the better feature-tracking performance of the HV channel. However, pattern-matching is less affected by changing linear features and more sensitive to areal pattern changes. This could potentially mean that the HH channel performs better than HV when it

comes to pattern-matching. However, at this point, these are just assumptions and will be addressed in more detail in our future work.

Utilising the advantage of dual polarisation (HH+HV) is certainly possible with the presented algorithm, but increases the computational effort. A simple approach is to combine the feature tracking vectors derived from HH and HV and produce a combined first-guess. Pattern-matching can be

performed based on this combined first-guess for both HH and HV individually and the results can be compared and eventually merged into a single drift product. Having two drift estimates for the same position, from HH and HV pattern-matching respectively, would also allow to disregard vectors that disagree significantly. However, this option would increase the computational effort by two, meaning that the presented Fram Strait example would need about 8 min processing time.

After implementing the presented algorithm into a super-computing facility, we aim to test and compare the respective performance of HV, HH and HH+HV on large datasets to identify the respective advantages.

The current setting of the feature-tracking algorithm applies a maximum drift filter of 0.5 m/s. We found this to be a reasonable value for our time period and area of interest. However, when consider-

ing extreme drift situations in Fram Strait and a short time interval between image acquisitions, this threshold should be adjusted.

As mentioned above, we deployed three GPS tracker in Fram Strait and they recorded their positions with a temporal resolution of 5-30 min between $8^{th}$ July until $9^{th}$ September 2016 in an area covering 75° N to 80° N and 4° W to 14° W. Considering the displacements with 30 min interval, we

found velocities above 0.5 m/s on a few occasions, when the tidal motion adds to an exceptionally fast ice drift.

The GPS data from the hovercraft expedition FRAM2014-2015 (https://sabvabaa.nersc.no), that was collected with a temporal resolution of 10 s between $31^{st}$ August 2014 until $6^{th}$ July 2015, did not reveal a single 30 min interval during which the hovercraft was moved by ice drift more than

0.45 m/s. The hovercraft expedition started at 280 km south from the North Pole towards the Siberian coast, crossed the Arctic Ocean towards Greenland and was picked up in the north-western part of Fram Strait.

In case the estimated drift from feature-tracking reaches velocities close to 0.5 m/s, the pattern-matching step might add an additional degree of freedom of up to 8 km, which could eventually

lead to a higher drift result than 0.5 m/s, depending on the time interval between the acquisitions. The smaller the time difference, the larger is the potentially added velocity. In order to be consistent when combining the drift information from several image pairs with different timings, one should apply a maximum drift filter on the final drift product of the presented algorithm that has the same maximum velocity as the feature-tracking filter. The corresponding function is implemented in the

distributed open-source algorithm. As an alternative, one could adjust the search window according to the time span. However, this would add additional complexity to both the algorithm and the parameter evaluation and needs more research on how the search window should be adjusted depending on the time span. For the sake of computational efficiency, we suggest the simple approach to remove final drift vectors above the maximum speed.

**Appendix A:  Open-source distribution**

The presented sea ice drift retrieval method is based on open-source satellite data and software to ensure free application and easy distribution. Sentinel-1 SAR images are distributed by ESA for free within a few hours of acquisition under https://scihub.esa.int/dhus/. The algorithm is programmed in Python (source code: https://www.python.org) and makes use of the open-source li-

braries Nansat, openCV and SciPy. Nansat is a Python toolbox for processing 2-D satellite Earth observation data (source code: https://github.com/nansencenter/nansat). OpenCV (Open Source Computer Vision) is a computer vision and machine learning software library and can be downloaded under http://opencv.org. SciPy (source code: https://www.scipy.org) is a Python-based ecosystem of

software for mathematics, science, and engineering. The presented sea ice drift algorithm, including
an application example, is distributed as open-source software as supplement to this manuscript.

*Author contributions.* Stefan Muckenhuber designed the algorithm and the experiments, performed the data analysis and interpretation of the results and wrote the manuscript. Stein Sandven critically revised the work and gave important feedback for improvement. Stefan Muckenhuber and Stein Sandven approved the final version for publication.

*Acknowledgements.* This research was supported by the Norwegian Research Council project IceMotion (High resolution sea-ice motion from Synthetic Aperture Radar using pattern tracking and Doppler shift, project number 239998/F50). We thank Polona Itkin and Gunnar Spreen for providing us the buoy GPS data that were collected as part of the N-ICE2015 project with support by the Norwegian Polar Institute's Centre for Ice, Climate and Ecosystems (ICE) and its partner institutes. The used satellite data were provided by the European
Space Agency. We thank Nuno Miranda for information on ESA's de-noising efforts for Sentinel-1.

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
