# Peer review of "Open-source sea ice drift algorithm for Sentinel-1 SAR imagery using a combination of feature-tracking and pattern-matching"

_The Cryosphere, 2016_

## Referee Comment (RC1) · Anonymous Referee #1 · 20 Dec 2016

General comments The authors present a new approach for sea ice motion tracking, combining a modified feature tracking algorithm (Muckenhuber, 2016) with a basic pattern matching approach using cross correlation. The authors thereby replace the often used iterative cross correlation approach within an image resolution pyramid by a feature tracking step (which involves a resolution pyramid as well) to predict the search direction for the higher resolution levels of the cross correlation step. A. Berg and L. E. B. Eriksson (2014) presented with their paper on "Investigation of a Hybrid Algorithm for Sea Ice Drift Measurements Using Synthetic Aperture Radar Images," based on the combination of pattern matching (cross and/or phase correlation) and feature tracking.

In 2014 Komarov and Barber published an algorithm (also referred in this paper), which uses a kind of correlation based feature tracking – since it first identifies characteristic points for the following correlation. The idea to combine feature tracking and pattern matching for sea ice drift estimation is tempting and I really like it. It would potentially allow estimating sea ice motion faster and in the case of appropriate feature descriptors even that are rotationally invariant for areas which contain not only translational motion but rotational motion as well. This characteristic can be especially useful in regions like the marginal ice zone, where rotational motion occurs relatively often. However, the devil is in the details. The idea of study is first step in the direction of a rotational invariant drift algorithm (or at least more robust against rotational motion) for the marginal ice zone and would therefore be worth being published in the Cryosphere after major revisions. However, due to some open questions regarding the implementation of the approach and its validation I cannot recommend its publication at this point. I would like to encourage the authors to continue the work on this interesting idea and resubmit a strongly revised version of this work in the future, but being a bit more careful next time.

My main concerns are: 1. the suggested logarithmic scaling and its surprising limits (I guess there is something wrong with the calibration routines, ) 2. The very vague description of the combination of feature tracking and pattern matching 3. And the slightly irritating validation approach

I would be happy if I was of any help for your review process and wish you Season's greetings and best wishes for the New Year!

Specific comments

Introduction Page 3 Line 62-63 "the resulting vectors are independent of their neighbours [which] is an important advantage ..." – I'm afraid I have to disagree at that point, especially given the implemented feature tracking algorithm. – It has the advantage that it is fast, that it does not get confused by rotational motion and is able

to estimate the translational motion even in regions with occurring rotational motion (and that is already great!) but since the employed feature tracking uses a resolution pyramid as well and simply combines all vectors from the different levels of the resolution pyramid, the resulting vectors are neither necessarily all independent nor have the same accuracy (given that some of them are based on a coarser version of the image). Regarding shear and deformation zones, I would claim that a pattern matching algorithm could do the same with an optimised search strategy. Even more problematic, the suggested feature tracking algorithm only identifies a given number of features for the whole scene. In the worst case, a shear zone or a divergence / convergence zone would not be covered at all, if other features in the scene have a higher score.

Page 3 Line 69 "comparable quality estimate for each vector" – I wish there were! There has been a first suggestion by Hollands, Linow and Dierking in 2015 and there is definitely the potential to do so but it is far from being a standard.

Data Page 3 Line 92 "this data type" - the dual pol version of this data type is only available for the southern part of the Arctic and the Coastal regions and not at all for Antarctica. Since their feature tracking algorithm prefers HV polarisation I wonder if the authors have analysed the results of their algorithm in the case of HH polarisation only to predict a potential performance for the otherwise omitted regions.

Method

Page 4 Line 118 "good geolocation accuracy" – I believe I remembered some discussions, that there were some geolocation problems with Nansat earlier, which effected the drift estimation. If I remember correctly: is there a chance that the authors could quantify what "good" means in this respect?

Page 5 Lines 126 – 135 For a start I would suggest to change the order of the explanation and first mention the conversion from linear to log scale before the authors mention the scaling to integer values between 0 – 255 but this is the easier part. The more difficult part might be that we have a problem if there are no typos in these lines

and I understood everything correctly. Log(0.013) = -1.88 dB while log(0.08) = -1.1 dB. If their minimum backscatter values are in dB as well (units missing!), it would mean, that the authors only use the range between -3.25dB – -1.88dB for HV and the range between -2.5dB – 1.1 dB. Could the authors please comment on this and even rephrase this part if I just misunderstood the authors? The problem I see is that their chosen backscatter range only represents a minor part of the backscatter range to be expected for sea ice in the logarithmic scale. If these are the correct numbers, the authors might as well want to check the calibration routines for their data.

Page 6 Line 166 "serves as a quality estimate of the matching performance" - After it has been shown by Hollands, Linow and Dierking (2015) that there is no relation between the matching error and the correlation coefficient I would prefer a proof why the authors can use it as a quality measure. Even their Fig. 7 shows that the authors also dismiss good values, using the correlation coefficient as a quality value. Admittedly there is a group of large error values in their histogram but I wonder if this is significant. A correlation coefficient is only meaningful if the respective texture is characteristic enough. - I suggest to google Anscombe's quartet. Combination

Page 6 Line 173 – 176 "To filter outliers, . . . removed" – I have to admit, it would help me, if the authors could describe this outlier removal in more detail – based on the current description it is difficult to evaluate what the authors actually did.

Page 6 Line 177 – 181 "The remaining feature vectors . . . neighbouring feature tracking vectors" – Just for the better understanding: What happens if there is a large area with no vectors at all framed by a few sparse vectors. Would the authors just triangulate over the whole area (potentially containing deformation or shear zones)?

Page 6 Line 181-183 "To provide a drift estimate . . . combination of x1 and y1." – similar to Line 173 -176 it is hard to say, what the authors actually did. May be the authors could add some details, making it easier to follow.

Page 6 Line 187- 190 I find it a bit confusing that we have a given size of the window

before it is tuned. The same is true for dmin and dmax: It only became clear when I reached section 3.3. I would suggest that the authors mention here that they are going to identify the optimal parameters and may be as well why the authors decided to choose formula (4) for the window size.

Page 7 Figure 1 It would be interesting so see a SAR image for the same area and may be a drift vector field. Is it correct that there is land where the distances are low and sea ice where the distance colour scale is saturated? Actually the authors already anticipate a result of their parameter tuning here. That makes it difficult to read. May be the authors should reorganise this part.

Page 7 Line 195 "–beta +beta with step delta beta" – it is confusing that the authors suddenly start to introduce rotation as well since it has not been mentioned beforehand. The authors should have at least introduced it in section 3.2 II.

Section 3.2 page 5-7 Given that this section is meant to be the innovative part of this study I suggest restructuring it, to make it more concise. Right now, it is quite confusing and has varying level of detail and order (e.g. the window size question is a specific cross correlation question. I would urge the authors to state clearly when they introduce a parameter which they want to tune in the later course of the paper. Additionally I would suggest adding a flow chart, highlighting the steps, described in this paper.

Page 8 Formula 6 Why did the authors choose this distance measure instead of the RMSD in Formula 5?

Section 4.1 Honestly, I would suggest skipping this section – it is not surprising that the logarithmic scaling leads to a higher number of features since the logarithmic histogram scaling favours the structures in the sea ice which are mainly represented in the shadow and medium backscatter values but hardly in the highlights.

Page 10 Section 4.2 / Table 2 I have various questions: ćI understood that the authors tuned their Influence domain parameter dmax based on one image pair over

Fram strait as well as the side length for their template but how did the authors tune their Dmin value and the MCCmin value? ć70 x 70 pixel for t1 means that their correlation window covers an area of approx. 6.3 x 6.3 km – how does this go along with their claim to resolve deformation and shear zones? ćSince their influence domain influences the size of their search window t2 it would mean that the authors add a degree of freedom of +/-1.8 to +/-11.25 km to their first feature tracking based guess, which would push their 0.5 m/s maximum ice drift limit for the feature tracking to about 0.6 m/s – right? Its contribution would however vary depending on the time span between both images of the scene. For the same constant drift velocity (but speed variations with in the scene), an image pair with a longer time span would then show larger displacement differences with in the scene while having the same maximum degree of freedom of +/- 11.25 km like an image pair that has been acquired at the same day – this might cause a problem, don't the authors think?

Page 10 Section 4.3 line 249: "on a grid with 8 km spacing" – I suggest to summarize the information of their resulting product somewhere. It is not necessarily obvious to find the information on their grid spacing in the Parameter tuning and Computational Efficiency Section.

Page 10 Section 4.3 Given the resolution of 8 x 8 km even pattern matching only based algorithms show a similar performance or even better. But I admit that the robustness to rotational motion is very useful in the marginal ice zone, where many of the pure pattern matching algorithms fail.

Page 10 Section 4.4 line 261: What does the size of 34 pixel mean? Is the feature described as a patch of 34 x 34 side length? May be the authors should add a short explanation to their feature tracking part on page 5.

Page 12 Line 268-271: Why do the authors choose a minimum Cross Correlation Coefficient of 0.35? If the authors found a logarithmic function their distance distribution seems to follow, the authors could name it. Otherwise less strict term would be that the

distance distribution seems to show a logarithmic behaviour or something like this. A peak at 300m is not necessarily meaningful (e.g. what would be the peak without their Cross Correlation Threshold? How many drift vectors form a peak?) but even if the authors have a peak, it does only represent the systematic component of the error and not the random one. In order to identify the distribution I would suggest smoothing the histogram and fitting a distribution to it.

Page 14 Table 4: I would think that it is not the best approach to validate an algorithm based on the drift vectors I tuned it to. For a real validation the authors need at least another independent image pair with an independent set of manually derived drift vectors. I would strongly encourage the authors to change this! The authors compare apple with oranges if the authors compare an algorithm tuned to this specific scene with algorithms like the one from CMEMS. Additionally it would be great, if the authors could quantify both systematic and random error.

Page 14 Line 290: “To further estimate the accuracy of the algorithm . . .” – here it would be interesting to see, how the other algorithms perform as well. Additionally it would be great, if the authors could quantify both systematic and random error. The authors might want to check the regular validation document for the CMEMS ice drift as a start: http://myocean.met.no/SIW-TAC/doc/myo-wp14-siw-dtu-icedrift-glob-obs-validation_latest.pdf The peak of a distribution is no error value!

Page 14 Line 302-303: “Hence, . . . image resolution” I agree there are various factors influencing the result of the algorithm and thereby influencing the validation but I cannot agree with this statement. It might be but the authors have not shown this yet!

Technical corrections

Page 1 Line 5: “respective advantages of the two approaches” - the authors should emphasise in more detail what the advantages are, since this is the basic justification for this paper and this not only in the abstract but in the introduction/motivation as well

Page 3 Line 37 “covers the Arctic every week with a spatial resolution of 5 km” – I’m not sure but the authors might want to check it: as far as I know the, RGPS covers a large part of the Western Arctic Ocean but not the entire Arctic, due to the acquisition area of Radarsat. Up to my knowledge, the 5 x 5 km spatial resolution is a gridded drift field, which does not necessarily represent the actual spatial resolution, given that the RGPS searches features in a 10 or 25 km grid respectively. See also the RGPS Data User’s Handbook (Fig. 1 and Fig. 2)

Page 3 Line 73 “respective advantages” – If possible, be clearer about the respective advantages and summarise them here together with the disadvantages the authors still have and those the authors bypass with their approach.

Page 2 Line 44 “pattern-marching and feature tracking respectively” – even terms are somehow flexible: I would claim, that Komarov and Barber do somehow a basic feature tracking as well, since they identify features, with certain characteristics before the correlate them – in that way, they have implemented the search for descriptors in a way. The use of correlation does not necessary mean that the approach is a pattern matching approach, since the correlation itself is the distance measure only, that is used to assess how similar a feature or a pattern is, compared to the reference. It might be a bit pedantic, but the authors might still want to give it a second thought.

Page 2 Line 52-55 “Making use . . . Copernicus.eu).” – I agree, that it is an important product, which should definitely be mentioned in the frame of this article but I think, the statement does not really fit there where it is right now because it interrupts their motivation.

Page 6 Line 185 -186 “Figure 2 shows. . .” – I would suggest moving the sentence a few sentence down to Line 195 after “. . .correlation value is returned

Page 10 Section 4.3 line 252-254: “NB: The vectors near . . . treated with caution” - I completely agree but it is no question of computational efficiency

Page 10 Section 4.4 line 256: Strictly speaking the authors should compare their estimated drift vectors to their manually derived vectors and not the other way round and the authors estimate a drift vector and do not calculate it but this is a minor technical issue I guess.

Page 11 Table 3: Is it correct, that their drift estimation is only based on HV polarisation? I guess the authors should state it somewhere in the beginning. Given their experience with dual pol motion tracking, I assumed that the authors used both polarisations here as well? I suggest being clearer about it from the beginning, if this is the case.

Page 15 Line 311: "The parameters can easily be varied. . ." – a short tabular overview on the range for the individual parameters and their effect on the algorithm performance would be nice even though probably difficult.

Page 15 Line 329: "the real sea ice velocity" – the velocity the authors observe is not wrong, they might underestimate the speed and its variation as well as the variation of the drift direction but velocity is defined as distance per time, and the resulting velocity vector, being a sum of velocity vector variations over the observation interval is the resulting velocity vector. A higher temporal resolution is interesting but it is as interesting and influences the "realness" of their velocity vector the same way higher spatial resolution does. It would be great if the authors could give this phrase a second thought.

Please also note the supplement to this comment:
http://www.the-cryosphere-discuss.net/tc-2016-261/tc-2016-261-RC1-supplement.pdf

---

## Referee Comment (RC2) · Anonymous Referee #2 · 30 Dec 2016

Dear members of the TC editorial board and authors of the manuscript tc-2016-261,

The topic manuscript in interesting and useful for the sea ice community. Pattern matching (normalized cc or phase correlation have been standard methods for operational ice drift monitoring for a few decades already) and feature-based approaches seem to be promising approach for fast ice drift monitoring.

Before publishing the manuscript needs to be updated and clarified at some points. Here are my comments:

Major comments:

[Figure]

Introduction, P2: The manuscript should include some additional background information on the sea ice drift in the study area (with possible references): what are the magnitudes of typical ice drift in the study area and whole Arctic (e.g. cm/s and daily) and in which areas they are located and which are their causes?

Method/Feature tracking, P5: It is mentioned that "The best match is accepted if the ratio of the two shortest Hammin Distances is below 0.75.". Explain why this is done and how the threshold was selected. Probably to reduce possibilty of similarization errors? What is magnitude of typical Hamming distances? If they are small, then 0.75 has quite different meaning than for larger values. I assume that the ratio is the ratio of the shortest and second shortest Hamming distance (also write this in the text).

Method/Combination, P6: To filter outliers each vector is simulated using two functions which are LS solutions... This need more explanation. Why third degree polynomial has been used and which data are used in the LS fit? Also in the extrapolation is also performed using a LS solutions. Also describe this in more detail. How is the traingulation constructed (Delauney?) in interpolation?

Parameter Tuning/Validation, P10: It is not exlpicitly mentioned which data were used for the parameter tuning. Were all the vlaidation data used for this? Then the validation with this data set is not fair as the algorithm has been tuned for this data. Then only the buoy data can be used for independent validation. Or if separate sets are used for parameter tuning and validation, indicate this in the manuscript.

Detailed comments:

P1L2: "computanional" -> "computationally"

P2L33: "90s" -> "90's"

P2L38: In the case of ENVISAT, rather give the name of the instrument i.e. ENVISAT ASAR, could also mention that RADARSAT was an instrument of CSA and ENVISAT ASAR of ESA.

P3L88: "((" -> "("

P3L88: "...dual polarization support..." "..also in wide swath mode". Also earlier instruments had a possibility to measure multiple polarizations but the covered area was small. This has been changed by RADARSAT-2 and SENTINEL-1.

P3L90: Give also the acronyms for the mode i.e. EW GRDM (thes are generally used by ESA in documentation and file names).

P4L125: You can remove "of 93m range x 87m azimuth", this information has already been given earlier.

P6 eq. 3: Here You give the formula for NCC. Also give the drift (dx,dy) detection as a formula, something like: (dx,dy)=argmax_(k,l)in W NCC(x+k,y+l) Is NCC computed according to this equation or by applying FFT and IFFT (which has been applied in many algorithms to fasten the computation)?

P6 Eq. 4: Define "side" in the text.

P7 Fig 1 and Fig 2. Use a, b, and c for the subfigures and to refer to them.

P7 L195: Explain here what is denoted by "beta". Itis is also in Fig. 2 caption.

P7 Fig.2 (and text): Why rectangular/square templates has been used? A circular template would be much easier (symmetric) to rotate. Consider using a circular templates instead.

Logarithmic scaling P8-9: I think logarithmic scale is the typical presentation of SAR sigma0 and often a fixed scaling to gray tone imagery is used for SAR imagery, e.g. scaling between -30dB -> 0 dB. You could mention this fact on the manuscript. This also leads to the question if any other "scaling" would produce even better results, e.g. applying some king of histogram derived image mapping (e.g. simple histogram equlization etc.). This could be one topic for further development.

P10/Computaional efficency: You give a time of less than 3.5 minutes here. Is this a

typical execution time or just execution time for a randomly selected example. Could you give average execution times and deviations or maybe estimate for the worst case? Does the execution time increase linearly as a function of the number of vectors or is there some other kibd of relationship?

P12 L270-271: also give the average D. "peak" is not a correct word here, the histogram/distribution has many peaks, possibly You could use "mode" here and also in the caption of Fig. 7.

P12 L276-277: The DTU method has not been documented very well in any publications I think. Also the reference given does not say much. I suppose there is not better reference for this?

P12 L279: "...used the nearest neighbors..." -> "...used the nearest neighbors (NN's)..." then NN can be used in Table 4.

P13 Fig. 6: Would it be possible to indicate the location of the detail in the coarse-scale image (without causing too much damage for the image)?

P14 Table 4: "Average distance" -> "Average NN distance" or something like that. Are the values after +- sign standard deviations or some multiples of tandard deviation or something else? Include this infomration in the table or caption.

Discussion: What is the possible error magnitude of the manually estimated drift (is it assumed to be sub-pixel, one pixel or more and what kind of possible error sources these vectors include?)?

P15 L319-320: Also ESA is going to improve their thermal noise removal by including more measurements along the azimuth direction. Rpbably this also could be mentioned. If necessary You can get more information on this from Nuno Miranda at ESA (nuno.miranda@esa.int).

Sincerely,

---

## Referee Comment (RC3) · Anonymous Referee #3 · 2 Jan 2017

Season's greetings to the editorial team, authors, and fellow reviewers.

The topic of "Open-source sea ice drift algorithm for Sentinel-1 SAR imagery using a combination of feature-tracking and pattern-matching" by Muckenhuber and Sandven (doi:10.5194/tc-2016-261) is obviously relevant for inclusion in The Cryosphere, and interesting to several of its readers.

This new manuscript extends upon the research presented in Muckenhuber et al. (2016), and introduces some new developments on a sea ice drift algorithm from Sentinel-1 SAR imagery for the "European" Arctic. Specifically, the authors present

a pattern-matching step that is applied after the feature-matching step of Muckenhuber et al. (2016). Tuning of algorithm parameters, as well as a validation against a pool of ground-truth estimates is discussed (such a validation was missing from their previous paper).

A general impression after reviewing this manuscript is that it requires more work and provision of additional details before being ready for publication in TC. The authors are thus invited to revise their manuscript before a new version is submitted.

Specifically, the following items should be addressed:

1) Description of the algorithms

The "pattern-matching" step is not well enough described and many questions are still open at the end of section 3.2.

1.a) The ordering of the sub-sections (I. Feature-Tracking, II. Pattern-matching, III. Combination) is maybe not optimal as you spend some of Section III to describe the rotation by angle beta (that should really go into II). Maybe it would be easier to follow if the sub-section followed the steps of the algorithms (feature-matching, fitting of polynomial for first-guess, filtering, patter-matching, etc...).

1.b) It is unclear if your pattern-matching step features a series of x,y shifts to maximize the cross-correlation in addition to the rotation by beta, or not. If you combine both x, y, and beta shifts, what is the relative order and does it matter?

1.c) As you recall in I. "Feature-Tracking", the ORB algorithms also gives an information about the rotation angle (delta between centroid-based orientation of the matched features). Is this feature-matching first-guess of the rotation used at all? If yes, how; and if no, why not?

1.d) What is "the initial rotation between the two Sentinel-1 image" (line 194) and how is it computed? Is it the same value across the image?

1.e) In subsection II. “Pattern-matching” you write the NCC formula for “two equally sized windows”. But later you seem to use two unequally sized windows (size t1 in SAR1, size t2 in SAR2). What is the NCC formula do you then use? Of is size t2 related to the size of the search window while t1 is the size of the pattern? The questions above are mostly to give an impression of the level of details expected when you re-formulate this section. Your first manuscript contained quite some details on the methodology, and this new one requires at least as many details.

2) Validation against GPS data.

2.a) The choice of validation metric (the distance between the end points of the reference and estimated vectors) is not peculiar. Virtually all other studies use the RMSE along two components (e.g. u and v). And the logarithmic distribution of the errors is not discussed or exploited. Please also discuss the RMSDs in u and v components and compare your results with that of other investigators.

2.b) The N-ICE campaign deployed many buoys, but very much in the vicinity of the vessel Lance. How many different buoys enter your validation database, and what is the average distance between them? Are we sampling more than few kilometres in each SAR pair?

2.c) N-ICE data should offer the possibility to discuss the accuracy when inside the pack versus at the marginal ice zone. Please see if you can segment your validation database to cover this. As you point out yourself, the added value of rotation should be most visible in the marginal ice zone.

2.d) Can you convince the reader (and the reviewer!) that the value of the maximum NCC indeed constitutes a quality measure (your Abstract)? Are matchups with lower NCC values really father away from GPS truth, than those with high NCC? Hollands et al. (2015) did not find any relation between the two. Is your threshold at 0.35 related to a significant drop in the documented accuracy against the buoy drift? (Hollands, T. , Linow, S. and Dierking, W. (2015): Reliability Measures for Sea Ice Motion Retrieval From Synthetic Aperture Radar Images , IEEE Journal of Selected Topics in Applied Earth Observations and Remote Sensing, 8 (1), pp. 67-75 . doi: 10.1109/JSTARS.2014.2340572)

2.e) You use a maximum velocity of 0.5 m/s for your feature-based results (line 171). Is this limit high-enough in view of your validation dataset in the Fram Strait region?

Finally, it would be good if the revision of the paper could include a thorough discussion of the robustness of the combined method to the success of the feature-matching step (not in terms of computation cost, but of introduction of artefacts).

---

## Author Comment (AC1) · 13 Feb 2017

Manuscript prepared for J. Name
with version 2015/09/17 7.94 Copernicus papers of the LaTeX class copernicus.cls.
Date: 13 February 2017

**Response to Referee # 1**

**'Open-source sea ice drift algorithm for Sentinel-1 SAR imagery using a combination of feature-tracking and pattern-matching'**

Stefan Muckenhuber and Stein Sandven

Nansen Environmental and Remote Sensing Center (NERSC), Thormøhlensgate 47, 5006 Bergen, Norway

*Correspondence to:* S. Muckenhuber (stefan.muckenhuber@nersc.no)

Dear Referee # 1,

Thank you very much for helping us improving our paper.

Please find here the **answers** to your comments and the corresponding ***changes in manuscript***:

**1   General comments**

The authors present a new approach for sea ice motion tracking, combining a modified feature tracking algorithm (Muckenhuber, 2016) with a basic pattern matching approach using cross correlation. The authors thereby replace the often used iterative cross correlation approach within an image resolution pyramid by a feature tracking step (which involves a resolution pyramid as well) to predict the search direction for the higher resolution levels of the cross correlation step. A. Berg and L. E. B. Eriksson (2014) presented with their paper on 'Investigation of a Hybrid Algorithm for Sea Ice Drift Measurements Using Synthetic Aperture Radar Images,' based on the combination of pattern matching (cross and/or phase correlation) and feature tracking. In 2014 Komarov and Barber published an algorithm (also referred in this paper), which uses a kind of correlation based feature tracking - since it first identifies characteristic points for the following correlation.

**The work from Berg and Eriksson needs to be mentioned and we included**

***Berg and Eriksson (2014) introduced a hybrid algorithm for sea ice drift retrieval from ENVISAT ASAR data using phase correlation and a feature based matching procedure that is activated if the phase correlation value is below a certain threshold.***

***Unlike Berg and Eriksson (2014), the feature-tracking step is performed initially and serves as a first guess to limit the computational effort of the pattern-matching step.***

**To specify the approach from Komarov and Barber 2014, we changed 'pattern-matching' to**

*combination of phase/cross-correlation.*

The idea to combine feature tracking and pattern matching for sea ice drift estimation is tempting and I really like it. It would potentially allow estimating sea ice motion faster and in the case of appropriate feature descriptors even that are rotationally invariant for areas which contain not only translational motion but rotational motion as well. This characteristic can be especially useful in regions like the marginal ice zone, where rotational motion occurs relatively often. However, the devil is in the details.

The idea of study is first step in the direction of a rotational invariant drift algorithm (or at least more robust against rotational motion) for the marginal ice zone and would therefore be worth being published in the Cryosphere after major revisions. However, due to some open questions regarding the implementation of the approach and its validation I cannot recommend its publication at this point. I would like to encourage the authors to continue the work on this interesting idea and resubmit a strongly revised version of this work in the future.

My main concerns are:

1. the suggested logarithmic scaling and its surprising limits (I guess there is something wrong with the calibration routines, )

2. The very vague description of the combination of feature tracking and pattern matching

3. And the slightly irritating validation approach

**There is a typo in previous Section 3: the values of the brightness boundaries were given in B and not dB. We corrected that and included a histogram of a representative image pair to illustrate the chosen boundaries. The description of the algorithm has been changed and**

**extended. The validation approach has been changed and the error analysis has been extended. Details are given below.**

**2   Specific comments**

Page 3 Line 62-63 'the resulting vectors are independent of their neighbours [which] is an important advantage ...' - I'm afraid I have to disagree at that point, especially given the implemented feature tracking algorithm. - It has the advantage that it is fast, that it does not get confused by rotational motion and is able to estimate the translational motion even in regions with occurring rotational motion (and that is already great!) but since the employed feature tracking uses a resolution pyramid as well and simply combines all vectors from the different levels of the resolution pyramid, the resulting vectors are neither necessarily all independent nor have the same accuracy (given that some of them are based on a coarser version of the image). Regarding shear and deformation zones, I would claim that a pattern matching algorithm could do the same with an optimised search strategy. Even more problematic, the suggested feature tracking algorithm only identifies a given number of features for the whole scene. In the worst case, a shear zone or a divergence / convergence zone would not be covered at all, if other features in the scene have a higher score.

**With the term 'independent', we wanted to refer to the fact that features are identified without taking the position of other features into account and matched from one image to the other without taking the drift and rotation information from surrounding vectors into account. It is true that features can overlap, the resolution varies due to the resolution pyramid and the independent feature positioning can lead to missing important drift information. We**
**changed the sentences to:**

*This can be done computationally efficient and the resulting vectors are often independent of their neighbours in terms of position, lengths, direction and rotation, which is an important advantage for resolving shear zones, rotation and divergence/convergence zones. The considered feature-tracking approach identifies features without taking the position of other features into*
*account and matches features from one image to the other without taking the drift and rotation information from surrounding vectors into account (Muckenhuber et al., 2016). However, due to the independent positioning of the features, very close features may share some pixels and since all vectors from the resolution pyramid are combined, the feature size varies among the matches, which implies a varying resolution. In addition, the resulting vector field is not evenly distributed*
*in space and large gaps may occur between densely covered areas, which can eventually lead to missing a shear or divergence/convergence zone.*

Page 3 Line 69 'comparable quality estimate for each vector' - I wish there were! There has been a first suggestion by Hollands, Linow and Dierking in 2015 and there is definitely the potential to do
so but it is far from being a standard.

**We agree and removed this part of the sentence.**

Page 3 Line 92 'this data type' - the dual pol version of this data type is only available for the southern part of the Arctic and the Coastal regions and not at all for Antarctica. Since their feature
tracking algorithm prefers HV polarisation I wonder if the authors have analysed the results of their algorithm in the case of HH polarisation only to predict a potential performance for the otherwise omitted regions.

**The focus of this paper is put on HV, since this polarisation has a better feature-tracking performance and we found a good coverage of this data type in our region of interest. We did**
**not yet analyse the HH performance of the algorithm on a large dataset, but this will certainly be addressed in our future work. We added the following to Section 2:**

*The introduced algorithm can utilise both HH and HV channel. However, the focus of this paper is put on using HV polarisation, since this channel provides on average four times more feature tracking vectors than HH Muckenhuber et al. (2016), representing a better initial drift estimate*

*for the combined algorithm.*

**To further address the polarisation topic, we added the following to Section 5:**

*The focus of this paper in terms of polarisation was put on the HV channel, since this polarisation provides on average four times more feature tracking vectors than HH and therefore delivers a finer initial drift for the first guess. We found our area of interest well covered with HV*

*images, but other areas in the Arctic and Antarctic reveal a better coverage in HH polarisation. Considering the four representative feature-tracking image pairs from Muckenhuber et al. (2016), the the relatively best HH polarisation performance (i.e. most vectors from HH, while at the same time fewest vectors from HV) was the image pair that showed the least time difference, i.e. 8 h, compared to 31 h, 33 h and 48 h. Therefore, we assume that the HV polarisation provides*

*more features that are better preserved over time. And more consistent features would also favour the performance of the pattern-matching step. However, at this point, this is just an assumption and will be addressed in more detail in our future work.*

*Utilising the advantage of dual polarisation (HH+HV) is certainly possible with the presented algorithm, but increases the computational effort. A simple approach is to combine the feature*

*tracking vectors derived from HH and HV and produce a combined first-guess. Pattern-matching can be performed based on this combined first-guess for both HH and HV individually and the results can be compared and eventually merged into a single drift product. Having two drift estimates for the same position, from HH and HV pattern-matching respectively, would also allow to disregard vectors that disagree significantly. However, this option would increase the*

*computational effort by two, meaning that the presented Fram Strait example would need about 8 min processing time.*

*After implementing the presented algorithm into a super-computing facility, we aim to test and compare the respective performance of HV, HH and HH+HV on large datasets to identify the respective advantages.*

Page 4 Line 118 'good geolocation accuracy' - I believe I remembered some discussions, that there were some geolocation problems with Nansat earlier, which effected the drift estimation. If I remember correctly: is there a chance that the authors could quantify what 'good' means in this respect?

**We discovered drift artefacts in high latitudes between the ground control points before we introduced spline interpolation and reprojection to stereographic. We tested the performance before and after introducing these steps and the artefact disappeared using either one of the steps. To ensure the best possible performance, we apply both steps. The geolocation accuracy depends on the accuracy and amount of ground control points that are delivered**

**in the metadata of the Sentinel-1 scene. At the ground control point the location accuracy should be highest. We cannot give an error value in meter, since we do not have validation**

**points on the ground and cannot control the accuracy of the Sentinel-1 ground control points. However, from our experience and comparison with buoy drift data, the geolocation accuracy is expected to be in the order of the image resolution.**

Page 5 Lines 126 - 135 For a start I would suggest to change the order of the explanation and first mention the conversion from linear to log scale before the authors mention the scaling to integer values between 0 - 255 but this is the easier part. The more difficult part might be that we have a problem if there are no typos in these lines and I understood everything correctly. Log(0.013) = -1.88

dB while log(0.08) = -1.1 dB. If their minimum backscatter values are in dB as well (units missing!), it would mean, that the authors only use the range between -3.25dB – -1.88dB for HV and the range between -2.5dB – -1.1 dB. Could the authors please comment on this and even rephrase this part if I just misunderstood the authors? The problem I see is that their chosen backscatter range only represents a minor part of the backscatter range to be expected for sea ice in the logarithmic scale.

If these are the correct numbers, the authors might as well want to check the calibration routines for their data.

**We changed the order of explanation and first mention the conversion from linear to log scale before the scaling to integer values. There has been a typo with regards to the units: the values of the brightness boundaries were given in B and not dB. We corrected that, added**

**the units and included a histogram of a representative image pair (Figure 1) to illustrate the chosen boundaries and show the representative image pair after the conversion into the integer range (Figure 2).**

Page 6 Line 166 'serves as a quality estimate of the matching performance' - After it has been shown by Hollands, Linow and Dierking (2015) that there is no relation between the matching error and the correlation coefficient I would prefer a proof why the authors can use it as a quality measure. Even their Fig. 7 shows that the authors also dismiss good values, using the correlation coefficient as a quality value. Admittedly there is a group of large error values in their histogram but I wonder if this is significant. A correlation coefficient is only meaningful if the respective texture is characteristic enough. - I suggest to google Anscombe's quartet.

**We agree and removed the claim that the cross correlation value could serve as quality estimate. However, using the considered validation data, we found for our data type, time period and area of interest that the probability for large errors decreases with increasing cross correlation value. The following was added to Section 3:**

*We found that the probability for a large $D$ value (representative for the error) decreases with increasing maximum cross coefficient value $MCC$. Therefore we suggest to exclude matches with a $MCC$ value below a certain threshold $MCC_{min}$. This option is embedded into the*

*algorithm, but can easily be adjusted or turned off by setting $MCC_{min} = 0$.*

Page 6 Line 173 - 176 'To filter outliers, ... removed' - I have to admit, it would help me, if the authors could describe this outlier removal in more detail - based on the current description it is difficult to evaluate what the authors actually did.

     **The algorithm description has been changed and more details have been included. A subsection 'II Filter' and Figure 4 have been added to describe and illustrate the filtering**

**process.**

     Page 6 Line 177 - 181 'The remaining feature vectors ... neighbouring feature tracking vectors' - Just for the better understanding: What happens if there is a large area with no vectors at all framed by a few sparse vectors. Would the authors just triangulate over the whole area (potentially containing deformation or shear zones)?

     **If the considered area lies in between three feature tracking vectors, we triangulate over the area to provide the first guess. This initial drift estimate however, will then be adjusted by the pattern-matching approach. If the closest feature-tracking vector is far away, we apply the lowest restrictions defined by $d_{max}$. We found a useful value for $d_{max}$ for our area and time**

**period of interest to be 100 pixels, meaning that the search area is defined by an 8 km radius around the first guess. The lowest restrictions can easily be adjusted according to expected ice conditions and computational performance.**

     Page 6 Line 181-183 'To provide a drift estimate ... combination of x1 and y1.' - similar to Line

173 - 176 it is hard to say, what the authors actually did. May be the authors could add some details, making it easier to follow.

     **The algorithm description has been changed and more details have been included. A subsection 'III First guess' and Figure 5 have been added to describe and illustrate the process that leads to the first guess.**

     Page 6 Line 187- 190 I find it a bit confusing that we have a given size of the window before it is tuned. The same is true for dmin and dmax: It only became clear when I reached section 3.3. I would suggest that the authors mention here that they are going to identify the optimal parameters and may be as well why the authors decided to choose formula (4) for the window size.

**We changed the description of the pattern-matching step and adjusted the order according to this comment. We clarify which parameters need to be specified for the introduced pattern-matching procedure and how we find the recommended setting for each parameter.**

Page 7 Figure 1 It would be interesting so see a SAR image for the same area and may be a drift vector field. Is it correct that there is land where the distances are low and sea ice where the distance colour scale is saturated- Actually the authors already anticipate a result of their parameter tuning here. That makes it difficult to read. May be the authors should reorganise this part.

**We added Figure 2 to illustrate the related SAR images and show the corresponding feature-tracking vectors in Figure 3. The colour scales of the left and middle panel represent the first guess of the end positions on $SAR_2$ and the colour scale of the right panel indicates the distance to the closest feature tracking vectors, i.e. values of $d = 10$ represent 0-10 pixel distance to the closest feature-tracking vector and values of $d = 100$ represent 100-$\infty$ pixel distance. We split the figure and changed the algorithm description accordingly to make the process better understandable.**

Page 7 Line 195 '-beta +beta with step delta beta' - it is confusing that the authors suddenly start to introduce rotation as well since it has not been mentioned beforehand. The authors should have at least introduced it in section 3.2 II.

**The algorithm description has been changed accordingly and rotation is introduced in the new subsection 'IV Pattern-matching'.**

Section 3.2 page 5-7 Given that this section is meant to be the innovative part of this study I suggest restructuring it, to make it more concise. Right now, it is quite confusing and has varying level of detail and order (e.g. the window size question is a specific cross correlation question. I would urge the authors to state clearly when they introduce a parameter which they want to tune in the later course of the paper. Additionally I would suggest adding a flow chart, highlighting the steps, described in this paper.

**We adjusted the algorithm description accordingly and added more details. Figure 3 includes a flow chart and respective example images to illustrate the algorithm steps and the resulting products.**

Page 8 Formula 6 Why did the authors choose this distance measure instead of the RMSD in Formula 5?

**The $RMSD$ equation (previous Equation 5) and the comparison to the manually drawn vectors have been removed. The distance measure $D$ (previous Equation 6) has been used to get an individual error value for each compared vector pair, consisting of one validation vector and one algorithm vector. Since we found a logarithmic error distribution for the buoy comparison, a mean value as expressed by the $RMSD$ does not represent the found distribution.**

Section 4.1 Honestly, I would suggest skipping this section - it is not surprising that the logarithmic scaling leads to a higher number of features since the logarithmic histogram scaling favours the structures in the sea ice which are mainly represented in the shadow and medium backscatter values but hardly in the highlights.

**We skipped this Section and briefly mention in the data pre-processing why a logarithmic distribution is used:**

*Using a logarithmic scaling provides a keypoint distribution for the feature tracking algorithm that depends less on high peak values, while the total number of vectors increases.*

Page 10 Section 4.2 / Table 2 I have various questions:

- I understood that the authors tuned their Influence domain parameter dmax based on one image pair over Fram strait as well as the side length for their template but how did the authors tune their Dmin value and the MCCmin value?

**The parameter tuning was removed from the manuscript. Instead, useful restrictions that limit the computational effort of the pattern-matching were found and a useful $MCC_{min}$ value was found according to the error distribution from the buoy comparison.**

- 70 x 70 pixel for t1 means that their correlation window covers an area of approx. 6.3 x 6.3 km - how does this go along with their claim to resolve deformation and shear zones?

**We agree that this resolution is not sufficient and changed the recommended setting to $34 \times 34$ pixels in order to be consistent with our goal. We added the following to Subsection 'IV Pattern-matching':**

*The size of the small template $t_{1s} \times t_{1s}$ defines the considered area that is tracked from one image to the next and hence, affects the resolution of the resulting drift product. In order to be consistent with the resolution of the feature-tracking step and achieve our goal of a sea ice drift product with a spatial scaling of less than $5\,km$, we use the size of the feature-tracking patch of the pyramid level with the highest resolution to define the size of $t_1$. That means, we use $t_{s1} = 34\,pixels\ (2.7\,km)$.*

- Since their influence domain influences the size of their search window t2 it would mean that the authors add a degree of freedom of +/-1.8 to +/-11.25 km to their first feature tracking based guess, which would push their 0.5 m/s maximum ice drift limit for the feature tracking to about 0.6 m/s - right? Its contribution would however vary depending on the time span between both images of the scene. For the same constant drift velocity (but speed variations with in the scene), an image pair with a longer time span would then show larger displacement differences with in the scene while having the same maximum degree of freedom of +/- 11.25 km like an image pair that has been acquired at the same day - this might cause a problem, don't the authors think?

**Yes, this is a good point and needs to be considered. We added the following to Section 5:**

*The current setting of the feature-tracking algorithm applies a maximum drift filter of 0.5 m/s. We found this to be a reasonable value for our time period and area of interest. However, when considering extreme drift situations in Fram Strait and a short time interval between image acquisitions, this threshold should be adjusted.*

*During a KV Svalbard cruise in summer 2016, we deployed three GPS tracker in Fram Strait that recorded their positions with a temporal resolution of 5-30 min between $8^{th}$ July until $9^{th}$ September 2016 in an area covering $75°$ N to $80°$ N and $4°$ W to $14°$ W. Considering the displacements with 30 min interval, we found velocities above 0.5 m/s on a few occasions, when the tidal motion adds to an exceptionally fast ice drift.*

*The GPS data from the hovercraft expedition FRAM2014-2015 (https://sabvabaa.nersc.no), that was collected with a temporal resolution of 10 s between $31^{st}$ August 2014 until $6^{th}$ July 2015, did not reveal a single 30 min interval during which the hovercraft was moved by ice drift more than 0.45 m/s. The hovercraft expedition started at 280 km south from the North Pole towards the Siberian coast, crossed the Arctic Ocean towards Greenland and was picked up in the north-western part of Fram Strait.*

*In case the estimated drift from feature-tracking reaches velocities close to 0.5 m/s, the pattern-matching step might add an additional degree of freedom of up to 8 km, which could eventually lead to a higher drift result than 0.5 m/s, depending on the time interval between the acquisitions. The smaller the time difference, the larger is the potentially added velocity. In order to be consistent when combining the drift information from several image pairs with different timings, one should apply a maximum drift filter on the final drift product of the presented algorithm that has the same maximum velocity as the feature-tracking filter. The corresponding function is implemented in the distributed open-source algorithm.*

Page 10 Section 4.3 line 249: 'on a grid with 8 km spacing' - I suggest to summarize the information of their resulting product somewhere. It is not necessarily obvious to find the information on their grid spacing in the Parameter tuning and Computational Efficiency Section.

**The considered grid is not meant as a given parameter of the resulting product, but serves only to provide an estimate for the computational efficiency of the presented approach. The points of interest, given in longitude and latitude, represent the input for the algorithm. This can be the position of a ship, the grid of a model or an evenly spaced grid with any wanted resolution. The algorithm includes a routine that can derive points of interest in lon-lat on an evenly spaced grid. We hope that the changed algorithm description improved the explanation regarding points of interest and considered grid.**

Page 10 Section 4.3 Given the resolution of 8 x 8 km even pattern matching only based algorithms show a similar performance or even better. But I admit that the robustness to rotational motion is very useful in the marginal ice zone, where many of the pure pattern matching algorithms fail.

**We changed the resolution of the example and put more focus on the rotational motion.**

Page 10 Section 4.4 line 261: What does the size of 34 pixel mean- Is the feature described as a patch of 34 x 34 side length- May be the authors should add a short explanation to their feature tracking part on page 5.

**We removed this Section and added a description of the considered feature patch sizes to Subsection 'I Feature-tracking'.**

Page 12 Line 268-271: Why do the authors choose a minimum Cross Correlation Coefficient of 0.35? If the authors found a logarithmic function their distance distribution seems to follow, the authors could name it. Otherwise less strict term would be that the distance distribution seems to show a logarithmic behaviour or something like this. A peak at 300m is not necessarily meaningful (e.g. what would be the peak without their Cross Correlation Threshold? How many drift vectors form a peak?) but even if the authors have a peak, it does only represent the systematic component of the error and not the random one. In order to identify the distribution I would suggest smoothing the histogram and fitting a distribution to it.

**We smoothed the histogram and fitted a logarithmic normal distribution to it. We found the chosen minimum cross correlation coefficient $MCC_{min} = 0.4$ by plotting $MCC$ values against distance $D$, that represents the error. This is now shown in Figure 11.**

Page 14 Table 4: I would think that it is not the best approach to validate an algorithm based on the drift vectors I tuned it to. For a real validation the authors need at least another independent image pair with an independent set of manually derived drift vectors. I would strongly encourage the authors to change this! The authors compare apple with oranges if the authors compare an algorithm tuned to this specific scene with algorithms like the one from CMEMS. Additionally it would be great, if the authors could quantify both systematic and random error.

**We removed the parameter tuning and do not compare our results against the manually drawn vectors anymore. Figure 11 is included to illustrate systematic and random error.**

Page 14 Line 290: 'To further estimate the accuracy of the algorithm ...' - here it would be interesting to see, how the other algorithms perform as well. Additionally it would be great, if the authors could quantify both systematic and random error. The authors might want to check the regular validation document for the CMEMS ice drift as a start: http:myocean.met.noSIW-TACdocmyo-wp14-siw-dtu-icedrift-glob-obs-validation_latest.pdf The peak of a distribution is no error value!

**We illustrate the error in Figure 11 according to this suggestion. We removed the compari-son with CMEMS and simple feature tracking, since we don't have drift results of these two algorithms at the buoy locations.**

Page 14 Line 302-303: 'Hence, ... image resolution' I agree there are various factors influenc-ing the result of the algorithm and thereby influencing the validation but I cannot agree with this statement. It might be but the authors have not shown this yet!
**We removed these two sentences.**

**3    Technical corrections**

Page 1 Line 5: 'respective advantages of the two approaches' - the authors should emphasise in more detail what the advantages are, since this is the basic justification for this paper and this not only in the abstract but in the introduction/motivation as well

**The feature-tracking and pattern-matching description in Section 1 has been improved and**
**the corresponding part in the abstract has been changed to:**
*Feature-tracking produces an initial drift estimate and limits the search area for the consecutive pattern-matching, that provides small to medium scale drift adjustments and normalised cross coefficient values. The algorithm is designed to combine the two approaches in the most meaningful way in order to benefit from the respective advantages. The main advantages of the*
*considered feature-tracking approach are the computational efficiency and the independence of the vectors in terms of position, lengths, direction and rotation. Pattern-matching on the other side allows better control over vector positioning and resolution.*

Page 3 Line 37 'covers the Arctic every week with a spatial resolution of 5 km' - I'm not sure but
the authors might want to check it: as far as I know the, RGPS covers a large part of the Western Arctic Ocean but not the entire Arctic, due to the acquisition area of Radarsat. Up to my knowledge, the 5 x 5 km spatial resolution is a gridded drift field, which does not necessarily represent the actual spatial resolution, given that the RGPS searches features in a 10 or 25 km grid respectively. See also the RGPS Data User-s Handbook (Fig. 1 and Fig. 2)
**We changed the sentence to:**
*The geophysical processor system from Kwok et al. (1990) has been used to calculate sea ice drift fields in particular over the Western Arctic (depending on SAR coverage) once per week with a spatial resolution of 10-25 km for the time period 1997–2012. This extensive dataset makes use of SAR data from Radarsat-1 and ENVISAT (Environmental Satellite).*

Page 3 Line 73 'respective advantages' - If possible, be clearer about the respective advantages and summarise them here together with the disadvantages the authors still have and those the authors bypass with their approach.

**The feature-tracking and pattern-matching description in Section 1 has been improved and**

**in addition, we added to Section 1:**

*The main advantages of the considered feature-tracking approach are the computational efficiency and the independence of the vectors in terms of position, lengths, direction and rotation. Pattern-matching on the other side allows better control over vector positioning and resolution, which is a necessity for computing divergence, shear and total deformation.*

Page 2 Line 44 'pattern-marching and feature tracking respectively' - even terms are somehow flexible: I would claim, that Komarov and Barber do somehow a basic feature tracking as well, since they identify features, with certain characteristics before the correlate them - in that way, they have implemented the search for descriptors in a way. The use of correlation does not necessary mean that the approach is a pattern matching approach, since the correlation itself is the distance measure only, that is used to assess how similar a feature or a pattern is, compared to the reference. It might be a bit pedantic, but the authors might still want to give it a second thought.

**We changed the sentence to:**

*Komarov and Barber (2014) and Muckenhuber et al. (2016) have evaluated the sea ice drift*

*retrieval performance of dual-polarisation SAR imagery using a combination of phase/cross-correlation and feature-tracking based on corner detetction respectively.*

Page 2 Line 52-55 'Making use ... Copernicus.eu).' - I agree, that it is an important product, which should definitely be mentioned in the frame of this article but I think, the statement does not really fit there where it is right now because it interrupts their motivation.

**We moved the sentence into the paragraph above.**

Page 6 Line 185 -186 'Figure 2 shows...' - I would suggest moving the sentence a few sentence down to Line 195 after '...correlation value is returned

**The method description has been restructured taking this comment into account. We refer to this Figure at a later point in the description.**

Page 10 Section 4.3 line 252-254: 'NB: The vectors near ... treated with caution' - I completely agree but it is no question of computational efficiency

**This sentence has been removed.**

Page 10 Section 4.4 line 256: Strictly speaking the authors should compare their estimated drift vectors to their manually derived vectors and not the other way round and the authors estimate a drift vector and do not calculate it but this is a minor technical issue I guess.

**The comparison with the manually drawn vectors have been removed and we took this comment into account, when describing the comparison to the GPS buoy dataset.**

Page 11 Table 3: Is it correct, that their drift estimation is only based on HV polarisation- I guess the authors should state it somewhere in the beginning. Given their experience with dual pol motion
tracking, I assumed that the authors used both polarisations here as well- I suggest being clearer about it from the beginning, if this is the case.

**Yes, the considered drift estimates in this work are based on HV. We added the following to Section 2:**

*The introduced algorithm can utilise both HH and HV channel. However, the focus of this paper*
*is put on using HV polarisation, since this channel provides on average four times more feature tracking vectors than HH (Muckenhuber et al., 2016), representing a better initial drift estimate for the combined algorithm.*

Page 15 Line 311: 'The parameters can easily be varied...' - a short tabular overview on the range
for the individual parameters and their effect on the algorithm performance would be nice even though probably difficult.

**We updated the algorithm description taking this comment into account. The parameters and their effect on the drift result are now explained more in detail for a better understanding of eventual changes from the recommended setting. The possible range of the parameters $t_{s1}$,**
$t_{s2}$**, $\beta$, $\Delta\beta$, $d_{min}$ and $d_{max}$ is not limited.**

Page 15 Line 329: 'the real sea ice velocity' - the velocity the authors observe is not wrong, they might underestimate the speed and its variation as well as the variation of the drift direction but velocity is defined as distance per time, and the resulting velocity vector, being a sum of velocity
vector variations over the observation interval is the resulting velocity vector. A higher temporal resolution is interesting but it is as interesting and influences the 'realness' of their velocity vector the same way higher spatial resolution does. It would be great if the authors could give this phrase a second thought.

**We removed the term 'real sea ice velocity' and changed it to** *sea ice displacements with*
*higher temporal resolution***, that** *reveal more details e.g. rotational motion due to tides***. Section 5 has been updated accordingly.**

Thanks again for your comments. We are looking forward to your reply!

Best regards,

S. Muckenhuber and S. Sandven

---

## Author Comment (AC2) · 13 Feb 2017

Manuscript prepared for J. Name
with version 2015/09/17 7.94 Copernicus papers of the LaTeX class copernicus.cls.
Date: 13 February 2017

**Response to Referee # 2**

**'Open-source sea ice drift algorithm for Sentinel-1 SAR imagery using a combination of feature-tracking and pattern-matching'**

Stefan Muckenhuber and Stein Sandven

Nansen Environmental and Remote Sensing Center (NERSC), Thormøhlensgate 47,
5006 Bergen, Norway

*Correspondence to:* S. Muckenhuber (stefan.muckenhuber@nersc.no)

Dear Referee # 2,

Thank you very much for helping us improving our paper.

Please find here the **answers** to your comments and the corresponding *changes in manuscript*:

5 **1 Major comments**

Introduction, P2: The manuscript should include some additional background information on the sea ice drift in the study area (with possible references): what are the magnitudes of typical ice drift in the study area and whole Arctic (e.g. cm/s and daily) and in which areas they are located and which are their causes?

10 **We added references and the following to Section 1:**

*Early work from Nansen (1902) established the rule-of-thumb that sea ice velocity resembles 2 % of the surface wind speed with a drift direction of about 45° to the right (Northern Hemisphere) of the wind. This wind driven explanation can give a rough estimate for instantaneous ice velocities. However, the respective influence of wind and ocean current strongly depends on the*
15 *temporal and spatial scale. Only about 50 % of the long-term (several months) averaged ice drift in the Arctic can be explained by geostrophic winds, whereas the rest is related to mean ocean circulation. This proportion increases to more than 70 % explained by wind, when considering shorter time scales (days to weeks). The wind fails to explain large-scale ice divergence patterns and its influence decreases towards the coast (Thorndike and Colony, 1982).*

20 *Using GPS drift data from the International Arctic Buoy Program (IABP), Rampal et al. (2009) analysed the general circulation of the Arctic sea ice velocity field and found that the fluctuations follow the same diffusive regime as turbulent flows in other geophysical fluids. The monthly mean drift using 12 h displacements was found to be in the order of 0.05 to 0.1 m/s and showed a*

*strong seasonal cycle with minimum in April and maximum in October. The IABP dataset also*
*revealed a positive trend in the mean Arctic sea ice speed of +17 % per decade for winter and*
*+8.5 % for summer considering the time period 1979–2007. This is unlikely to be the consequence*
*of increased external forcing. Instead, the thinning of the ice cover is suggested to decrease the*
*mechanical strength which eventually causes higher speed given a constant external forcing*
*(Rampal et al.; 2009b).*

*Fram Strait represents the main gate for Arctic ice export and high drift velocities are generally*
*found in this area with direction southward. Based on moored Doppler Current Meters mounted*
*near 79° N 5° W, Widell et al. (2003) found an average southward velocity of 0.16 m/s for the*
*period 1996–2000. Daily averaged values were usually in the range 0–0.5 m/s with very few*
*occasions above 0.5 m/s.*

Method/Feature tracking, P5: It is mentioned that 'The best match is accepted if the ratio of the
two shortest Hamming Distances is below 0.75.'. Explain why this is done and how the threshold
was selected. Probably to reduce possibilty of similarization errors? What is magnitude of typical
Hamming distances? If they are small, then 0.75 has quite different meaning than for larger values.
I assume that the ratio is the ratio of the shortest and second shortest Hamming distance (also write
this in the text).

**The Hamming distances are embedded in the feature-tracking algorithm and are not re-
turned during application of the algorithm. This makes the evaluation of the value distribution
difficult. However, based on visual interpretation of drift results using different Hamming
distances, Muckenhuber et al. (2016) found a suitable value for our time period and area of
interest. We added the following to Section 3:**
*The best match is accepted if the ratio of the shortest and second shortest Hamming distances is*
*below a certain threshold. Given a suitable threshold, the ratio test will discard a high number of*
*false matches, while eliminating only a few correct matches.*
*Muckenhuber et al. (2016) found the most suitable parameter setting for our area and time period*
*of interest, including a Hamming distance threshold of 0.75, ...*

Method/Combination, P6: To filter outliers each vector is simulated using two functions which are
LS solutions... This need more explanation. Why third degree polynomial has been used and which
data are used in the LS fit? Also in the extrapolation is also performed using a LS solutions. Also
describe this in more detail. How is the traingulation constructed (Delauney?) in interpolation?

**Section 2 has been changed according to this comment. We included more detailed descrip-
tions of LS solutions and triangulation. Equations were added to specify the procedure. The**

60    Parameter Tuning/Validation, P10: It is not exlpicitly mentioned which data were used for the parameter tuning. Were all the vlaidation data used for this? Then the validation with this data set is not fair as the algorithm has been tuned for this data. Then only the buoy data can be used for independent validation. Or if separate sets are used for parameter tuning and validation, indicate this in the manuscript.

65    **The Parameter Tuning Section has been removed and validation is now only done against buoy data.**

**2   Detailed comments**

P1L2: 'computanional' -> 'computationally'

**Agree, we changed the manuscript accordingly.**

70

P2L33: '90s' -> '90's'

**Agree, has been changed.**

P2L38: In the case of ENVISAT, rather give the name of the instrument i.e. ENVISAT ASAR,
75   could also mention that RADARSAT was an instrument of CSA and ENVISAT ASAR of ESA.

**Agree, has been changed.**

P3L88: '((' -> '('

**Agree, has been changed.**

80

P3L88: '...dual polarization support...' '..also in wide swath mode'. Also earlier instruments had a possibility to measure multiple polarizations but the covered area was small. This has been changed by RADARSAT-2 and SENTINEL-1.

**We changed the sentence to:**

85   *The mission includes two identical satellites, Sentinel-1A (launched in April 2014) and Sentinel-1B (launched in April 2016), each carrying a single C-band SAR with a centre frequency of 5.405 GHz and dual-polarisation support (HH+HV, VV+VH) also for wide swath mode.*

P3L90: Give also the acronyms for the mode i.e. EW GRDM (thes are generally used by ESA in
90   documentation and file names).

**Agree, has been added.**

P4L125: You can remove 'of 93m range x 87m azimuth', this information has already been given earlier.

95     **Agree, has been removed.**

P6 eq. 3: Here You give the formula for NCC. Also give the drift (dx,dy) detection as a formula, something like: $(dx, dy) = argmax_(k,l) in W NCC(x+k, y+l)$ Is NCC computed according to this equation or by applying FFT and IFFT (which has been applied in many algorithms to fasten the

100    computation)?

    **The matrix NCC is computed according to the new Equation 8 and FFT and IFFT are not applied. The used python function is matchTemplate from OpenCV (http://docs.opencv.org/2.4/modules/imgproc/doc/object_detection.html). We changed the pattern-matching description according to this comment and added the following:**

105    *The highest value in the matrix NCC, i.e. the the maximum normalised cross coefficient value $MCC$, represents the location of the best match and the corresponding location adjustment is given by $dx$ and $dy$.*

$$(\frac{1+t_{s2}-t_{s1}}{2} + dx, \frac{1+t_{s2}-t_{s1}}{2} + dy) = argmax(\boldsymbol{NCC}(x,y) \tag{1}$$

110

P6 Eq. 4: Define 'side' in the text.

    **This phrase has been removed and replaced by $t_{1s}$ and $t_{2s}$.**

P7 Fig 1 and Fig 2. Use a, b, and c for the subfigures and to refer to them.

115    **We added titles to the subfigures to refer to them and make the algorithm description easier understandable.**

P7 L195: Explain here what is denoted by 'beta'. It is is also in Fig. 2 caption.

    **The algorithm description has been changed according to this comment and the following**

120    **has been added:**

*To account for rotation adjustment, the matrix NCC is calculated several times: template $t_1$ is rotated around the initially estimated rotation $\alpha$ from $\alpha - \beta$ to $\alpha + \beta$ with step size $\Delta\beta$. The angle $\beta$ is the maximum additional rotation and represents therefore the rotation restriction. The NCC matrix with the highest cross coefficient value $MCC$ is returned.*

125

P7 Fig.2 (and text): Why rectangular/square templates has been used? A circular template would be much easier (symmetric) to rotate. Consider using a circular templates instead.

    **We agree with this comment. Regarding $t_1$ however, the current version of the used OpenCV function matchTemplate does not allow circular templates and work-arounds would influence**

130    **the result and the computational efficiency. We hope that a later version of matchTemplate will allow to use masks. Regarding $t_2$, we included a circular mask for the matching result to**

**limit the search area to a circle rather than a square.**

Logarithmic scaling P8-9: I think logarithmic scale is the typical presentation of SAR sigma0 and often a fixed scaling to gray tone imagery is used for SAR imagery, e.g. scaling between -30dB -> 0 dB. You could mention this fact on the manuscript. This also leads to the question if any other 'scaling' would produce even better results, e.g. applying some king of histogram derived image mapping (e.g. simple histogram equlization etc.). This could be one topic for further development.

**We agree. The corresponding section has been removed and the logarithmic scaling description has been moved to Section 3 and adjusted according to this comment. Muckenhuber et al. (2016) tested different scalings procedures on four representative image pairs to retrieve the best possible feature-tracking results. We apply the same scaling for pattern-matching for both computational efficiency and because we assume that a scaling that is preferable for feature-tracking is also preferable for pattern-matching. This assumption however, has not been proven and is certainly a topic for further development.**

P10/Computational efficiency: You give a time of less than 3.5 minutes here. Is this a typical execution time or just execution time for a randomly selected example. Could you give average execution times and deviations or maybe estimate for the worst case? Does the execution time increase linearly as a function of the number of vectors or is there some other kind of relationship?

**The given time is representative for an image pair with large overlap, good coverage with feature-tracking vectors and the given resolution of the final product. We adjusted and extended the Section Computational efficiency according to this comment. The step 'II Pattern-matching and III Combination' is proportional to the number of chosen points of interest, i.e. the number of drift vectors of the final product. The first two steps can be seen representative for all Sentinel-1 image pairs with $400 \times 400$ km coverage. We added a corresponding analysis of the different steps and the influencing parameters.**

P12 L270-271: also give the average D. 'peak' is not a correct word here, the histogram/distribution has many peaks, possibly You could use 'mode' here and also in the caption of Fig. 7.

**We agree. The error estimation has been changed accordingly. We now fit a logarithmic normal distribution to the histogram and found a median $e^{\mu} = 341.9$ m.**

P12 L276-277: The DTU method has not been documented very well in any publications I think. Also the reference given does not say much. I suppose there is not better reference for this?

**The comparison with the DTU drift field in Fram Strait has been removed. The DTU product however, is still mentioned in Section 1. We did not find any better reference than**

**Pedersen et al. (2015), http://www.seaice.dk/ and http://marine.copernicus.eu.**

P12 L279: '...used the nearest neighbors...' -> '...used the nearest neighbors (NN's)...' then NN can be used in Table 4.

**We agree. However, the corresponding comparison using image pair Fram Strait has been removed.**

P13 Fig. 6: Would it be possible to indicate the location of the detail in the coarse-scale image (without causing too much damage for the image)?

**We agree. However, the corresponding image has been removed.**

P14 Table 4: 'Average distance' -> 'Average NN distance' or something like that. Are the values after +- sign standard deviations or some multiples of standard deviation or something else? Include this information in the table or caption.

**The +- sign indicated one standard deviation. However, the corresponding comparison using image pair Fram Strait has been removed.**

Discussion: What is the possible error magnitude of the manually estimated drift (is it assumed to be sub-pixel, one pixel or more and what kind of possible error sources these vectors include?)?

**The estimated error is in the order of several 100 m. However, the corresponding comparison using image pair Fram Strait has been removed.**

P15 L319-320: Also ESA is going to improve their thermal noise removal by including more measurements along the azimuth direction. Probably this also could be mentioned. If necessary you can get more information on this from Nuno Miranda at ESA (nuno.miranda@esa.int).

**Thank you for this information. We contacted Nuno Miranda from ESA and added the following to Section 5:**

*The European Space Agency is also in the process of improving their thermal noise removal for Sentinel-1 imagery. Noise removal in range direction is driven by a function that takes measured noise power into account. Until now, noise measurements are done at the start of each data acquisition, i.e. every 10-20 minutes, and a linear interpolation is performed to provide noise values every 3 seconds. The distribution of noise measurements showed a bimodal shape and it was recently discovered that lower values are related to noise over ocean while higher values are related to noise over land. This means, that Sentinel-1 is able to sense the difference of the earth surface brightness temperature similar to a passive radiometer. When the data acquisition includes a transition from ocean to land or vice versa, the linear interpolation fails to track the noise variation. The successors of Sentinel-1A/B are planned to include more frequent noise*

*measurements. Until then, ESA wants to use the 8-10 echoes after the burst that are recorded while the transmitted pulse is still travelling and the instrument is measuring the noise. This will provide noise measurements every 0.9 seconds and allows to track the noise variations in more detail. In addition, ESA is planning to introduce a change in the data format during 2017 that* 210 *shall remove the noise shaping in azimuth. These efforts are expected to improve the performance of the presented algorithm significantly.*

**We thank Nuno Miranda in the Acknowledgement for the provided informations.**

Thanks again for your comments. We are looking forward to your reply!

215

Best regards,

S. Muckenhuber and S. Sandven

---

## Author Comment (AC3) · 13 Feb 2017

Manuscript prepared for J. Name
with version 2015/09/17 7.94 Copernicus papers of the LATEX class copernicus.cls.
Date: 13 February 2017

**Response to Referee # 3**

**'Open-source sea ice drift algorithm for Sentinel-1 SAR imagery using a combination of feature-tracking and pattern-matching'**

Stefan Muckenhuber and Stein Sandven

Nansen Environmental and Remote Sensing Center (NERSC), Thormøhlensgate 47,
5006 Bergen, Norway

*Correspondence to:* S. Muckenhuber (stefan.muckenhuber@nersc.no)

Dear Referee # 3,

Thank you very much for helping us improving our paper.

Please find here the **answers** to your comments and the corresponding ***changes in manuscript***:

5 **General comment**

A general impression after reviewing this manuscript is that it requires more work and provision of additional details before being ready for publication in TC. The authors are thus invited to revise their manuscript before a new version is submitted. Specifically, the following items should be addressed.

**We increased the level of detail and added new figures to improve the manuscript.**

10

**1 Description of the algorithm**

The 'pattern-matching' step is not well enough described and many questions are still open at the end of section 3.2.

**The pattern-matching description has been rewritten and more details have been added.**

15

1.a) The ordering of the sub-sections (I. Feature-Tracking, II. Pattern-matching, III. Combination) is maybe not optimal as you spend some of Section III to describe the rotation by angle beta (that should really go into II). Maybe it would be easier to follow if the sub-section followed the steps of the algorithms (feature-matching, fitting of polynomial for first-guess, filtering, patter-matching, 20 etc...).

**We changed to order according to this comment. The new subsections are: 'I Feature-tracking', 'II Filter', 'III First guess', 'IV Pattern-matching' and 'V Final Product'. We added Figure 3 incl. flow chart to illustrate the steps and the respective products.**

1.b) It is unclear if your pattern-matching step features a series of x,y shifts to maximize the cross-correlation in addition to the rotation by beta, or not. If you combine both x, y, and beta shifts, what is the relative order and does it matter?

**The pattern-matching description has been changed according to this comment. The matrix NCC(x,y), containing all normalised cross coefficient values for all possible x,y shift, is calculated several times: one for each rotation $\beta$. The highest cross coefficient value is found considering all NCC matrizes.**

1.c) As you recall in I. 'Feature-Tracking', the ORB algorithms also gives an information about the rotation angle (delta between centroid-based orientation of the matched features). Is this feature-matching first-guess of the rotation used at all? If yes, how; and if no, why not?

**This is a good point and we adjusted the algorithm according to this comment. We included the usage of the feature-tracking rotation: a filtered rotation field based on the rotation found for the individual features serves now as initial rotation for the pattern-matching step.**

1.d) What is 'the initial rotation between the two Sentinel-1 image' (line 194) and how is it computed? Is it the same value across the image?

**The 'initial rotation between two Sentinel-1 images' was derived as angle between the left edges of the images. It was calculated by re-projecting the left edge of the second image onto the projection of the first image. This is the same value for the entire scene. However, after including the feature-tracking rotation (see above), the algorithm is not using this rotation anymore, but rather a rotation field, that varies over the image (see $\alpha$ in Figure 5), based on the rotation of the individual features.**
**The 'initial rotation between two Sentinel-1 images' is still calculated since it allows to exclude the different projections of the two scenes and derive the actual rotation of the sea ice at each point of interest.**

1.e) In subsection II. 'Pattern-matching' you write the NCC formula for 'two equally sized windows'. But later you seem to use two unequally sized windows (size t1 in SAR1, size t2 in SAR2). What is the NCC formula do you then use? Of is size t2 related to the size of the search window while t1 is the size of the pattern? The questions above are mostly to give an impression of the level of details expected when you re-formulate this section. Your first manuscript contained quite some details on the methodology, and this new one requires at least as many details.

**We changed the pattern-matching description according to this comment and included a more detailed formulation of the NCC equation.**

**2  Validation against GPS data**

2.a) The choice of validation metric (the distance between the end points of the reference and estimated vectors) is not peculiar. Virtually all other studies use the RMSE along two components (e.g. u and v). And the logarithmic distribution of the errors is not discussed or exploited. Please also discuss the RMSDs in u and v components and compare your results with that of other investigators.

**We changed the validation procedure and fitted a logarithmic normal distribution to the histogram. We did not see any specific pattern when considering u and v component separately, but we added plots to further investigate the systematic and random error (Figure 11). To our knowledge, we are currently the only ones using this GPS dataset for validation. It is hard to compare these results with other drift products, since they resemble a different resolution and we don't have drift estimates at the buoy locations. However, we tried to make our validation procedure similar to the regular validation of the CMEMS ice drift to improve the possibility for future comparison.**

2.b) The N-ICE campaign deployed many buoys, but very much in the vicinity of the vessel Lance. How many different buoys enter your validation database, and what is the average distance between them? Are we sampling more than few kilometres in each SAR pair?

**The Norwegian Polar Institute provided us with data from 32 buoys. Based on that dataset, we automatically searched for fitting Sentinel-1 image pairs that provided more than 300 feature-tracking vectors and had a time differences of less than three days. We added a map with the resulting buoy trajectories (Figure 8) to illustrate location, spread and drift distance.**

2.c) N-ICE data should offer the possibility to discuss the accuracy when inside the pack versus at the marginal ice zone. Please see if you can segment your validation database to cover this. As you point out yourself, the added value of rotation should be most visible in the marginal ice zone.

**To describe the ice conditions during the collection of the validation data, we added the following to Section 2:**
*The ice conditions during the N-ICE2015 expedition are describe on the project website (http://www.npolar.no/en/projects/n-ice2015.html) as challenging. The observed ice pack, mainly consisting of 1.3-1.5 m thick multiyear and first-year ice, drifted faster than expected and was very dynamic. Closer to the ice edge, break up of ice floes has been observed due to rapid ice drift and the research camp had to be evacuated and re-established four times. This represents a good study field, since these challenging conditions are expected in our area and time period of*

*interest.*

**The automatic search algorithm, that allows to perform the validation on a high number of**

95 **image pairs, is only comparing location and timing of buoy and satellite data and does not**
**include any information on ice condition. To segment the validation dataset according to ice**
**condition, we would need to describe the ice conditions for each validation vector individually.**
**Unfortunately However, future work will cover experiments of the algorithm performance in**
**different ice conditions.**

100

2.d) Can you convince the reader (and the reviewer!) that the value of the maximum NCC indeed constitutes a quality measure (your Abstract)? Are matchups with lower NCC values really father away from GPS truth, than those with high NCC? Hollands et al. (2015) did not find any relation between the two. Is your threshold at 0.35 related to a significant drop in the documented

105 accuracy against the buoy drift? (Hollands, T. , Linow, S. and Dierking, W. (2015): Reliability Measures for Sea Ice Motion Re?trieval From Synthetic Aperture Radar Images , IEEE Journal of Selected Topics in Applied Earth Observations and Remote Sensing, 8 (1), pp. 67-75 . doi: 10.1109/JS-TARS.2014.2340572)

**We removed the term 'quality measure' throughout the manuscript. However, we found**
110 **that the probability for a high error decreases with increasing maximum cross coefficient**
**value (Figure 11) and added the following to the validation section:**
*We found that the probability for a large $D$ value (representative for the error) decreases with*
*increasing maximum cross coefficient value $MCC$. Therefore we suggest to exclude matches*
*with a $MCC$ value below a certain threshold $MCC_{min}$. This option is embedded into the*
115 *algorithm, but can easily be adjusted or turned off by setting $MCC_{min} = 0$. Based on the findings*
*shown in Figure 11, we recommend a cross coefficient threshold $MCC_{min} = 0.4$ for our time*
*period and area of interest.*
**A corresponding statement was added to the method section.**
**After changing the recommended size of the smaller template $t_1$ to $34 \times 34$ pixels (to be**
120 **consistent with the feature-tracking resolution and the aimed accuracy of the drift product),**
**we also adjusted the cross coefficient threshold to 0.4.**

2.e) You use a maximum velocity of 0.5 m/s for your feature-based results (line 171). Is this limit high-enough in view of your validation dataset in the Fram Strait region?

125 **To discuss the maximum velocity limit of 0.5 m/s, we added a general drift assessment to the**
**Introduction and the following to Section 5:**
*The current setting of the feature-tracking algorithm applies a maximum drift filter of 0.5 m/s.*
*We found this to be a reasonable value for our time period and area of interest. However, when*
*considering extreme drift situations in Fram Strait and a short time interval between image*

130 *acquisitions, this threshold should be adjusted.*

*As mentioned above, we deployed three GPS tracker in Fram Strait and they recorded their positions with a temporal resolution of 5-30 min between $8^{th}$ July until $9^{th}$ September 2016 in an area covering 75° N to 80° N and 4° W to 14° W. Considering the displacements with 30 min interval, we found velocities above 0.5 m/s on a few occasions, when the tidal motion adds to an*
135 *exceptionally fast ice drift.*

*The GPS data from the hovercraft expedition FRAM2014-2015 (https://sabvabaa.nersc.no), that was collected with a temporal resolution of 10 s between $31^{st}$ August 2014 until $6^{th}$ July 2015, did not reveal a single 30 min interval during which the hovercraft was moved by ice drift more than 0.45 m/s. The hovercraft expedition started at 280 km south from the North Pole*
140 *towards the Siberian coast, crossed the Arctic Ocean towards Greenland and was picked up in the north-western part of Fram Strait.*

**We removed the validation procedure with the considered image pair over Fram Strait, even though it did not include velocities above 0.5 m/s.**

145     Finally, it would be good if the revision of the paper could include a thorough discussion of the robustness of the combined method to the success of the feature-matching step (not in terms of computation cost, but of introduction of artefacts).

**We did not find any artefacts in the test images that we considered so far. However, we would like to increase the number of image pairs significantly and produce large drift field**
150 **datasets (and corresponding divergence, shear and total deformation datasets) to further evaluate the algorithm performance and investigate its robustness in terms of artefacts. To do that, we recently established a cooperation with TU Wien to embed our algorithm into their super-computing facility and learn from their experience with handling large Sentinel-1 datasets. The aim of this paper is mainly the presentation of the methodology and our next**
155 **goal is the application on large datasets for further testing. To specify our next steps, we added the following to Section 5:**

*Our next step is to embed the algorithm into a super-computing facility to further test the performance in different regions, time periods and ice conditions. The goal is to deliver large ice drift datasets and open-source operational sea ice drift products with a spatial resolution of less*
160 *than 5 km.*

    Thanks again for your comments. We are looking forward to your reply!

    Best regards,
165 S. Muckenhuber and S. Sandven

---

## Referee Report (RR1)

**Review on "Open-source sea ice drift algorithm for Sentinel-1 SAR imagery using a combination of feature-tracking and pattern matching"**

Dear authors,
thanks a lot for all your work on your paper. It is a pleasure to see a paper on such an interesting topic evolving. I hope my comments help you to improve your paper even further (even though I'm aware that they are numerous :- ( )

**Comments to the answers to Referee 1**

Most of the comments are fully satisfying but not all. They might double with later comments on the revised version of the article.

Page 2 Line 42-46: The problem with the backscatter values is not solved by correcting a typo and claiming that the values were shown in Bell (a very uncommon unit for backscatter by the way).  The correction of the typo shifts the backscatter values into the right magnitude. However, given a noise level of 22 dB for Sentinel-1 I would still like to question the calibration routines used. Using scaling suggested by the authors on correctly calibrated S1 data would mean to focus on a value range mainly containing noise (-32.5 dB - -22 dB) and only partly containing potentially meaningful backscatter values in the range -22 dB to -18.86 dB. The authors might want to check again there calibration routines.

Page 8 Line 245: the use of logarithmic scaling is the standard for all SAR based sea ice research and professional monitoring by universities, ice services and research institutes. But it is good to see that this promising approach uses dB scaling now as well.

Page 8 Line 255: I did not want to vote against parameter tuning or the validation using displacement vectors derived from satellite image pairs. Both instruments are extremely useful. I just thought that a validation using the same dataset originally used for tuning is not the best approach for an independent validation. It is a pity that the authors decided to skip both parts completely but then it is for sure the most effective approach to deal with my remark.

Page 9 Line 278 – 301 I like this new paragraph you added into section5, but as far as I can sayit does not consider the specific problem, I outlined in my question and they suggest to handle with this paragraph (the same search window size means different degrees of freedom depending on the time span between the two considered image acquisitions)  - It does however discuss the general aspect of time span regarding the "hard" 0.5 m/s maximum speed threshold but I guess, that's o.k.

**The article itself**

**General comments**

The article improved a lot since the last time and the authors handled the comments of the reviewers in a very efficient manner. It reads like a completely new manuscript. Personally I think that the authors tend to use superlatives like "optimize" or "most meaningful" a bit too often given the flexibility of their approach (various changes between both versions) but nevertheless it seems to be on a right track now and worth being published in "The Cryosphere" after major revisions. Within the frame of the revision it would however be necessary to discuss the differences of this paper to

the recently published paper by Anton Korosov and Pierre Rampal for the same institute on the same topic: *A.Korosov and P. Rampal (2017): "A Combination of Feature Tracking and Pattern Matching with Optimal Parametrization for Sea Ice Drift Retrieval from SAR Data", Remote Sens. 2017, 9(3), 258; doi:10.3390/rs9030258,* which looks quite similar to me. Personally, I think that this parallel publishing from the same institute is at least a bit unfortunate (even though I'm aware, that the manuscript at hand was originally submitted earlier than the now published article by Korosov and Rampal).

**Specific comments**

Page 4 Line 113-115 I'm not sure how much of the vector independence from the feature tracking actually exists, given the overlap of features from the various resolution levels, the filtering and polynomial least square fitting to identify outliers and the linear interpolation to get first guess estimates for the subsequent pattern matching. What do the authors think?

Page 6 Line 173 – 192 I would like to repeat my statement from the last review: there seems to be something wrong with your calibration routine and it is not only a typo. It would be great for sea ice research if S1 had a sensitivity of -32.5 of even -25dB only but its noise level is at -22dB, which means that especially your lower boundaries for the scaling cause a problem for correctly calibrated S1 images.

Page 6 Line 186 The scaling to 8 bit / 256 grey level is an input requirement for feature tracking using the ORB algorithm but reduces the signal variation. This reduces the effectiveness of the NCC which can handle any numerical precision you like. It is not a problem for medium to larger resolutions (like CMES service provided by the DTU) since there are still enough pixel in each correlation window but as soon as you go to higher resolutions, even slight variations in the signal become more important (since there might be not so many clear variations left in a smaller window). Have the authors tried to use the original dB backscatter values for the pattern matching instead of the scaled 256 greyscale image?

Page 9, Line 223-224 As I mentioned before, the use of logarithmic scaling for SAR is common and I'm actually surprised that the authors did not do it from the start.

II Filters: I have various questions:

1. For the polynomial fit the authors use all vectors (those from coarser resolution levels and those from finer resolution levels – a single pixel on the coarser resolution level covers about 3.6 pixel on the finest resolution level: subpixel uncertainty of +/-145 m vs a subpixel uncertainty of +/- 40 m?) – isn't this a bit of a problem if you merge it all in one polynomial surface and then try calculate if the predicted vector is 100 px away from your polynomial surface?
2. You are applying a least square fit to an angle and calculate a polynomial surface from it. How do you handle the zero-crossing problem and all the other problems you are facing once you start averaging and interpolating angle values?
3. It seems that your angular polynomial surface has a peak at about 130°. Just for curiosity:  Is this the rotation between both images?

Page 11 Line 272-273: least square fit and linear function for angles? How do the authors handle the specific problems related with this (see my comments on II Filters)

Page 11 Line 285-287: "The uncertainty … vector" I would claim that it is not the uncertainty of the estimate but the representativeness of the estimate and that it does not only depends on the distance to the next feature tracking based vector but as well on the heterogeneity of the other surrounding feature tracking based drift vectors.

Page 12 Line 307-308: As far as I know the idea to restrict the smaller correlation template to a circle has been strongly propagated by Roberto Saldo from the DTU who uses it for his drift algorithm. I'm not sure if this is what the authors mean. While I'm aware of the advantages to limit the smaller correlation template to a circle, I have no idea why you would want to restrict your search window to a circle (if not only for the reason to safe computation) but may be the authors want to outline the advantages of a circled search window while having a rectangular correlation template. If they meant to describe a circled correlation window the authors want to refer to Roberto Saldo (if there is no publication regarding this matter may be as personal communication).

Page 13 Line 318-319: "In order to be consistent with the resolution of the feature tracking". What is the advantage of being consistent with the resolution of the feature tracking?

Page 14 Line 344-352 A higher MCC value seems to indicate a higher chance of a better match for your algorithm compared to buoy data. This is an interesting finding since most pattern matching algorithms have lower MCC_min values or none at all and since it was shown, that there is no direct relation between error and correlation coefficient.
This could mean that the author's initial guess and search strategy are less effective than those of pure pattern matching algorithms (and thereby needing a higher correlation to find a suitable match) especially since the maximum shift from the feature tracking based first guess is 1.6 (d=10) to 16 km (d=100) at maximum. Every difference above that is a hint that the first guess did not work or that the buoy drift has nothing to do with the surrounding ice conditions (Figure 11 a). Additionally I suggest checking the spatial distribution of the vectors the authors rejected due to their low MCC and high spatial distance (e.g. a shearing zone within the correlation window would for example cause as well a drop in the correlation value and potentially a larger difference in displacement relatively to an originally close drift buoy in another drift regime) and study the behavior compared to manually tracked vectors from the image pair ensure that the difference comes really from the algorithm and not from the limitations of the comparison with buoys, the authors are well aware of (page 20 line 439 – 450).

Page 15 3.3 Comparison with buoy data: Just for my understanding: Did the authors center a 34x34 pixel window at the original buoy position? The original buoy position has been acquired every hour or has been interpolate to hourly intervals

Page 15 Line 356 "Each pair yielded more than … three days"  I'm not sure but it would probably more interesting to mention the number of vectors, which the recent version of the feature tracking component provides, using HV only and dB scaling, if at all …

Page 17 Figure 9 and 10: Based on the Figures it is difficult to see an advantage for the variation of the search window size or the rotation angle aside from computational efficiency. Would the authors agree?

Page 18 Fig 11 a: something seems to be wrong with the legend

Page 18 Fig 11 b: how would the fit look like if the authors would include the values you rejected due to their low correlation value?

Page 18 Line 407 / Page 20 Line 441 "median and 341.9 m" What's the significance? Obviously it means that 50% of your data has an error of less than 340 m with respect to the buoy data but it means as well that 50 % have an error, that is larger (and based on the distribution: 1% < 20m, 5% < 43m, 10% < 68m, 68% (1 sigma criterion) < 620 m, 95% (2 sigma criterion) < 2700 m and 99% (3 sigma criterion) < 6400m). But could the authors give a hint what it could mean regarding the quality of the algorithm, given the various influences they identified which could have biased the result? May be a solution would again be to use the manually derived vectors from the original draft for it to account for the algorithm "accuracy" only (well with an "uncertainty" hidden in the manually collected drift vectors of course)?

Page 20 Line 458-460 "The parameters can … strong rotation)." I have a fair idea which parameters to vary to meet changes in computational power, area of interest and expected ice conditions but I have no idea why availability of time, computational power and number of image pairs are mentioned individually since the only parameter behind it is computation time or rather the selection of a finer or coarser resolution. Additionally I would be interested what parameter to change to influence the accuracy!

Page 20 Line 470-484 I guess this paragraph is based on information you received from Nuno Miranda. It would be useful to refer to it as Personal Communication in the text since otherwise the readers might wonder where this information comes from.

Page 21 Line 487-490 "Our next step … less than 5 km" That's really nice! I'm really looking forward to such a dataset with a good quality. However, given that most of the S1-data in the Arctic and Antarctic is in HH polarization (in Antarctica completely) wouldn't it been better to perform this study in HH from the beginning, even if there seems to be an advantage for HV in the feature tracking?

Page 21 Line 503 "better coverage in HH pol" – that's nicely put – the S1 observation scenario does not plan any HV acquisition Antarctic part of the southern ocean and none in the central Arctic ocean but it could of be a reason to demonstrate the need for HV data in these regions as well.

Page 21 Line 507 - 510 "Therefore …future work" A good idea! Just a thought I would like to add. Feature tracking needs linear features which tend to depend a bit more on incidence angle, orbit, and changes in ice condition in HH than in HV and depending on the robustness of your feature detector it might even be quite robust to the quite strong noise in the S1-HV data (it is actually impressive how many feature tracking based vectors you were able to retrieve from the HV scene compared to the HH one). This could explain the better performance of the feature tracking part for the HV component. Cross correlation based pattern matching however is less sensitive to changes in linear feature and more sensitive to areal pattern changes which might potentially favor the HH channel – but as I said: just a thought.

**Technical comments**

Page 1 Line 9 "pre-processing of S1 data … has been optimized" – Are the authors refereeing to their use of the logarithmic scale for the backscatter again? As far as I'm aware that is all you did and it is a

bit of disappointing if the abstract promises an optimized preprocessing while the article offers the conversion to dB only.

Page 1 Line 10-13 I'm not sure if computational efficiency is necessary in the abstract, the authors might want to give it a second thought but I'm fine if you want to keep it there.

Introduction: I think the introduction improved a lot but especially of page 3 I miss something running like a common thread through it. It reads like an accumulation of independent facts.

Page 3 Line 69 If I'm not mistaken, Hollands and Dierking (2011) implemented their own modified version of the algorithm and did not just continue the work on the algorithm but the authors might want to check that.

Page 3 Line 86 -88 "Muckenhuber … Sentinel-1 data" I suggest adding at the end of the sentence something like: "as a frontend to the ORB algorithm from Rublee included in the OpenCV package" just to give the reader an impression of the used technique, but I might be mistaken.

Page 5 Line 139 "HV polarization" – given that most of the Arctic and the whole Antarctic is only covered by HH data this limits the conclusions for the application of the presented algorithm to the European Part of the Arctic and the Baltic sea.

Page 5 Line 165 "Nansat … gdal.org)." It reads like a commercial – may be the authors want to give this line a second thought.

Page 9, Line 215 I suggest to add :"Given a suitable threshold [and unique features]…"

Page 9, Line 217 I suggest to change: "Muckenhuber … found the most suitable …" to "Muckenhuber … found a suitable …" given the flexibility of your approach and your suggested parameters

Page 10 Line 260-261: I suggest adding something like: "The quality of this "first guess", however depends on the density of the feature vector field and the local ice conditions"

---

## Referee Report (RR2)

**Introduction**

Thank you for your detailed reply to my comments and the significant improvements in your paper. It is nice to see this tempting idea evolving and over all the paper is now worth being publishing after some more revisions to be done. The only fundamental problem I see is the parallel publication on the same topic from two other authors (actually two research group leaders!) from the same institute but this should not be a problem of an individual PhD student especially since the this respective discussion paper by Muckenhuber and Sandven has been published earlier and the associated code has been made publically available. At the end, it is a decision only the editorial board of the Cryosphere can make.

This having said, I'm afraid that I have still some comments. During the last revisions I asked to be more careful regarding some claims by asking for "softer" phrases and adding a more detailed analysis. The three main claims of the paper are:

1. The authors use feature tracking for a first guess to be computational more efficient than pure pattern matching
2. The first guess is based on independent vectors – therefore the authors are more capable to resolve deformation zones
3. HV polarization is better for feature tracking and therefore potentially as well for pattern matching
4. The algorithm can handle rotation

It is obvious that the setup of the algorithm allows a better handling of rotation but it is less obvious and not shown in the paper, that the algorithm is actually faster than pure pattern matching algorithms, that they can handle deformations and shear regions in a better way or that the use of HV data is more advantageous for pattern matching as well (see as well my thought in the last review which the authors included in the paper as well). I won't say that these claims are not potentially true but the authors neither show nor prove it. It's merely a question of the wording and no extra science that needs to be done but the first three claims are not as obvious as the authors suggest. (e. g. it might be better to say that the first guess by feature tracking potentially allows a computational efficient computation than claiming that it is more efficient that pure pattern matching. If you want to claim it: prove it!). It might not sound that sexy but it would be more precise. The same holds true for claim 2 and 3.

One last thing before I deal with the comments in detail: Personally I think that the authors stress the computational efficiency of their algorithm a bit too much and avoid every operation that could hamper this goal. That's okay but it should be discussed in the text (e.g. "a better way to determine d would be to take into account the variability of the vectors as well we avoided it for the sake of computational efficiency") In this way the readers would be aware of these limitations. Just scan through your last answers to my comments for the term computational efficiency and add some explanation at the respective position in the

manuscript. Besides that the DTU shows that operational products a possible with pure pattern matching as well.

**Comments on Response to Referee #1**

The authors answered most of my questions and solved many of the problems I mentioned. Thanks for that. In the following I will focus on the points only, where I do not agree with the answers which thereby might sound more negative than actually meant.

Page 1 Line 14ff the authors are of course right that the noise level depends on the incident angle. However, there is a pattern of the noise level (e.g. https://doi.org/10.1016/j.procs.2016.09.247 and ESA mentions -25 dB and -22 dB). But if the authors basically use a fixed threshold for the scaling of all scenes, I would like to ask for a reference that suggests doing so or containing information about the variation of noise level over the swath width. It might hold true for some scenes but definitely not for all and would lead to a scaling over a value range containing only e.g. 50 % meaning full values or throw away 50 % of values containing information in the case the authors showed and used to make the point that there approach is correct . But if the distribution has so many values in in the range below the noise level it is at least worth checking if the calibration routines of Nansat work properly (the scaling itself provides  somehow has to provide a reasonable image otherwise, there would not be any data.

Page 2 Line 40 "… an even better parameter set, because a higher number Is considered …" It is not the same, it is a completely different approach and therefore not better but just different, providing different information but I'm okay with it.

Page 2 Line 50 "… more feature tracking vectors are found on image pair with a smaller time span" – given a sufficient number of linear features visible in HV, only and not in the less textures areas where the feature tracking algorithm tends to fail I would like to add but as I said, I guess, that's okay.

Page 3 Line 80: I understand the point of the authors and agree with them! Having said this, I'm not happy with it and don't think that things like this should happen since it is bad science and from an outside perspective creates the impression that NERSC has more expertise in the field of sea ice drift than there is actually available plus increases the noise level for publications in this field in general, which makes it more difficult to keep an overview of relevant publications. However, as I stated in my introduction it is up to the editorial board of the Cryosphere to decide.

Page 4 Line 93: The mentioned independence is not only reduced by overlapping / close features but as well by very characteristic and bright features that might be represented in all resolution levels of the ORB algorithm but I guess that's okay

Page 4 Line 105: The feature tracking algorithm chooses the best detectable features automatically but this means that there can also  be vast areas with no feature tracking

vectors at all if there are enough well defined ridges is one corner of the scene. For the first guess the motion field is now interpolated over these vast "vector"-less areas as well. I have my doubts if this is better or more independent than a fine tuned resolution pyramid in the case of pattern matching. I suggest skipping this comparative statement

Page 5 Line 134 f It is not necessarily true that a scaling that favours feature tracking favours pattern matching as well. 1. The NCC needs no scaling at all since it is normalized (the N of NCC) and 2. Imagine an extreme scaling that would separate ridges in white from the surrounding background in black. For feature tracking, this would be perfect, since it would highlight the linear feature. For pattern matching it would be a problem, especially in the regions outside the ridged areas since it is based on areal pattern variation.

Page 8 Line 242 Thanks for the explanation. Just mention it in the manuscript

Page 9 Line 291 "We believe … " I suggest to show it or be more careful with such a statement. The ice drift of the e.g. for example is pure pattern matching without any fancy resolution pyramid and runs pretty fast to provide the ice drift within the framework of CMEMS.

Page 12 Line 409 / Page 13 Line 437 the authors promise to bring feature tracking performance of HH closer to HV by an increased coverage of S1 images and a better pre-processing. I would claim that HH and HV show different characteristics and that's it. You won't change the stronger dependency of linear features in HH polarised images from the incident angle by increasing the temporal resolution but may see other features. Having said this, a better pre-processing and a higher temporal resolution never hurts and will improve the results for sure.

**The Manuscript itself**

Most of the remaining points I already handled in my answers to your comments, meaning there is not much left.

**Still open points from previous reviews**

Page 4 Line 119 f – see as well my comment above: the independence is difficult to assess and its effect on resolving deformation zones is not shown in this paper – may be more careful

Page 7 Line 210 – 215 See my comment to your answers

Page 16 Line 393 I think we had this discussion already in the first draft. I think it is difficult to use a buoy dataset for validation if you used the same dataset beforehand to optimise the setup of the algorithm (Page 16 Line 370-377).

It does not result in a real assessment of the performance of the algorithm but in the potentially performance in the case that the algorithm is tuned in the right way to the respective ice conditions.

**Technical comments**

Page 3 Line 62: "On the one hand… on the other hand" – there is "the one hand" missing

Page 4 Line 105 since you mention in line 104 that the area has a revisit time of less than one day it is obvious that your area of interest is monitored on a daily basis

Page 13 Line 324 "accurate drift" – it not necessary a more accurate drift but merely a more localised/ locally tuned/ locally adapted one. The term "accurate" sounds good but has a meaning which has nothing to do with the case at hand – nevertheless we might hope that a locally adapted drift vector represents the drift at a certain position more accurately than the first guess but that is not necessarily the case.

Page 15 Lin 363 "since the uncertainty increases with distance d (Figure 7)" – may be rephrase the sentence – the authors are correct but it suggests that Figure 7 shows the increase of "uncertainty" depending on the distance. Having said this: "uncertainty" is a statistical term – may be choose something less well defined.

Page 15 Line 368 / Page 17 Line 405 – it is more or less the same sentence, may be skip one

Page 21 Line 482 "accuracy of the introduced algorithm" I suggest to add "for given image pairs, given ice conditions, given region and given time" since the results are not transferable to other ice conditions or regions…

I hope my comments help you improving your paper and I'm sorry for the spot of work I left you with.   It is an interesting topic and I'm really looking forward to the final paper!

---

## Author Response (AR2)

Manuscript prepared for J. Name
with version 2015/09/17 7.94 Copernicus papers of the LaTeX class copernicus.cls.
Date: 2 May 2017

**Response to Editor**

**'Open-source sea ice drift algorithm for Sentinel-1 SAR imagery using a combination of feature-tracking and pattern-matching'**

Stefan Muckenhuber and Stein Sandven

Nansen Environmental and Remote Sensing Center (NERSC), Thormøhlensgate 47,
Bergen, Norway

*Correspondence to:* S. Muckenhuber (stefan.muckenhuber@nersc.no)

Dear Professor Lars Kaleschke,

Thank you very much for helping us improving our paper.

Please find here the **answers** to your comments:

## 1 Comment from the editor

The manuscript requires further major revisions before it can be accepted. In addition to the comments raised by the referees I'd like to add that some figures need to be improved, e.g. the black vectors in Fig. 3 are largely overlapping and Figure 8 looks really strange.

**We changed several parts of the manuscript, adjusted the algorithm, recomputed the pa-**
**rameter evaluation and validation with the new algorithm version and included an evaluation of the algorithm performance for HH polarisation.**
**We changed the vector spacing in Figure 3 to improve the visibility of the individual vectors and replaced the vectors in Figure 8 by dots to get a better impression of the buoy data locations. In addition, we included information about the retrieved cross-correlation values in**
**Figure 3.**
**We hope that the new version will meet the high standards of The Cryosphere.**

Please find attached the corrected manuscript with changes marked in blue and red.

Thanks again for your comments. We are looking forward to your reply!

Best regards,
S. Muckenhuber and S. Sandven

Manuscript prepared for J. Name
with version 2015/09/17 7.94 Copernicus papers of the LaTeX class copernicus.cls.
Date: 2 May 2017

**Response to Referee # 1**
**'Open-source sea ice drift algorithm for Sentinel-1 SAR imagery using a combination of feature-tracking and pattern-matching'**

Stefan Muckenhuber and Stein Sandven

Nansen Environmental and Remote Sensing Center (NERSC), Thormøhlensgate 47,
Bergen, Norway

*Correspondence to:* S. Muckenhuber (stefan.muckenhuber@nersc.no)

Dear Referee # 1,

Thank you very much for helping us improving our paper.

Please find here the **answers** to your comments and the corresponding ***changes in manuscript***:

**5   1   Comments to the answers to Referee 1**

Page 2 Line 42-46: The problem with the backscatter values is not solved by correcting a typo and claiming that the values were shown in Bell (a very uncommon unit for backscatter by the way). The correction of the typo shifts the backscatter values into the right magnitude. However, given a noise level of 22 dB for Sentinel-1 I would still like to question the calibration routines used. Using scaling suggested by the authors on correctly calibrated S1 data would mean to focus on a value range mainly containing noise (-32.5 dB - -22 dB) and only partly containing potentially meaningful backscatter values in the range -22 dB to -18.86 dB. The authors might want to check again there calibration routines.

**The way we understood the Sentinel-1 data is that -22 dB represents the maximum NESZ**

**(Noise Equivalent Sigma Zero) and the NESZ depends on the incidence angle. The maximum NESZ is found at the lowest incidence angle, but the NESZ values decrease with increasing incidence angle, which means that values below -22 dB might contain useful information. (Otherwise around half of the pixels in Figure 1 would be below the noise level.) The algorithm is mainly sensitive to changes of the upper brightness boundary and less sensitive to the**

**lower brightness boundary (see Muckenhuber et al. 2016). Muckenhuber et al. 2016 tuned the algorithm to find suitable brightness boundaries and we converted the corresponding backscatter values to dB. The lower boundary in Muckenhuber et al. 2016 was set to 0. Since it is impossible to convert 0 to dB, we looked into the backscatter distributions of images**

**over sea ice (e.g. Figure 1) and chose a value that includes the majority of the measured dB**

**values. This means that our lower dB threshold is set to an even higher value than used in Muckenhuber et al. 2016.**

Page 8 Line 245: the use of logarithmic scaling is the standard for all SAR based sea ice research and professional monitoring by universities, ice services and research institutes. But it is good to see that this promising approach uses dB scaling now as well.

**We agree and just wanted to mention why we did not apply the linear pre-processing as suggested by Muckenhuber et al. 2016 for the feature-tracking approach.**

Page 8 Line 255: I did not want to vote against parameter tuning or the validation using displace- ment vectors derived from satellite image pairs. Both instruments are extremely useful. I just thought that a validation using the same dataset originally used for tuning is not the best approach for an independent validation. It is a pity that the authors decided to skip both parts completely but then it is for sure the most effective approach to deal with my remark.

**We agree that comparison against manually derived drift vectors represent an important**

**tool for parameter tuning, but we hope that the new approach delivers an even better parameter set, because a higher number of satellite image pairs is now considered.**

Page 9 Line 278 - 301 I like this new paragraph you added into section 5, but as far as I can say it does not consider the specific problem, I outlined in my question and they suggest to handle with this paragraph (the same search window size means different degrees of freedom depending on the time span between the two considered image acquisitions) - It does however discuss the general aspect of time span regarding the 'hard' 0.5 m/s maximum speed threshold but I guess, that's o.k.

**It is true that the same search window size for short and long time spans between acquisitions, results in different degrees of freedom. Considering short time spans allows for a higher**

**velocity adjustment in the pattern-matching step. However, more feature-tracking vectors are usually found on image pairs with shorter time spans, which reduces the search window sizes, and acts to a certain extent against the higher degree of freedom. Adjusting the search window according to the time span would add additional complexity to both the algorithm and the parameter tuning and needs more research on how the search window should be adjusted**

**depending on the time span. We aim for a simple algorithm both for computational efficiency and easy open-source distribution to users and therefore hope that the simple approach to remove drift vectors above the maximum speed is sufficient to satisfy the users need for now. Our current and future efforts include the investigation of ice drift behaviour on different time scales to combine drift information from image pairs with different time spans. This work will**

**hopefully help us to improve our understanding on how to adjust the pattern-matching search**

**window in the most meaningful way.**

**2    The article itself**

**2.1    General comments**

The article improved a lot since the last time and the authors handled the comments of the reviewers in a very efficient manner. It reads like a completely new manuscript. Personally I think that the authors tend to use superlatives like 'optimize' or 'most meaningful' a bit too often given the flexibility of their approach (various changes between both versions) but nevertheless it seems to be on a right track now and worth being published in 'The Cryosphere' after major revisions. Within the frame of the revision it would however be necessary to discuss the differences of this paper to the recently published paper by Anton Korosov and Pierre Rampal for the same institute on the same topic: A.Korosov and P. Rampal (2017): 'A Combination of Feature Tracking and Pattern Matching with Optimal Parametrization for Sea Ice Drift Retrieval from SAR Data', Remote Sens. 2017, 9(3), 258; doi:10.3390/rs9030258, which looks quite similar to me. Personally, I think that this par- allel publishing from the same institute is at least a bit unfortunate (even though I'm aware, that the manuscript at hand was originally submitted earlier than the now published article by Korosov and Rampal).

**We replaced 'optimised' with *adjusted* and removed 'most meaningful' throughout the manuscript.**

**Regarding the recent publication from Korosov and Rampal 2017: The open-source character of the review process in TC and the open-source distribution of the presented algorithm make it impossible to compete with journals that have a review process of approximately 31 days plus 7 days until acceptance to publication (median values for papers published in Remote Sensing in 2016 from http://www.mdpi.com/journal/remotesensing). Since the presented**

**manuscript was earlier submitted and due to the fact that our manuscript and algorithm was already distributed in TCD and on github more than 2 months before the submission of Korosov and Rampal 2017, we would like to disregard the paper from Korosov and Rampal 2017 in our manuscript.**

### 2.2    Specific comments

Page 4 Line 113-115 I'm not sure how much of the vector independence from the feature tracking actually exists, given the overlap of features from the various resolution levels, the filtering and polynomial least square fitting to identify outliers and the linear interpolation to get first guess estimates for the subsequent pattern matching. What do the authors think?

**In this sentence, we refer to the feature-tracking vectors as such (blue in Figure 3), without considering filtering, fitting etc. The mentioned independence here is only reduced by overlapping, i.e. in the case of very close features. All other feature-tracking vectors can be seen completely independent from each other. We try to preserve this independence as good as possible during filtering and interpolation. The filter parameters were set according to our**

**experience with the algorithm performance and visual interpretation of several representative image pairs. We tried to allow for the largest possible degree of freedom without including too many erroneous vectors. The first guess is constructed by triangulating the input data and performing linear barycentric interpolation on each triangle. Hence, the first guess is only affected by the three closest vectors and reveals therefore a relatively high degree of**

**independence compared to methods that apply e.g. a pattern-matching resolution pyramid. The interpolation method was chosen in order to allow for as much independence as possible in the first guess. Our next goal is to test the algorithm on large datasets to identify potential weaknesses in areas with strong deformation, shear zones etc. and this work will provide further information on the independence of the final drift vectors.**

Page 6 Line 173-192 I would like to repeat my statement from the last review: there seems to be something wrong with your calibration routine and it is not only a typo. It would be great for sea ice research if S1 had a sensitivity of -32.5 of even -25 dB only but its noise level is at -22 dB, which means that especially your lower boundaries for the scaling cause a problem for correctly calibrated

S1 images.

**See the first comment in Comments to the answers to Referre 1.**

Page 6 Line 186 The scaling to 8 bit / 256 grey level is an input requirement for feature tracking using the ORB algorithm but reduces the signal variation. This reduces the effectiveness of the

NCC which can handle any numerical precision you like. It is not a problem for medium to larger resolutions (like CMES service provided by the DTU) since there are still enough pixel in each correlation window but as soon as you go to higher resolutions, even slight variations in the signal become more important (since there might be not so many clear variations left in a smaller window). Have the authors tried to use the original dB backscatter values for the pattern matching instead of the scaled 256 greyscale image?

**We thought about applying another scaling for the pattern-matching step, but decided to use the same scaling as used for feature-tracking for the following three reasons:**

1. **Applying an additional conversion from the digital numbers (provided in the Sentinel-1 file) into dB values in a 32 bit floating point format increases the computational effort of**

**the pre-processing step.**

2. **The utilised pattern-matching python package can handle both 8-bit and 32-bit floating-point format. However, using 32-bit increases the computational effort of the pattern-matching step.**

3. **The scaling that is currently applied was tuned to provide a high number of feature-tracking vectors. We assume that a scaling that favours the recognition of features is also favourable for the applied pattern-matching procedure.**

Page 9, Line 223-224 As I mentioned before, the use of logarithmic scaling for SAR is common and I'm actually surprised that the authors did not do it from the start.

**We agree that logarithmic scaling represents the common procedure for SAR. We mention the procedure in this line to address the differences to the approach from Muckenhuber et al. 2016.**

II Filters: I have various questions:

1. For the polynomial fit the authors use all vectors (those from coarser resolution levels and those from finer resolution levels - a single pixel on the coarser resolution level covers about 3.6 pixel on the finest resolution level: subpixel uncertainty of +/-145 m vs a subpixel uncertainty of +/- 40 m?) - isn't this a bit of a problem if you merge it all in one polynomial surface and then try calculate if the predicted vector is 100 px away from your polynomial surface?

**Since we down-sample the image by a factor of two, the pixels have a size of 80 m and 287 m for the highest and lowest resolution level. The resolution levels are only considered inside the ORB python package and retrieving the information from which resolution level the vector is derived, is unfortunately not implemented at the moment. A work around would be to downsample the satellite image with Nansat and apply the feature-tracking algorithm individually for each resolution pyramid. However, that would increase the computational effort significantly and the vectors from the different resolution levels would not be linked and compared to each other in terms of Harris corner measure etc. This means that we have to treat all vectors equally regardless of the resolution level (since we don't know the resolution level). The threshold of 100 pixels in this context refers to 8 km away from the simulated starting point, since we work with a resolution of 80 m outside the feature-tracking procedure.**

2. You are applying a least square fit to an angle and calculate a polynomial surface from it. How do you handle the zero-crossing problem and all the other problems you are facing once you start averaging and interpolating angle values?

**We agree that the issue of zero-crossing during filter and inter-/extrapolation of the angle values needs to be addressed. We added a function that centres the feature-tracking rotations around $180°$ before filter and inter-/extrapolation and we remove the adjustment after filter**

**and inter-/extrapolation has been applied. The following has been added to Section 3.2 in II Filter:**

*To avoid zero-crossing issues during the following filter and inter-/extrapolation process (in case the image rotation $\delta$ between $SAR_1$ and $SAR_2$ is close to $0°$), a factor $|180 - \delta|$ is added to the raw rotation values $\alpha_{raw\,f}$ using the following Equation:*

$$\alpha_f = \begin{cases} \alpha_{raw\,f} + |180 - \delta| & \textit{if } \alpha_{raw\,f} + |180 - \delta| < 360 \\ \alpha_{raw\,f} + |180 - \delta| - 360 & \textit{if } \alpha_{raw\,f} + |180 - \delta| > 360 \end{cases} \tag{1}$$

*This centres the reasonable rotation values in the proximity of $180°$. After applying the filter and inter-/extrapolation process, the estimated rotation $\alpha$ is corrected by subtracting $|180 - \delta|$.*

**The following has been added to Section 3.2 in III First Guess:**

*As mentioned above, the rotation estimates $\alpha$ are now corrected for the adjustment applied in Equation 3, by subtracting $|180 - \delta|$.*

3. It seems that your angular polynomial surface has a peak at about $130°$. Just for curiosity: Is this the rotation between both images?

**In this case, the rotation between the two images is $129.08°$ and the mean feature rotation is $132.24°$. The peak of the rotation distribution is usually close to the image rotation.**

Page 11 Line 272-273: least square fit and linear function for angles? How do the authors handle the specific problems related with this (see my comments on II Filters)

**We agree, that this issue needs to be addressed. See answer above to second comment under II Filters.**

Page 11 Line 285-287: 'The uncertainty ... vector' I would claim that it is not the uncertainty of the estimate but the representativeness of the estimate and that it does not only depends on the distance to the next feature tracking based vector but as well on the heterogeneity of the other surrounding feature tracking based drift vectors.

**We agree that the distance $d$ indicates the representativeness rather than the uncertainty and changed the sentence to:**

*The representativeness of this estimate however, depends on the distance $d$ to the closest feature-tracking vector.*

**We consider only the distance and not the heterogeneity of the surrounding feature-tracking vectors to be as computational efficient as possible. Our first attempt to estimate the drift at any given location on $SAR_1$, was to apply a search mechanism to identify the surrounding feature-tracking vectors. This procedure would allow to derive a search window size based on the heterogeneity of the surrounding vectors. However, this approach was computationally**

**very inefficient. Our current method to derive a matrix that includes the distances to the closest location of a feature-tracking vector and the applied extra-/interpolation procedure are**

**implemented in a computational efficient manner in the python toolbox SciPy. This allows us to derive drift estimates for the entire scene in a few minutes. The computational efficiency of the algorithm is one of our main goals, since we want to derive sea ice drift from large datasets and prepare for an operational product.**

Page 12 Line 307-308: As far as I know the idea to restrict the smaller correlation template to a circle has been strongly propagated by Roberto Saldo from the DTU who uses it for his drift algorithm. I'm not sure if this is what the authors mean. While I'm aware of the advantages to limit the smaller correlation template to a circle, I have no idea why you would want to restrict your search window to a circle (if not only for the reason to safe computation) but may be the authors want to outline the advantages of a circled search window while having a rectangular correlation template. If they meant to describe a circled correlation window the authors want to refer to Roberto Saldo (if there is no publication regarding this matter may be as personal communication).

**Here we refer to the larger search window $t_2$ on $SAR_2$ and not to the small template $t_1$ on $SAR_1$ around the start position of the vector.**

**From the public discussion, we understood that Referee #2 wanted circular templates for both the small and large template. See comment:**

**'P7 Fig.2 (and text): Why rectangular/square templates has been used? A circular template would be much easier (symmetric) to rotate. Consider using a circular templates instead.'**

**And our answer from 13.02.2017:**

**'We agree with this comment. Regarding $t_1$ however, the current version of the used OpenCV function matchTemplate does not allow circular templates and work-arounds would influence the result and the computational efficiency. We hope that a later version of matchTemplate will allow to use masks. Regarding $t_2$, we included a circular mask for the matching result to limit the search area to a circle rather than a square.'**

**To us, it seems more reasonable to search in a circular area rather than a square. This limits the distance from the first guess to a constant value, rather than to an arbitrary value depending on the looking angle of the satellite. This should also be beneficial when searching for a suitable size for the search window. If there is a correct match in the corner of a square during parameter tuning, a size for $t_2$ might be recommended that is too small considering**

**another looking angle of the satellite.**

**We added the following sentence:**

*This limits the distance from the first guess to a constant value, rather than to an arbitrary value depending on the looking angle of the satellite.*

Page 13 Line 318-319: 'In order to be consistent with the resolution of the feature tracking'. What is the advantage of being consistent with the resolution of the feature tracking?

**Sea ice drift might be different on different resolution scales. This is particularly an issue in the case of rotation. The feature-tracking vectors provide the first guess and this vector field should represent the same drift resolution as considered by the pattern-matching step.**

**Also, during the tuning of the feature-tracking algorithm, we found the considered template size to be a good compromise between high resolution and capturing enough information for a reliable recognition performance. We assume that this also means that the template size represents a good choice for the pattern-matching step.**

Page 14 Line 344-352 A higher MCC value seems to indicate a higher chance of a better match for your algorithm compared to buoy data. This is an interesting finding since most pattern matching algorithms have lower MCC_min values or none at all and since it was shown, that there is no direct relation between error and correlation coefficient.

This could mean that the author's initial guess and search strategy are less effective than those of pure pattern matching algorithms (and thereby needing a higher correlation to find a suitable match) especially since the maximum shift from the feature tracking based first guess is 1.6 (d=10) to 16 km (d=100) at maximum. Every difference above that is a hint that the first guess did not work or that the buoy drift has nothing to do with the surrounding ice conditions (Figure 11 a). Additionally I suggest checking the spatial distribution of the vectors the authors rejected due to their low MCC and high spatial distance (e.g. a shearing zone within the correlation window would for example cause as well a drop in the correlation value and potentially a larger difference in displacement relatively to an originally close drift buoy in another drift regime) and study the behavior compared to manually tracked vectors from the image pair ensure that the difference comes really from the algorithm and not from the limitations of the comparison with buoys, the authors are well aware of (page 20 line

439 - 450).

**It is correct, that the drift vectors with higher cross correlation provided by our algorithm match better with the buoy drift than the vectors with low MCC values.**

**We believe that our initial guess provides a reliable estimate for the following reasons:**

**Initial guess in the proximity of the feature-tracking vectors: The error of the feature-tracking**

**vectors is in the order of a few 100 m compared to manually derived vectors, as shown by Muckenhuber et al. 2016. This error does not prohibit the pattern-matching step to search at the correct location.**

**Initial guess further away from feature-tracking vectors: Every distance $D$ above 16 km indicates that the first guess was not close enough to the buoy position, to allow the pattern-**

**matching to search for a match at the buoy location. However, only very few points are above this limit and the buoy location does not necessarily represent the same ice drift as**

the algorithm. As shown in Section 4.1 'Search restrictions evaluation', increasing the search window and allowing up to 200 pixels deviation from the first guess, did not provide many additional vectors with a distance $D$ below 1000 m. Therefore we believe that the search window restrictions are suitable for our area and time period of interest.

Increasing the search window can also increase the error potential, in case the initial guess is correct. E.g. in very homogeneous ice conditions the pattern-matching could fail and accept a random match. In this case, the search window restricts the deviation from the first guess and hereby the error potential.

There is certainly a small remaining probability that the first guess is far away from the correct location and the search window is too small to include the correct location. We tried to find a good compromise between minimising this probability and being computational efficient enough to perform high resolution drift analysis within a few minutes.

We believe that the presented approach outperforms pure pattern-matching procedures in terms of computational effort while, at the same time, allowing for higher drift and rotation deviation in the drift field. A simple pattern-matching approach with no restrictions is certainly not computational efficient to provide high-resolution ice drift from a large satellite database. Therefore, pure pattern-matching approaches usually apply a resolution pyramid to limit the computational power, which means that high resolution estimates are based on low resolution drift. This, in combination with additional filtering and smoothing that is usually applied to discard outliers, decreases the independence of the high-resolution drift vectors significantly.

The initial guess of our approach builds on feature-tracking vectors that may point in any direction regardless of the neighbouring vector and may include any rotation from 0 to $360°$. During the filter and inter-/extrapolation process, we try to preserve the high degree of independence to allow for relatively strong deviation in terms of drift and rotation.

We did not reject matches according to high spatial distance, but only based on the MCC value.

We changed Figure 8 to illustrate the spatial distribution of accepted and rejected vectors and added the following sentence to the caption:

*Green and blue colour indicates start locations (on $SAR_1$) to which the algorithm provided vectors with a $MCC$ value above and below $0.4$ using (a) HV and (b) HH polarisation.*

As shown in the first draft of the manuscript, we compared the algorithm results with manually derived drift vectors and found that the results compared well. According to the previous comments, we removed the comparison with the manually derived drift vectors and decided to use the buoy data also for parameter evaluation to increase the size of the dataset and cover a larger range of ice conditions, time spans etc.

As stated in the discussion, the recommended parameters are not meant as a fixed setting,

**but should rather give a suggestion and guideline to estimate the expected results and the corresponding computational effort. They can easily be varied in the algorithm setup and should be chosen according to the needs of the user. To provide an improved parameter setting depending on ice conditions, area of interest etc. and further test and adjust the initial guess procedure, the next step of our research is to apply the algorithm on large datasets.**

Page 15 3.3 Comparison with buoy data: Just for my understanding: Did the authors center a 34x34 pixel window at the original buoy position? The original buoy position has been acquired every hour or has been interpolate to hourly intervals

**Yes, a $34 \times 34$ pixel window was centred around the buoy start position. Based the conversation with Polona Itkin and Gunnar Spreen, most buoys acquired the GPS position every hour, but some every three hours. In the later case, the positions were interpolated to every hour by NPI.**

Page 15 Line 356 'Each pair yielded more than ... three days' I'm not sure but it would probably more interesting to mention the number of vectors, which the recent version of the feature tracking component provides, using HV only and dB scaling, if at all ...

**We used the recent version of our feature-tracking algorithm and changed the sentence to:** *Each pair yielded more than 300 drift vectors applying the feature-tracking algorithm from Section 3.2 and had a time difference between the two acquisitions of less than three days.*

Page 17 Figure 9 and 10: Based on the Figures it is difficult to see an advantage for the variation of the search window size or the rotation angle aside from computational efficiency. Would the authors agree?

**The computational efficiency is an important argument for the variation of the restrictions (search window and rotation angle). In addition, several red dots which where closer than $100$ pixels ($d < 100$ pixels) are excluded by applying the recommended restrictions. Apparently, a large pattern-matching adjustment close to a feature-tracking vector increases the possibility of an error (represented by $D > 1000$ m). We assume that in these cases, the initial guess provided already a good estimate and a large search window and rotation range rather increases the error potential.**

Page 18 Fig 11 a: something seems to be wrong with the legend
**We adjusted the legends in Figure 11 according to this comment.**

Page 18 Fig 11 b: how would the fit look like if the authors would include the values you rejected due to their low correlation value?

**We included a logarithmic normal distribution for all results and changed the sentence in the caption to the following:**

*Logarithmic histogram of distance $D$ with 100 bins between 10 m and $10^5$ m including two logarithmic normal distributions that were fitted to all results (grey) and to the filtered results with $MCC > 0.4$ (solid red line).*

Page 18 Line 407 / Page 20 Line 441 'median and 341.9 m' What's the significance? Obviously it means that 50% of your data has an error of less than 340 m with respect to the buoy data but it means as well that 50% have an error, that is larger (and based on the distribution: 1% < 20m, 5% < 43m, 10% < 68m, 68% (1 sigma criterion) < 620 m, 95% (2 sigma criterion) < 2700 m and 99% (3 sigma criterion) < 6400m). But could the authors give a hint what it could mean regarding the quality of the algorithm, given the various influences they identified which could have biased the result? May be a solution would again be to use the manually derived vectors from the original draft for it to account for the algorithm 'accuracy' only (well with an 'uncertainty' hidden in the manually collected drift vectors of course)?

**According to the comment of Referee #2 during the first review process, 'the validation with this data set (NB: the manually derived vectors) is not fair as the algorithm has been tuned for this data. Then only the buoy data can be used for independent validation.' Therefore, we had to excluded the comparison with the manually derived vectors. In addition, the buoy dataset represents a larger dataset, accounts for a larger range of ice conditions and time spans and is therefore more representative for our area and time period of interest, despite its limitations.**

Page 20 Line 458-460 'The parameters can ... strong rotation).' I have a fair idea which parameters to vary to meet changes in computational power, area of interest and expected ice conditions but I have no idea why availability of time, computational power and number of image pairs are mentioned individually since the only parameter behind it is computation time or rather the selection of a finer or coarser resolution. Additionally I would be interested what parameter to change to influence the accuracy!

**We meant resolution instead of accuracy and changed the sentence to the following:**

*The parameters can easily be varied in the algorithm setup and should be chosen according to availability of computational power, needed resolution, area of interest and expected ice conditions (e.g. strong rotation).*

Page 20 Line 470-484 I guess this paragraph is based on information you received from Nuno Miranda. It would be useful to refer to it as Personal Communication in the text since otherwise the readers might wonder where this information comes from.

**Yes, this paragraph is based on a personal communication with Nuno Miranda and we added the following to the end of the paragraph:**

*(Personal Communication with Nuno Miranda, January 2017)*

Page 21 Line 487-490 'Our next step ... less than 5 km' That's really nice! I'm really looking forward to such a dataset with a good quality. However, given that most of the S1-data in the Arctic and Antarctic is in HH polarization (in Antarctica completely) wouldn't it been better to perform this study in HH from the beginning, even if there seems to be an advantage for HV in the feature tracking?

**So far, we found good coverage in HV for our area and time period of interest. However, it is true that some areas are currently only monitored in HH and we included an algorithm performance evaluation also for HH polarisation.**

**We changed the sentence to:**

*Our next step is to embed the algorithm into a super-computing facility to further test the performance in different regions, time periods and ice conditions and evaluate and combine the results of different polarisation modes.*

**We aim to eventually combine drift information from the two channels in the most meaningful way. This is also described further down in Section 5 'Discussion and outlook'. We expect that**

**the number of feature-tracking vectors derived from HH will increase in the future due to an increased coverage provided by Sentinel-1A and Sentinel-1B (a shorter time span between the acquisitions is favourable for HH as shown by Muckenhuber et al. 2016) and a better pre-processing system that we will be able to access through our ongoing cooperation with TU Wien. This will bring the feature-tracking performance of HH closer to the performance of**

**HV and therefore the same (or a very similar) parameter setting is expected to be favourable for HH.**

Page 21 Line 503 'better coverage in HH pol' - that's nicely put - the S1 observation scenario does not plan any HV acquisition Antarctic part of the southern ocean and none in the central Arctic ocean but it could of be a reason to demonstrate the need for HV data in these regions as well.

**We changed the sentence to:**

*We found our area of interest well covered with HV images, but other areas in the Arctic and Antarctic are currently only monitored in HH polarisation.*

**We agree, that the shown HV performance to track features could demonstrate the need for**

**HV data in regions that are currently only monitored in HH. However, before we suggest to monitor these regions with HV, we would like to compare the two channels more thoroughly and perform case studies in the respective regions. Based on the presented work we can only refer to our area and time period of interest, which is well covered with HV data. Testing the**

**algorithm on large HV and HH datasets will hopefully provide us with more information on**
**the necessity of HV monitoring for sea ice drift retrieval.**

Page 21 Line 507 - 510 'Therefore ...future work' A good idea! Just a thought I would like to add. Feature tracking needs linear features which tend to depend a bit more on incidence angle, orbit, and changes in ice condition in HH than in HV and depending on the robustness of your feature detector
it might even be quite robust to the quite strong noise in the S1-HV data (it is actually impressive how many feature tracking based vectors you were able to retrieve from the HV scene compared to the HH one). This could explain the better performance of the feature tracking part for the HV component. Cross correlation based pattern matching however is less sensitive to changes in linear feature and more sensitive to areal pattern changes which might potentially favor the HH channel -
but as I said: just a thought.

**We agree and added the following to Section 5:**

*Another argument is that the presented feature-tracking approach identifies and matches corners, which represent linear features. The linear features on HH images are more sensitive to changes in incidence angle, orbit and ice conditions than the linear features on HV images. This could*
*explain the better feature-tracking performance of the HV channel. However, pattern-matching is less affected by changing linear features and more sensitive to areal pattern changes. This could potentially mean that the HH channel performs better than HV when it comes to pattern-matching.*

**2.3  Technical comments**

Page 1 Line 9 'pre-processing of S1 data ... has been optimized' - Are the authors refereeing to their use of the logarithmic scale for the backscatter again? As far as I'm aware that is all you did and it is a bit of disappointing if the abstract promises an optimized preprocessing while the article offers the conversion to dB only.
**The sentence has been changed to:**
*The pre-processing of the Sentinel-1 data has been adjusted to retrieve a feature distribution that depends less on SAR backscatter peak values.*

Page 1 Line 10-13 I'm not sure if computational efficiency is necessary in the abstract, the authors
might want to give it a second thought but I'm fine if you want to keep it there.

**In case the 'computational efficiency' in Line 6-7 is addressed by this comment, we would like to keep the phrase. Our long-term goals are an operational sea ice drift product and to provide large drift datasets. Therefore the computational efficiency of the algorithm is a ma-**

**jor factor for us and one of the main arguments for the development of the presented approach.**

Introduction: I think the introduction improved a lot but especially of page 3 I miss something running like a common thread through it. It reads like an accumulation of independent facts.

**We changed the introduction according to this comment. The history and evolution of sea ice drift retrieval from SAR is to a certain extent an accumulation of independent facts, since**

**it includes the work of different people and institutes, but we tried to re-write the part in order to emphasize the connections.**

Page 3 Line 69 If I'm not mistaken, Hollands and Dierking (2011) implemented their own modified version of the algorithm and did not just continue the work on the algorithm but the authors might want to check that.

**We changed the sentence to:**

*Hollands and Dierking (2011) implemented their own modified version of this algorithm to derive sea ice drift from ENVISAT ASAR data.*

Page 3 Line 86 -88 'Muckenhuber ... Sentinel-1 data' I suggest adding at the end of the sentence something like: 'as a frontend to the ORB algorithm from Rublee included in the OpenCV package' just to give the reader an impression of the used technique, but I might be mistaken.

**We changed the sentence to:**

*This paper follows up the work from Muckenhuber et al. (2016), who published an open-source*

*feature-tracking algorithm to derive computationally efficient sea ice drift from Sentinel-1 data based on the open-source ORB algorithm from Rublee et al. (2011), that is included in the OpenCV Python package.*

Page 5 Line 139 'HV polarization' - given that most of the Arctic and the whole Antarctic is only covered by HH data this limits the conclusions for the application of the presented algorithm to the European Part of the Arctic and the Baltic sea.

**We changed the sentence to:**

*However, the focus of this paper is put on using HV polarisation (mainly acquired over the European Arctic and the Baltic sea), since this channel provides in our area of interest on average*

*four times more feature tracking vectors than HH (Muckenhuber et al., 2016), representing a better initial drift estimate for the combined algorithm.*

Page 5 Line 165 'Nansat ... gdal.org).' It reads like a commercial - may be the authors want to give this line a second thought.

**We changed the sentence to:**

*To process Sentinel-1 images within Python (extraction of backscatter values and corresponding geolocations, reprojection, resolution reduction etc.), we use the Python toolbox Nansat (Korosov et al., 2016), that builds on the Geospatial Data Abstraction Library (http://www.gdal.org).*

Page 9, Line 215 I suggest to add :'Given a suitable threshold [and unique features]...'
       **We agree and changed the sentence accordingly.**

       Page 9, Line 217 I suggest to change: 'Muckenhuber ... found the most suitable ...' to 'Muckenhuber ... found a suitable ...' given the flexibility of your approach and your suggested parameters
**We agree and changed the sentence accordingly.**

       Page 10 Line 260-261: I suggest adding something like: 'The quality of this 'first guess', however depends on the density of the feature vector field and the local ice conditions'
       **We added the following sentence after the first sentence in Section 'III First guess':**
*The quality of this 'first guess', however depends on the density of the feature-tracking vector field and the local ice conditions.*

       Please find attached the corrected manuscript with changes marked in blue and red.

Thanks again for your comments. We are looking forward to your reply!

       Best regards,
       S. Muckenhuber and S. Sandven

Manuscript prepared for J. Name
with version 2015/09/17 7.94 Copernicus papers of the LATEX class copernicus.cls.
Date: 2 May 2017

**Response to Referee # 2**

**'Open-source sea ice drift algorithm for Sentinel-1 SAR imagery using a combination of feature-tracking and pattern-matching'**

Stefan Muckenhuber and Stein Sandven

Nansen Environmental and Remote Sensing Center (NERSC), Thormøhlensgate 47,
Bergen, Norway

*Correspondence to:* S. Muckenhuber (stefan.muckenhuber@nersc.no)

Dear Referee # 2,

Thank you very much for helping us improving our paper.

Please find here the **answers** to your comments and the corresponding ***changes in manuscript***:

**Comments**

P13 (in the version with the changes indicated): 'normalized cross coefficient', I think this should be 'normalized cross-correlation coefficient' or just 'normalized cross-correlation', Check this throughout the manuscript, I think this appears more than once.

**We changed 'cross coefficient' to *cross-correlation* throughout the manuscript.**

Eq 11: One closing parenthesis missing "argmax(NCC(x,y)" -> agrmax(NCC(x,y)). Could also add (x,y) under argmax to indicate that argmax is with respect to (x,y).

**We changed Equation 11 to the following:**

$$(\frac{1+t_{s2}-t_{s1}}{2}+dx, \frac{1+t_{s2}-t_{s1}}{2}+dy) = \underset{x,y}{argmax}(\mathbf{NCC}(x,y)) \tag{1}$$

Please find attached the corrected manuscript with changes marked in blue and red.

Thanks again for your comments. We are looking forward to your reply!

Best regards,

S. Muckenhuber and S. Sandven

[revised manuscript text omitted]

---

## Author Response (AR3)

Manuscript prepared for J. Name
with version 2015/09/17 7.94 Copernicus papers of the LaTeX class copernicus.cls.
Date: 14 June 2017

**Response to Editor**

**'Open-source sea ice drift algorithm for Sentinel-1 SAR imagery using a combination of feature-tracking and pattern-matching'**

Stefan Muckenhuber and Stein Sandven

Nansen Environmental and Remote Sensing Center (NERSC), Thormøhlensgate 47, 5006 Bergen, Norway

*Correspondence to:* S. Muckenhuber (stefan.muckenhuber@nersc.no)

Dear Professor Lars Kaleschke,

Thank you very much for helping us improving our paper.

Please find here the **answers** to your comments:

**1  Comment from the editor**

The manuscript requires still further improvements before it can be accepted. In addition to the comments raised by the referee I'd like to stress that the quality of the presentation is still not yet satisfying. As already stated previously, the vectors in Fig. 3 are largely overlapping, Fig. 5 and 7 have no axis labels and the meaning and significance is not clear.

**To avoid the large vector overlaps in Figure 3 (now Figure 4), we zoomed into a smaller part of the image pair and depict the coverage of the corresponding zoomed area and of the entire image pair Fram Strait in an additional plot (Figure 1). We changed and added the captions according to this adjustment.**

**We added axis labels to Figure 5 and 7 (now Figure 6 and 8) and changed the captions accordingly. We believe that Figure 5 and 7 are essential to understand the algorithm and in particular to understand the python code. The two figures depict the matrices that are used in the algorithm code and show how the algorithm creates and handles the first guess as well as the distance $d$ distribution. We hope that by adding axis labels and the following sentences, the meaning and significance of the two figures will become more clear:**

*(this figure illustrates the matrices that the algorithm considers as first guess)*

*The figure depicts the matrix that the algorithm considers for the distribution of $d$.*

A section of "author contributions" is missing.

**We added the following section:**

25 *Author contributions. Stefan Muckenhuber designed the algorithm and the experiments, performed the data analysis and interpretation of the results and wrote the manuscript. Stein Sandven critically revised the work and gave important feedback for improvement. Stefan Muckenhuber and Stein Sandven approved the final version for publication.*

30 I further highly recommended to obtain a DOI for the code in the Github repository. Otherwise it is not clear to which version you refer to.

**Due to the recent rivalling publication, we do not dare to distribute the recent version of our code before the final publication of this manuscript. However, we will add the algorithm including an application example as supplement to this manuscript. This way, the DOI of the**
35 **manuscript will refer to both the manuscript and the code. We changed the last sentence in Section 'Open-source distribution' to:**

*The presented sea ice drift algorithm, including an application example, is distributed as open-source software as supplement to this manuscript.*

40 Please find attached the corrected manuscript with changes marked in blue and red.

Thanks again for your comments. We are looking forward to your reply!

Best regards,
45 S. Muckenhuber and S. Sandven

Manuscript prepared for J. Name
with version 2015/09/17 7.94 Copernicus papers of the LaTeX class copernicus.cls.
Date: 14 June 2017

**Response to Referee # 1**

**'Open-source sea ice drift algorithm for Sentinel-1 SAR imagery using a combination of feature-tracking and pattern-matching'**

Stefan Muckenhuber and Stein Sandven

Nansen Environmental and Remote Sensing Center (NERSC), Thormøhlensgate 47, 5006 Bergen, Norway

*Correspondence to:* S. Muckenhuber (stefan.muckenhuber@nersc.no)

Dear Referee # 1,

Thank you very much for helping us improving our paper.

Please find here the **answers** to your comments and the corresponding ***changes in manuscript***:

**1   Introduction**

Thank you for your detailed reply to my comments and the significant improvements in your paper. It is nice to see this tempting idea evolving and over all the paper is now worth being publishing after some more revisions to be done. The only fundamental problem I see is the parallel publication on the same topic from two other authors (actually two research group leaders!) from the same institute but this should not be a problem of an individual PhD student especially since the this respective discussion paper by Muckenhuber and Sandven has been published earlier and the associated code has been made publically available. At the end, it is a decision only the editorial board of the Cryosphere can make.

**We agree that this has been a very unfortunate development and we are happy that the editor acknowledges that our manuscript was submitted earlier and published first in The Cryosphere Discussion. In the future we will avoid the distribution of any unpublished material if not absolutely necessary.**

This having said, I'm afraid that I have still some comments. During the last revisions I asked to be more careful regarding some claims by asking for "softer" phrases and adding a more detailed analysis. The three main claims of the paper are:

1. The authors use feature tracking for a first guess to be computational more efficient than pure pattern matching

2. The first guess is based on independent vectors - therefore the authors are more capable to resolve deformation zones

3. HV polarization is better for feature tracking and therefore potentially as well for pattern matching

4. The algorithm can handle rotation

It is obvious that the setup of the algorithm allows a better handling of rotation but it is less obvious and not shown in the paper, that the algorithm is actually faster than pure pattern matching algorithms, that they can handle deformations and shear regions in a better way or that the use of HV data is more advantageous for pattern matching as well (see as well my thought in the last review which the authors included in the paper as well). I won't say that these claims are not potentially true but the authors neither show nor prove it. It's merely a question of the wording and no extra science that needs to be done but the first three claims are not as obvious as the authors suggest. (e.g. it might be better to say that the first guess by feature tracking potentially allows a computational efficient computation than claiming that it is more efficient that pure pattern matching. If you want to claim it: prove it!). It might not sound that sexy but it would be more precise. The same holds true for claim 2 and 3.

**We agree that there are certainly pattern-matching approaches that are computationally more efficient than the presented combined approach and we did not anticipate to claim that our feature-tracking algorithm is faster than all pattern-matching methods. We rather wanted to introduce a new approach that combines the advantages of the considered feature-tracking method and the considered (simple) pattern-matching method. We tried to rephrase the manuscript according to that.**

**We did not want to claim that our algorithm is more capable to resolve shear and deformation zones than all pattern-matching approaches, but rather wanted to mention that the independence of the feature-tracking vectors in terms of position, lengths, direction and rotation can potentially be an important advantage for resolving shear zones, rotation and divergence/convergence zones. We also mention that the feature-tracking approach might miss shear and convergence/divergence zones due to the uneven vector coverage. We rephrased the following sentence accordingly:**

*This can be done computationally efficient and the resulting vectors are often independent of their neighbours in terms of position, lengths, direction and rotation, which can potentially be an important advantage for resolving shear zones, rotation and divergence/convergence zones.*

**We believe that the considered feature-tracking approach performs better using HV polarisation. However, we agree that this does not mean that HV is the better approach for pattern-matching (or any other type of feature-tracking). It was not our intention to claim that. This was expressed in the Discussion with the sentence 'However, at this point, these are**

**just assumptions and will be addressed in more detail in our future work.' We also mentioned that HH could be the better channel for pattern-matching. We tried to further remove phrases that could lead to this misunderstanding and changed among others the following sentence:**
*And more consistent features could potentially also favour the performance of the pattern-matching step, but this is only an assumption and has not been tested yet.*

One last thing before I deal with the comments in detail: Personally I think that the authors stress the computational efficiency of their algorithm a bit too much and avoid every operation that could hamper this goal. That's okay but it should be discussed in the text (e.g. "a better way to determine d would be to take into account the variability of the vectors as well we avoided it for the sake of computational efficiency") In this way the readers would be aware of these limitations. Just scan through your last answers to my comments for the term computational efficiency and add some explanation at the respective position in the manuscript. Besides that the DTU shows that operational products a possible with pure pattern matching as well.

**We aim to deliver an open-source algorithm that is computationally efficient enough that a relatively high number of image pairs can be processed without the need for an expensive server or super-computing facility. This way a higher number of stackholder shall be reached and the possibility for computing sea ice drift on a standard PC shall be created for everyone. Therefore, the computational efficiency of the presented algorithm has a high priority for us. It is true that DTU provides an operational product based on pure pattern-matching. However the considered resolution is in the order of $10\,\mathrm{km}$ and the algorithm is not open source, which also means that we don't know the computational efficiency. We scanned through the last comments and added the following explanations:**
*For the sake of computational efficiency, the same intensity value scaling is used for the pattern-matching step.*
*For the sake of computational efficiency, the vectors from all resolution pyramid levels are treated equally.*
*(NB: the representativeness also depends on the variability of the surrounding vectors, but for the sake of computational efficiency, we only consider the distance $d$ as representativeness measure)*
*As an alternative, one could adjust the search window according to the time span. However, this would add additional complexity to both the algorithm and the parameter evaluation and needs more research on how the search window should be adjusted depending on the time span. For the sake of computational efficiency, we suggest the simple approach to remove final drift vectors above the maximum speed.*

**2 Comments on Response to Referee #1**

The authors answered most of my questions and solved many of the problems I mentioned. Thanks for that. In the following I will focus on the points only, where I do not agree with the answers which thereby might sound more negative than actually meant.

Page 1 Line 14ff the authors are of course right that the noise level depends on the incident angle. However, there is a pattern of the noise level (e.g. https://doi.org/10.1016/j.procs.2016.09.247 and ESA mentions -25 dB and -22 dB). But if the authors basically use a fixed threshold for the scaling of all scenes, I would like to ask for a reference that suggests doing so or containing information about the variation of noise level over the swath width. It might hold true for some scenes but definitely not for all and would lead to a scaling over a value range containing only e.g. 50% meaning full values or throw away 50% of values containing information in the case the authors showed and used to make the point that there approach is correct . But if the distribution has so many values in in the range below the noise level it is at least worth checking if the calibration routines of Nansat work properly (the scaling itself provides somehow has to provide a reasonable image otherwise, there would not be any data.

**We agree that there are several different noise pattern in the SAR data, but for the sake of simplicity we use fixed thresholds for the scaling. The reference that suggested using the applied thresholds for the feature-tracking approach is Muckenhuber et al. (2016). Muckenhuber et al. (2016) tuned the brightness boundaries to achieve the highest number of vectors and we want to build on this experience, rather than concentrating on a new pre-processing procedure. Angular dependent pre-processing for HH and thermal noise correction for HV has certainly the potential to improve the results significantly and we are looking forward to apply upcoming procedures for our algorithm. For this manuscript however, it is not our main goal to find the best possible pre-processing procedure, since there are other more qualified people working on this topic and we want to concentrate on the algorithm steps after the pre-processing. We checked the calibration routines of Nansat, discussed the topic with people at TU Wien and JPL and did not find any mistakes so far. We included Figure 3 that shows the images with the intensity values and range that we use in the algorithm. The figure has been produced using the presented pre-processing and therefore represents the input matrix for the feature-tracking and pattern-matching step. Most of the values above our considered brightness threshold are over land and therefore not of interest for us. Some low incidence angle HH values over sea ice might have been excluded, but we believe that this is a reasonable compromise for applying a simple (and computationally efficient) pre-processing to which we already have a parameter suggestion from Muckenhuber et al. (2016).**

130 Page 2 Line 40 "... an even better parameter set, because a higher number Is considered ..." It is not the same, it is a completely different approach and therefore not better but just different, providing different information but I'm okay with it.

  **We agree that this is only a different but not necessarily better approach.**

135 Page 2 Line 50 "... more feature tracking vectors are found on image pair with a smaller time span" - given a sufficient number of linear features visible in HV, only and not in the less textures areas where the feature tracking algorithm tends to fail I would like to add but as I said, I guess, that's okay.

  **We agree. This only holds in areas where a sufficient number of corners are detected by the**
140 **feature-tracking algorithm.**

  Page 3 Line 80: I understand the point of the authors and agree with them! Having said this, I'm not happy with it and don't think that things like this should happen since it is bad science and from an outside perspective creates the impression that NERSC has more expertise in the field of sea ice
145 drift than there is actually available plus increases the noise level for publications in this field in general, which makes it more difficult to keep an overview of relevant publications. However, as I stated in my introduction it is up to the editorial board of the Cryosphere to decide.

  **We agree and we are also very unhappy with this development. We will stronger protect our unpublished work in the future.**

150

  Page 4 Line 93: The mentioned independence is not only reduced by overlapping / close features but as well by very characteristic and bright features that might be represented in all resolution levels of the ORB algorithm but I guess that's okay

  **We agree that the independence is also reduced by features that are detected in more than**
155 **one resolution pyramid. We meant to include those features in the phrase 'overlapping'. Defining not only the corner, but the area around the corner (considered by the descriptor) as 'feature', the features from different resolution levels are usually overlapping too.**

  Page 4 Line 105: The feature tracking algorithm chooses the best detectable features automatically
160 but this means that there can also be vast areas with no feature tracking vectors at all if there are enough well defined ridges is one corner of the scene. For the first guess the motion field is now interpolated over these vast "vector"-less areas as well. I have my doubts if this is better or more independent than a fine tuned resolution pyramid in the case of pattern matching. I suggest skipping this comparative statement

165 **We agree. There has been no comparison or proof that would support this sentence. This sentence is rather an idea or motivation for us to follow this new approach than a proven**

**statement and should therefore be skipped. A well tuned pattern-matching approach has certainly the potential to outperform the presented combined algorithm.**

170    Page 5 Line 134 f It is not necessarily true that a scaling that favours feature tracking favours pattern matching as well. 1. The NCC needs no scaling at all since it is normalized (the N of NCC) and 2. Imagine an extreme scaling that would separate ridges in white from the surrounding background in black. For feature tracking, this would be perfect, since it would highlight the linear feature. For pattern matching it would be a problem, especially in the regions outside the ridged areas since it is

175    based on areal pattern variation.

**We agree that the scaling that favours our feature-tracking approach not necessarily favours the considered pattern-matching method. Nevertheless, for the sake of simplicity and computational efficiency, we would like to use the same image matrix for both feature-tracking and pattern-matching. This is now also mentioned in the sentence:**

180    *For the sake of computational efficiency, the same intensity value scaling is used for the pattern-matching step.*

Page 8 Line 242 Thanks for the explanation. Just mention it in the manuscript

**We added the following to the manuscript:**

185    *Sea ice drift might be different on different resolution scales. This is particularly an issue in the case of rotation. The feature-tracking vectors provide the first guess and this vector field should represent the same drift resolution as considered by the pattern-matching step.*

Page 9 Line 291 "We believe ... " I suggest to show it or be more careful with such a statement.

190    The ice drift of the e.g. for example is pure pattern matching without any fancy resolution pyramid and runs pretty fast to provide the ice drift within the framework of CMEMS.

**We agree. There are certainly very reliable pattern-matching procedures that can outperform our combined approach in terms of computational effort, independence and rotation. We were focusing too much on the simple pattern-matching procedure that we implemented**

195    **as part of this algorithm and wanted to highlight the potential advantages of the combined algorithm rather than claiming a real comparison with existing and well tuned pure pattern-matching procedures. This paragraph should be skipped.**

Page 12 Line 409 / Page 13 Line 437 the authors promise to bring feature tracking performance

200    of HH closer to HV by an increased coverage of S1 images and a better pre-processing. I would claim that HH and HV show different characteristics and that's it. You won't change the stronger dependency of linear features in HH polarised images from the incident angle by increasing the

temporal resolution but may see other features. Having said this, a better pre-processing and a higher temporal resolution never hurts and will improve the results for sure.

205 **We agree that HH and HV show different characteristics and that alone could explain the difference in feature-tracking performance. Muckenhuber et al. (2016) found the best HH performance on the image pair with shortest time gap. Since they compared only four image pairs (from different regions), this could certainly be an coincidence. But it could also potentially mean that HH features are more likely to be destroyed over time. However, this is only an as-**
210 **sumption and since the data is not sufficient, we don't want to claim it. We rather wanted to introduce the idea.**

**3 The Manuscript itself**

Most of the remaining points I already handled in my answers to your comments, meaning there is not much left.

215 ### 3.1 Still open points from previous reviews

Page 4 Line 119 f - see as well my comment above: the independence is difficult to assess and its effect on resolving deformation zones is not shown in this paper - may be more careful

**We agree that the general comparison of feature-tracking and pattern-matching in terms of vector independence and effects on resolving shear zones is very difficult and has not been**
220 **done here. We rather wanted to highlight the characteristics of the considered feature-tracking method and the considered pattern-matching method and tried to rephrase the manuscript accordingly. See also the answer to the comment above.**

Page 7 Line 210 - 215 See my comment to your answers
225 **See answer to first comment in Comments on Response to Referee #1**

Page 16 Line 393 I think we had this discussion already in the first draft. I think it is difficult to use a buoy dataset for validation if you used the same dataset beforehand to optimise the setup of the algorithm (Page 16 Line 370-377). It does not result in a real assessment of the performance of the
230 algorithm but in the potentially performance in the case that the algorithm is tuned in the right way to the respective ice conditions.

**Rather than optimising or tuning the algorithm, we wanted to find search restrictions for the search area $t_{2s}$ and rotation $\beta$ that are reasonable for our area and time period. We understand your concerns regarding this and tried to rephrase the manuscript accordingly.**
235 **We changed the following sentences:**
*To assess the potential performance after finding suitable search restrictions, calculated drift*

*results from 246 Sentinel-1 image pairs have been compared to buoy GPS data, collected in 2015 between $15^{th}$ January and $22^{nd}$ April and covering an area from 80.5° N to 83.5° N and 12° E to 27° E. We found a logarithmic normal distribution of the displacement difference with a median at 352.9 m using HV polarisation and 535.7 m using HH polarisation.*

*The algorithm description including data pre-processing is given in Section 3, together with tuning and performance assessment methods. Section 4 presents the pre-processing, parameter tuning and performance assessment results and provides a recommended parameter setting for the area and time period of interest.*

*To evaluate suitable search limitations and assess the potential algorithm performance, we use GPS data from drift buoys that have been set out in the ice covered waters north of Svalbard as part of the Norwegian Young Sea Ice Cruise (N-ICE2015) project of the Norwegian Polar Institute (Spreen and Itkin, 2015).*

*Performance assessment*

*Using the recommended search restrictions from above, the algorithm has been compared to the N-ICE2015 GPS buoy data set (Figure 9) to assess the potential performance after finding suitable search restrictions for the area and time period of interest.*

*The results of the conducted performance assessment are shown in Figure 12.*

*The conducted performance assessment also reveals a logarithmic normal distribution of the distance $D$ (Equation 15) that can be expressed by the following probability density function (solid red line in Figure 12):*

*Based on the restriction evaluation, our experience with the algorithm behaviour, and considering a good compromise between computational efficiency and high quality of the resulting vector field, we recommend the parameter setting shown in Table 1 for our area and time period of interest.*

*To estimate the potential performance of the introduced algorithm for given image pairs, given ice conditions, given region and given time, we compared drift results from 246 Sentinel-1 image pairs with corresponding GPS positions from the N-ICE2015 buoy data set.*

**3.2 Technical comments**

Page 3 Line 62: "On the one hand... on the other hand" - there is "the one hand" missing

**We agree and changed the sentence to:**

*Unlike optical sensors, Space-borne Synthetic Aperture Radar (SAR) are active sensors, operate in the microwave spectrum and can produce high resolution images regardless of solar illumination and cloud cover.*

Page 4 Line 105 since you mention in line 104 that the area has a revisit time of less than one day it is obvious that your area of interest is monitored on a daily basis

**We agree and changed the sentence to:**

*The sea ice covered oceans in the European Arctic Sector represent an important area of interest for the Sentinel-1 mission and due to the short revisit time in the Arctic, our area of interest is monitored by Sentinel-1 on a daily basis (ESA, 2012).*

Page 13 Line 324 "accurate drift" - it not necessary a more accurate drift but merely a more localised/ locally tuned/ locally adapted one. The term "accurate" sounds good but has a meaning which has nothing to do with the case at hand - nevertheless we might hope that a locally adapted drift vector represents the drift at a certain position more accurately than the first guess but that is not necessarily the case.

**We agree and changed the sentence to:**

*We apply pattern-matching at chosen points of interest to adjust the drift and rotation estimate at these specific locations.*

Page 15 Lin 363 "since the uncertainty increases with distance d (Figure 7)" - may be rephrase the sentence - the authors are correct but it suggests that Figure 7 shows the increase of "uncertainty" depending on the distance. Having said this: "uncertainty" is a statistical term - may be choose something less well defined.

**We agree and changed the sentence to:**

*Since the representativeness of the first guess decreases with distance $d$ to the closest feature-tracking vector (an example to illustrate the distribution of $d$ is shown Figure 8), the search restrictions $t_{2s}$ and $\beta$ should increase with $d$.*

Page 15 Line 368 / Page 17 Line 405 - it is more or less the same sentence, may be skip one

**We skipped the sentence on page 15 Line 368 and changed the following sentence to:**

*Based on the performed search restriction evaluation (Section 4), we found the following functions to represent useful restrictions for our area and time period of interest.*

Page 21 Line 482 "accuracy of the introduced algorithm" I suggest to add "for given image pairs, given ice conditions, given region and given time" since the results are not transferable to other ice conditions or regions...

**We agree and changed the sentence to:**

*To estimate the potential performance of the introduced algorithm for given image pairs, given ice conditions, given region and given time, we compared drift results from 246 Sentinel-1 image*

*pairs with corresponding GPS positions from the N-ICE2015 buoy data set.*

310    Please find attached the corrected manuscript with changes marked in blue and red.

Thanks again for your comments. We are looking forward to your reply!

Best regards,

315    S. Muckenhuber and S. Sandven

[revised manuscript text omitted]